# EdgeMask-HGNN: Learning to Sparsify Hypergraphs in Hypergraph Neural Networks

## Abstract

Hypergraph Neural Networks (HGNNs) have achieved remarkable performance in various learning tasks involving hypergraphs. However, existing HGNNs usually treat the input hypergraph as fixed. This can be limiting when the observed incidence structure contains noisy, redundant, or task-irrelevant relations: propagating over all observed relations may introduce harmful messages and hurt test performance. This motivates learning which incidences or hyperedges to retain under an explicit sparsity budget. To address these issues, we propose **EdgeMask-HGNN**, a supervised sparsification method that learns a discrete subhypergraph under an explicit budget constraint. EdgeMask-HGNN offers two distinct sparsification schemes: a fine-grained sparsification and a coarse-grained sparsification, both trained end-to-end using supervision from the downstream task. Extensive experiments on node classification benchmarks demonstrate that EdgeMask-HGNN is effective on heterophilic hypergraphs. On more homophilic datasets, its performance is often comparable to strong baselines. Beyond node classification, an EdgeMask-HGNN adaptation improves Hyper-SAGNN on link prediction benchmarks, with ablations showing that learned context selection is most useful when only part of the available context is helpful for prediction.

## 1 Introduction

Hypergraph Neural Networks (HGNNs) have emerged as powerful tools for learning on hypergraphs, where a single edge can connect multiple nodes (beyond pairwise connections), unlike traditional graphs. There are numerous real-world instances of such relations involving multiple entities: set of researchers collaborating on a paper (Han et al., 2009), set of products purchased in a shopping cart (Xia et al., 2021), group of legislators co-sponsoring a bill (Benson et al., 2018), or set of molecules participating together in biological processes (Gaudelet et al., 2018). A wide range of learning problems arises in hypergraphs– from node classification (Duta et al., 2023) to node clustering (Chodrow et al., 2021) to hyperlink prediction (Yadati et al., 2020). HGNNs have proven to be effective for these tasks, demonstrating strong empirical performance.

Many benchmark and application hypergraphs contain large incidence sets (Table 1) and are often built from observed co-occurrence, similarity, or other construction heuristics rather than from task-certified higher-order interactions (Yadati et al., 2019; Chien et al., 2021; Cai et al., 2022). Such construction can include relations that are redundant or weakly aligned with the target task. In such settings, treating every observed incidence as equally useful can introduce harmful message passing, a claim we later evaluate through controlled noisy-hyperedge experiments. A natural remedy is to sparsify the hypergraph before or during training. Simple strategies include dropping hyperedges uniformly at random (*random sparsification*) or sampling edges proportionally to their degree (Leskovec & Faloutsos, 2006), but these may remove nodes important for downstream tasks and do not preserve spectral properties. *Spectral sparsifiers* sample edges based on hypergraph effective resistance to approximately preserve eigenvalues and eigenvectors (Soma & Yoshida, 2019), yet they may still discard task-relevant hyperedges because they do not exploit label information. In short, existing hypergraph sparsification methods are largely task-agnostic and do not leverage supervision to identify task-relevant higher-order relations.

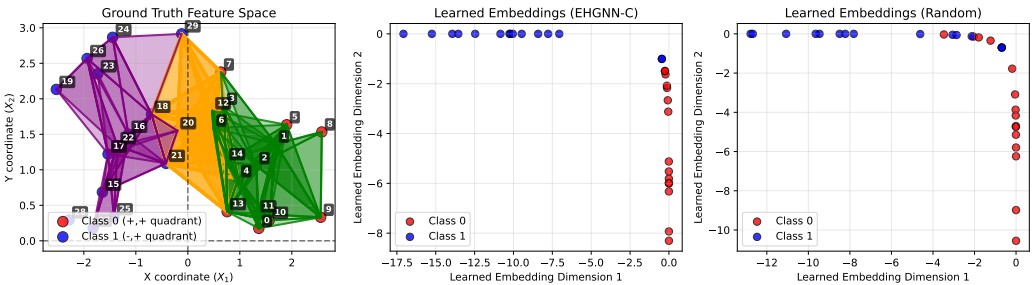

Figure 1: A synthetic 3-uniform hypergraph (number of nodes = 30, number of hyperedges = 134) where (x,y)-coordinates are ground truth node features (left). 50% sparsified hypergraph node embeddings via EdgeMask-HGNN (middle) and via Random sparsification (right). The node embeddings learned by EdgeMask-HGNN are better separable than those of Random. Furthermore, Full training accuracy = Random pruning accuracy = 83.33% while EdgeMask-HGNN accuracy = 100%, illustrating a setting where task-aware sparsification improves the learned representation.

**Why task-aware hypergraph sparsification.** In many applications, hypergraph structures are constructed heuristically and can contain task-irrelevant higher-order relations. Treating the input hypergraph as fixed implicitly assumes that all hyperedges are equally valid for the downstream task, which may not hold. Figure 1 shows one illustrative case where sparsifying the hypergraph improves predictive performance over training on the full hypergraph. More generally, sparsification is useful when removing noisy or task-irrelevant relations helps more than the approximation bias introduced by pruning useful relations. These observations motivate viewing hypergraph sparsification not merely as a heuristic compression technique but as a supervised subhypergraph selection problem, where a task-relevant subset of incidences or hyperedges is selected under a fixed budget. Building on this perspective, we formulate hypergraph sparsification as a supervised discrete mask learning problem subject to an explicitly specified global sparsity budget.

**Our contributions.** (I) We propose EdgeMask-HGNN– a budget-constrained, task-aware hypergraph sparsification method for HGNNs. EdgeMask-HGNN offers two distinct masking strategies: a fine-grained masking (incidence-level) and a coarse-grained masking (hyperedge-level), both trained end-to-end using supervision from the downstream task. The masks are constructed via weighted sampling without replacement, where sampling weights are either parameterized by learnable retention logits at the incidence/edge-level or learned by a feature-conditioned parameterized function. Unlike unsupervised methods (e.g., random, degree-based, and spectral sparsification) that rely on fixed hypergraph structures sampled a priori, EdgeMask-HGNN learns task-dependent retention scores during training and then uses the learned scores to construct a budget-feasible sparse subhypergraph for evaluation. Figure 1 shows that under the same sparsification budget, embeddings learned by EdgeMask-HGNN yield better class separation than the random sparsifier.

(II) We theoretically analyze the stability of the learned sparsifier. To be precise, we show that as the retention logits evolve across epochs, the expected number of changes in the sampled masks is bounded by the magnitude of logit updates. This result characterizes the stability of the sampling process.

(III) Empirical evaluation and analysis show that EdgeMask-HGNN is more effective than alternative sparsifiers such as random, degree-based, and spectral sparsifiers, as well as full hypergraph training on heterophilic hypergraphs. On more homophilic datasets, its performance is often comparable to strong baselines. For link prediction, EdgeMask-HGNN enabled contextual aggregation improves over plain Hyper-SAGNN and random masked context; comparisons with a full-context control show that the best choice between full and learned budgeted context is dataset-dependent. Finally, EdgeMask-HGNN is simple by design and easily adaptable to different HGNN backbones.

**Positioning.** Our method should be understood as a supervised sparsification approach that selects a task-relevant subhypergraph under an explicit budget constraint, rather than a general hypergraph structure learning framework. This distinction is important because our goal is not to reconstruct or augment

the hypergraph, but to identify the most relevant subset of relations for the given learning task. This is particularly important in settings where (i) the observed hypergraph encodes real, interpretable relations (e.g., co-authorship, biological interactions), (ii) adding or hallucinating hyperedges may introduce spurious semantics, and (iii) computational constraints require controlling the number of incidences or edges explicitly.

## 2 Related Works

**Hypergraph Neural Networks (HGNNs).** There are broadly two families of HGNNs: *spectral* and *spatial* (Antelmi et al., 2023). The *spectral HGNNs* construct the hypergraph Laplacian matrix encoding the higher-order relationships, perform filtering operations in the spectral domain using eigenvalues and eigenvectors, and finally transform node features into node embeddings using these spectral filters. The *spatial HGNNs* typically define two-stage message-passing and aggregation: node → hyperedge and hyperedge → node. Despite the apparent difference, the operations performed by certain spectral HGNNs can be interpreted as forms of message passing (Hayhoe et al., 2024). Feng et al. (2019) proposed HGNN by generalizing spectral graph convolutions to hypergraphs through the hypergraph Laplacian. Subsequently, several variants have been proposed, such as HyperGCN (Yadati et al., 2019), and HNHN (Dong et al., 2020). HyperGCN reformulates hypergraph convolution using clique expansion, and HNHN introduces attention mechanisms for hyperedge importance. *spatial* HGNNs typically define two-stage message aggregation: node to hyperedge and hyperedge to node. HyperSAGE (Arya et al., 2020), HGNN+ (Gao et al., 2022), and UniGCN (Huang & Yang, 2021) belong to this category. Recently, Chien et al. (2021) showed that propagation rules of many existing spatial HGNNs can be represented as a composition of two multiset functions, and proposed two different multiset encoding functions: DeepSets and SetTransformer. In addition, several recent frameworks, such as Wang et al. (2025); Saxena et al. (2024), focus on general and expressive message-passing designs for hypergraphs. Our work focuses on learning by identifying and preserving only task-relevant hyperedges, making it a practical augmentation for such general-purpose designs.

**Existing Hypergraph sparsification methods.** Graph sparsification has a long legacy, starting with Benczúr & Karger (1996) for preserving cuts via edge sampling. Subsequently, significant progress was made by Spielman & Teng (2011) and Spielman & Srivastava (2008), who introduced spectral sparsification via effective resistance. These methods ensure that a small subset of edges preserves global graph properties. Among earlier works on hypergraph sparsification, Deveci et al. (2013) proposed hyperedge sampling heuristics to reduce hyperedges while preserving cut structure for hypergraph partitioning tasks. Going beyond heuristics, Soma & Yoshida (2019) extended spectral sparsification to hypergraphs, defining a nonlinear Laplacian quadratic form and constructing $\epsilon$-spectral sparsifiers of size $\mathcal{O}(n^3/\epsilon^2)$ in polynomial time. Recently, Kapralov et al. (2022) proved a significant result showing that it is possible to obtain an $\epsilon$-spectral sparsifier of linear size. These methods are unsupervised, as they do not incorporate node labels into their sparsification decisions. As a result, they may not be suitable for representation learning tasks that require preserving only task-relevant hyperedges.

Among task-aware methods, HSL (Cai et al., 2022) proposes to reconstruct and enhance the input hypergraph by jointly pruning and adding node-hyperedge connections under no global sparsity constraint. In contrast, EdgeMask-HGNN focuses on selecting a task-relevant *subhypergraph* from a given hypergraph under an *explicit global sparsity budget*. Our method does not augment connectivity, as in many practical settings, such as recommendation systems or scientific data analysis, modifying or hallucinating higher-order relations can be undesirable or difficult to operationalize. Consequently, EdgeMask-HGNN is designed to be complementary to HSL, prioritizing controllable sparsity and scalability over structural reconstruction. To the best of our knowledge, there is no prior purely pruning-based supervised hypergraph sparsifier with the same explicit budgeted-subhypergraph objective. HSL is therefore the closest task-aware baseline.

**Graph structure learning and sparsification.** Several works study sparsification of neural network parameters, such as the unified lottery ticket hypothesis for GNNs (Chen et al., 2021; Hui et al., 2023; Liao et al., 2025) and graph gradual pruning (Liu et al., 2023a). These methods jointly prune GNN weights and graph edges to reduce parameter count and training cost, with the primary goal of model compression. Our work is complementary: we keep the HGNN parameters dense while learn a structural sparsifier that

selectively retains a subset of node-hyperedge incidences. There have also been several notable recent works regarding degree-based spectral sparsification (Liu et al., 2023b) and supervised graph sparsification (Zheng et al., 2020; Ye & Ji, 2021; Zhang et al., 2025). Unlike graph pruning methods that operate on pairwise edges, our method is 'higher-order' in the sense that it operates on hypergraph incidences.

While supervised sparsification and edge masking have been explored in GNNs Ye & Ji (2021), extending these ideas to hypergraphs is nontrivial due to the higher-order incidence structure. In particular, hyperedges induce node-to-edge bipartite incidence relations rather than node-to-node pairwise relations, giving rise to distinct sparsification granularities at the incidence and hyperedge levels. Our method explicitly models these two levels of sparsification, which have no direct analogue in standard graph settings. EdgeMask-HGNN learns discrete hypergraph masks using sigmoid retention weights, budgeted sampling without replacement, and a masked straight-through estimator. These choices are part of the broader family of differentiable discrete-selection methods, including Gumbel/Concrete relaxations (Jang et al., 2017; Maddison et al., 2017), hard-concrete gates (Louizos et al., 2018), and straight-through estimators (Bengio et al., 2013). The selected object, however, is different: our mask selects a budget-feasible subhypergraph in the original incidence representation, rather than pairwise graph edges, attention coefficients, or neural-network weights. This distinction matters because a hyperedge-level mask removes an entire higher-order relation, whereas an incidence-level mask changes which nodes participate in a retained relation. These operations change hyperedge cardinalities and the two-stage HGNN normalization, and they cannot be represented faithfully by independently pruning edges in a clique expansion.

**Relation to graph signal processing and graph learning.** There is also a substantial graph signal processing literature on learning graph topology from smooth signals or spectral templates (Kalofolias, 2016; Dong et al., 2016; Segarra et al., 2017; Mateos et al., 2019; Dong et al., 2019). These methods usually infer a weighted graph Laplacian or adjacency matrix under smoothness, sparsity, or spectral constraints. EdgeMask-HGNN differs in both object and supervision: it starts from an observed hypergraph, selects a discrete subhypergraph under an explicit incidence or hyperedge budget, and optimizes the mask using downstream task loss. Incorporating such structural priors into supervised hypergraph sparsification is an interesting direction for future work (SarcheshmehPour et al., 2023).

**Why hypergraph sparsification is not a direct graph-edge analogue.** Extending supervised graph sparsification to hypergraphs raises several concrete issues. First, the maskable unit is ambiguous: one may remove an entire hyperedge, or remove only selected node–hyperedge incidences inside it. These choices are identical in ordinary graphs but different in hypergraphs. Second, the two choices have different semantics. Hyperedge masking deletes a whole higher-order relation among a set of nodes, whereas incidence masking edits the membership of a relation and can repair a partially corrupted hyperedge without discarding all of it. Third, both choices change HGNN message passing differently: incidence masking changes hyperedge cardinalities, node degrees, and the two-stage node-to-hyperedge and hyperedge-to-node normalization, while hyperedge masking preserves each retained relation but removes all messages through deleted hyperedges. Fourth, clique expansion does not preserve this distinction. A single hyperedge becomes many pairwise graph edges, so pruning one clique edge does not correspond to either deleting the original higher-order relation or editing one node's membership in that relation. These issues make the feasible set, normalization, and inductive bias of hypergraph sparsification different from pairwise graph-edge pruning, motivating the incidence-level and hyperedge-level programs in Section 3.

*A formal way to see the mismatch.* Consider a hyperedge $e = \{v_1, \ldots, v_r\}$ with $r > 2$. In a clique expansion, $e$ is represented by the $\binom{r}{2}$ pairwise edges among its nodes. A hyperedge-level mask has only two feasible decisions for this relation: keep all $\binom{r}{2}$ pairwise edges induced by $e$, or delete all of them. An incidence-level mask can instead remove one membership $(v_i, e)$, which changes the relation to $e \setminus \{v_i\}$ and corresponds to deleting exactly the $r - 1$ pairwise edges incident to $v_i$ inside the clique while keeping the pairwise edges among $e \setminus \{v_i\}$. A generic graph-edge mask on the clique has $2^{\binom{r}{2}}$ feasible pairwise subsets, most of which do not correspond to either a valid hyperedge deletion or a valid incidence-edited hyperedge. Conversely, once the hyperedge is expanded into pairwise edges, the identity of the original incidence $(v_i, e)$ is not available unless one adds extra group constraints tied to the original hyperedge. Thus graph-edge sparsification on a clique expansion optimizes over a different feasible set from both Equations (1) and (2).

# 3 Problem Statement

Let $\mathcal{H} \triangleq (V, E, \boldsymbol{X}) \equiv (\boldsymbol{H}, \boldsymbol{X})$ be a hypergraph, where $V$ is the set of nodes, $E$ is the set of hyperedges, $\boldsymbol{H} \in \{0,1\}^{|V| \times |E|}$ is the incidence matrix of $\mathcal{H}$, and $\boldsymbol{X} \in \mathbb{R}^{n \times F}$ is the node feature matrix. A hyperedge $e \in E$ is a subset of $V$, i.e., $H_{v,e} = 1$ if $v \in e$, or 0 otherwise. Throughout the paper, a hyperedge is a subset of nodes, possibly containing more than two nodes. Let $\mathcal{Y}$ denote the output space, $\mathcal{S}$ denote the set of supervised elements (e.g., nodes, hyperedges), and $\mathcal{S}_L \subseteq \mathcal{S}$ denote the labeled subset with labels $\boldsymbol{y}_L$. The goal of a supervised learning task on $\mathcal{H}$ is to learn a model $f_\theta : \mathcal{S} \to \mathcal{Y}$ that predicts labels for elements $s \in \mathcal{S}$ by minimizing an empirical task loss: $\theta^* = \arg\min_\theta \mathcal{L}_{task}(f_\theta(s; \boldsymbol{H}, \boldsymbol{X}), \boldsymbol{y}_L)$. This formulation is task-agnostic and applies to node classification, hyperedge classification, and link prediction tasks.

**Budget-constrained task-aware sparsification.** Let $t = \sum_{e \in E} |e|$ denote the total number of incidence pairs before sparsification. Since a hyperedge can contain more than two nodes, the budget constraint can be applied at two levels: an incidence budget $k$, the number of node–hyperedge incidence pairs retained after sparsification, or a hyperedge budget $\kappa$, the number of hyperedges retained after sparsification.

Given an incidence budget $k$, the goal is to jointly learn HGNN parameters and a binary incidence mask $\hat{\boldsymbol{M}} \in \{0,1\}^{|V| \times |E|}$ that selects a task-relevant subhypergraph:

$$(\theta^*, \hat{\boldsymbol{M}}^*) \in \arg\min_{\theta, \hat{\boldsymbol{M}}} \mathcal{L}_{\text{task}}(f_\theta(s; \boldsymbol{H} \odot \hat{\boldsymbol{M}}, \boldsymbol{X}), \boldsymbol{y}_L) \quad \text{s.t.} \quad \hat{M}_{v,e} = 0 \text{ if } H_{v,e} = 0, \quad \|\hat{\boldsymbol{M}}\|_0 \le k. \tag{1}$$

For hyperedge-level sparsification, we instead learn a binary edge mask $\hat{\mathbf{m}} \in \{0,1\}^{|E|}$:

$$(\theta^*, \hat{\mathbf{m}}^*) \in \arg\min_{\theta, \hat{\mathbf{m}}} \mathcal{L}_{\text{task}}(f_\theta(s; \boldsymbol{H} \odot \mathbf{1}_{|V|} \hat{\mathbf{m}}^\top, \boldsymbol{X}), \boldsymbol{y}_L) \quad \text{s.t.} \quad \|\hat{\mathbf{m}}\|_0 \le \kappa. \tag{2}$$

The downstream task loss can be any differentiable loss depending on the supervised task. For example, for classification tasks:

$$\mathcal{L}_{task} = \frac{1}{|\mathcal{S}_L|} \sum_{s \in \mathcal{S}_L} \ell(\boldsymbol{y}_s, \hat{\boldsymbol{y}}_s),$$

where $\hat{\boldsymbol{y}}_s = f_\theta(s; \tilde{\boldsymbol{H}}, \boldsymbol{X})$ denotes the prediction for the supervised element $s \in \mathcal{S}_L$.

**Remarks.** Equations (1)–(2) explicitly optimize over a discrete mask under a fixed sparsity budget constraint. Hence, EdgeMask-HGNN is to be viewed as a differentiable, supervised approximation to these constrained programs. EHGNN-F and its feature-conditioned variants approximate the incidence-level program in Equation (1); EHGNN-C and its feature-conditioned variants approximate the hyperedge-level program in Equation (2). Random, degree-based, and spectral sparsifiers are task-agnostic budget-feasible comparators: they choose a feasible mask without using the downstream task gradient.

The constrained objective also explains why supervised sparsification can prefer task-aligned relations. For fixed HGNN parameters, replacing a retained incidence or hyperedge by another feasible unit is preferred by the empirical program only if it lowers the supervised loss. During training, the masked straight-through estimator sends gradient only through selected incidences or hyperedges. Thus the retained units that help reduce task loss receive score updates that make them more likely to be selected again, while retained units that increase task loss receive the opposite pressure. Under the budget, this creates competition among candidate incidences or hyperedges according to their downstream loss contribution. This is not a guarantee that sparsification always improves test performance: the budget $k$ or $\kappa$ induces a trade-off between removing task-irrelevant or noisy relations and discarding useful weak relations. Thus the expected benefit of sparsification is conditional on the structure and noise level of the input hypergraph. We evaluate this trade-off empirically through train–test gap analysis and controlled noisy-hyperedge experiments in Appendix G.

# 4 EdgeMask-HGNN: Learnable Hypergraph Sparsification

In this section, we introduce EdgeMask-HGNN, a task-aware, learnable sparsification framework for hypergraph neural networks. The main idea is to devise a differentiable module that can learn to selectively

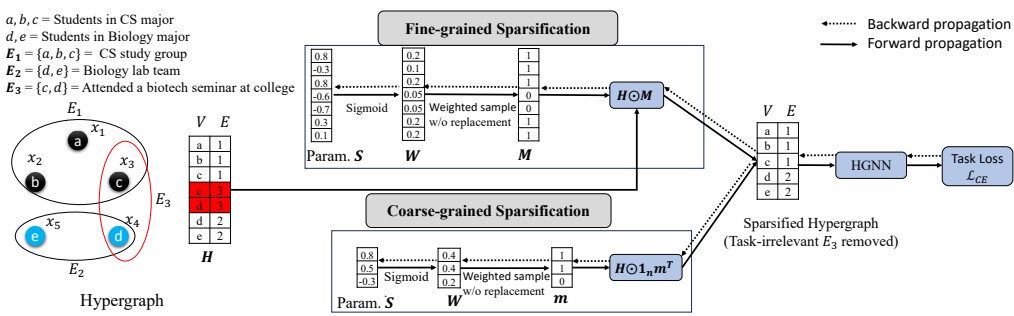

Figure 2: Schematic of EHGNN-F (top) and EHGNN-C (bottom) with a small example.

mask hyperedges based on their relevance to the downstream learning task. Unlike prior hypergraph prun-ing methods that rely on fixed heuristics, EdgeMask-HGNN is trained end-to-end, allowing the model to jointly optimize the hypergraph structure and the HGNN parameters under a strict global sparsity budget constraint.

Sparsification can be applied at two structural levels of a hypergraph: (i) individual node–hyperedge inci-dences and (ii) entire hyperedges. Thus, two distinct masking schemes emerge: *Incidence-level masking* and *Edge-Level masking*. Incidence-level masking learns an incidence-level mask by sampling incidences based on incidence-level weights, whereas Edge-Level masking learns an edge-level mask by sampling edges based on edge-level weights. These weights are either parameterized by independent learnable logits or learned by a shared parameterized function. Finally, based on the task loss of the downstream HGNN, the gradients are backpropagated to compute the updated masks as well as the HGNN parameters. Figure 2 illustrates a schematic of EdgeMask-HGNN parameterized by independent learnable logits with a small example.

### 4.1   Fine-grained Masking (EHGNN-F)

Fine-grained incidence-level masking focuses on learning a soft importance weight for individual node–edge connections so as to achieve a fine-grained control over the sparsified hypergraph structure. For each incidence pair $(v, e)$, we maintain a learnable logit $s_{v,e}$, which is converted to a non-negative weight via sigmoid transformation

$$w_{v,e} = \sigma(s_{v,e}) \tag{3}$$

where $\sigma$ is the sigmoid function. Using a sigmoid to bound a learned scalar gate is a standard neural parameterization; related bounded-gate parameterizations appear in earlier work such as Trask et al. (2018).

Let $\boldsymbol{w} := (w_{v,e})$ denote the vector of weights over all incidence pairs. To construct a sparsified hypergraph with exactly $k$ active incidences, we sample $k$ distinct incidence pairs without replacement according to these weights. This sampling procedure is equivalent to drawing the first $k$ elements of a permutation generated by a *Plackett–Luce (PL) distribution*(Luce et al., 1959) parameterized by $\mathbf{w}$. Specifically, if $(i_1, \ldots, i_k)$ denote the sampled ordered indices corresponding to incidence pairs, their probability under the PL model is

$$P(i_1, \ldots, i_k) = \prod_{t=1}^{k} \frac{w_{i_t}}{\sum_{j \notin \{i_1, \ldots, i_{t-1}\}} w_j}.$$

The resulting hard incidence mask $\hat{\mathbf{m}}$ is defined as

$$\hat{m}_{v,e} = \begin{cases} 1 & \text{if } (v,e) \text{ is selected,} \\ 0 & \text{otherwise,} \end{cases} \qquad \sum_{v,e} \hat{m}_{v,e} = k.$$

Thus, the sparsifier samples exactly $k$ active incidences according to a *weighted sampling-without-replacement scheme*, which can be viewed as the top-$k$ marginal of a Plackett–Luce ranking distribution. Let $\hat{M} \in$

$\{0,1\}^{n \times m}$ denote the binary incidence mask whose entries are $\hat{m}_{v,e}$. The sparsified hypergraph is obtained using the *hard mask* $\hat{M}$ in the forward pass:

$$\tilde{H} = H \odot \hat{M}, \tag{4}$$

while gradients are propagated through the surrogate mask $m$ defined by the straight-through estimator (Equation (5)).

**Differentiability.** The sampling step used to construct the incidence mask is discrete and therefore non-differentiable. To enable gradient-based optimization, we employ a masked straight-through estimator (STE) (Bengio et al., 2013). In the forward pass, a hard binary mask $\hat{m}_{v,e} \in \{0,1\}$ is produced by sampling $k$ incidences according to the weights $w_{v,e} = \sigma(s_{v,e})$. In the backward pass, we define

$$m_{v,e} = \hat{m}_{v,e} + \big(w_{v,e} - \text{stop\_grad}(w_{v,e})\big)\hat{m}_{v,e}, \tag{5}$$

so that $\partial m_{v,e}/\partial w_{v,e} = \hat{m}_{v,e}$. Consequently, gradients are propagated only through $k$ incidences selected by the hard mask, while unselected incidences receive zero gradient during that iteration. By the chain rule, the gradient with respect to the logit satisfies

$$\frac{\partial \mathcal{L}}{\partial s_{v,e}} = \frac{\partial \mathcal{L}}{\partial \tilde{H}_{v,e}} \, H_{v,e} \, \hat{m}_{v,e} \, w_{v,e}(1 - w_{v,e}).$$

This estimator provides a practical surrogate for backpropagation through the discrete masking operation: the forward computation remains exactly sparse, while the backward pass uses the continuous weights $w_{v,e}$ to propagate gradients to the scorer parameters. Our implementation uses sigmoid retention weights, budgeted sampling without replacement, and the masked STE above. These are related to a broader family of differentiable discrete-selection tools, including Gumbel/Concrete relaxations and hard-concrete gates (Bengio et al., 2013; Jang et al., 2017; Maddison et al., 2017; Louizos et al., 2018; Trask et al., 2018), but the modeling distinction is that the hard forward-pass object is an exactly budgeted subhypergraph in the original incidence matrix. Thus, EHGNN-F learns competition among node–hyperedge incidences, while EHGNN-C learns competition among whole hyperedges; both are coupled to HGNN message passing through the retained incidence structure and its normalization. This differs from graph-edge masking, hard attention over pairwise neighborhoods, and neural parameter pruning, where each mask entry usually corresponds to a pairwise edge, neighbor weight, or model parameter rather than membership in a higher-order relation. Although the estimator is biased relative to the true gradient of the discrete sampling process, it has been widely adopted in discrete neural architectures (Zheng et al., 2020) and works well empirically.

**Variant of EHGNN-F conditioned on node features.** EHGNN-F is feature-agnostic, each incidence pair $(v, e)$ is assigned an independent learnable parameter $s_{v,e}$. While flexible, this parameterization does not share information across incidence pairs and may generalize poorly when supervision is limited. To address this issue, feature-conditioned EHGNN-F variant **EHGNN-F (cond.)** uses a shared parameter *scorer* module that learns $s_{v,e}$ as functions of node features:

$$s_{v,e} = \texttt{MLP}(\boldsymbol{X}_v || \hat{\boldsymbol{X}}_e), \tag{6}$$

where $\boldsymbol{X}_v$ is the feature vector of node $v$ and the aggregated embedding of hyperedge $e$ is $\hat{\boldsymbol{X}}_e = \frac{1}{|e|} \sum_{v \in e} \boldsymbol{X}_v$. The shared MLP parameters allow limited labeled nodes to shape sparsification decisions across the whole hypergraph, alleviating the lack of supervision issue.

## 4.2 Coarse-grained Masking (EHGNN-C)

Although Incidence-level masking allows fine-grained control, it may result in a large parameter size since the network keeps one parameter per incidence pair. Coarse-grained Edge-Level masking alleviates this by maintaining one logit parameter per edge. For instance, if the input hypergraph is $k$-uniform (every edge contains $k$ nodes), edge-level masking reduces the model parameters by $k$.

The learnable edge score parameter is converted to non-negative edge weights via sigmoid transformation:

$$w_e = \sigma(s_e) \tag{7}$$

To construct a sparsified hypergraph with exactly $\kappa$ edges, we sample $\kappa$ edges without replacement according to weights $\boldsymbol{w}_e$ via PL sampling procedure. The sampled hard edge mask is defined as

$$\hat{m}_e = \begin{cases} 1 & \text{if } e \text{ is selected,} \\ 0 & \text{otherwise,} \end{cases} \qquad \sum_e \hat{m}_e = \kappa.$$

To backpropagate through this discrete edge selection, similar to EHGNN-F, we use a masked STE: $m_e = \hat{m}_e + (w_e - \text{stop\_grad}(w_e))\hat{m}_e$. This allows gradients to propagate only through the selected edges at that epoch. The sparsified incidence matrix becomes:

$$\tilde{\boldsymbol{H}} = \boldsymbol{H} \odot \boldsymbol{1}_n \boldsymbol{m}^T, \tag{8}$$

where $\boldsymbol{1}_n \in \mathbb{R}^n$ is a vector of ones, $\boldsymbol{m}^T \in \mathbb{R}^{1 \times m}$ is the binary mask.

**Variant of EHGNN-C conditioned on node features.** EHGNN-C is feature-agnostic, each edge has a free learnable parameter $s_e$, which do not depend on its constituent node features. This feature-agnostic formulation may limit supervision propagation in semi-supervised settings. To address this issue, feature-conditioned EHGNN-C (**EHGNN-C (cond.)**) learns a shared scorer as a parameter of node features using a permutation-invariant operator, such as mean pooling:

$$\hat{\boldsymbol{X}}_e = \frac{1}{|e|} \sum_{v \in e} \boldsymbol{X}_v, \quad s_e = \texttt{MLP}(\hat{\boldsymbol{X}}_e) \tag{9}$$

### 4.3  Issues with Feature Conditioned Variants and Low-Rank Reparameterization

The MLP-based scorer defined in Equations (6) and (9) is memory-intensive on large-scale hypergraphs. For instance, EHGNN-F (cond) defines incidence scores by evaluating $w_{v,e} = \sigma\left(\texttt{MLP}_\theta(\boldsymbol{X}_v || \hat{\boldsymbol{X}}_e)\right)$, where $\boldsymbol{X}_v \in \mathbb{R}^F$ is the feature of node $v$, $\hat{\boldsymbol{X}}_e$ is hyperedge representation pooled from $\{\boldsymbol{X}_v : v \in e\}$. When the number of incidences $t$ is large, evaluating $\texttt{MLP}_\theta$ across all incidences induces high activation memory. With a two-layer MLP scorer with hidden width $h$, EHGNN-F(cond) requires $\approx \Theta(tF + th)$ activation storage, since both the concatenated incidence input $[X_v \| \hat{X}_e] \in \mathbb{R}^{2F}$ and the hidden layer activations must be retained for backpropagation. To address this, we adopt a lightweight low-rank parameterization of EdgeMask-HGNN.

**Low-Rank feature-conditioned incidence scoring: EHGNN-F(cond,LR).** To obtain a scalable scorer while preserving the conditioning on node features, we replace $\texttt{MLP}_\theta$ with a rank-$r$ bilinear parameterization:

$$\boldsymbol{S} = \boldsymbol{U}\boldsymbol{Z}^\top,$$

where $\boldsymbol{U}$ and $\boldsymbol{Z}$ are the latent node factors and edge factors, respectively. The node factor for node $v$ and the edge factor for edge $e$ are constructed as

$$\boldsymbol{u}_v = \boldsymbol{W}_x^\top \boldsymbol{X}_v \in \mathbb{R}^r, \qquad \boldsymbol{z}_e \in \mathbb{R}^r, \tag{10}$$

where $\boldsymbol{W}_x \in \mathbb{R}^{F \times r}$ is a learnable projection matrix and $r << F$. Finally, the incidence score and weights are computed as

$$s_{v,e} = \langle \boldsymbol{u}_v, \boldsymbol{z}_e \rangle, \qquad w_{v,e} = \sigma(s_{v,e}). \tag{11}$$

This parameterization constrains the unnormalized score matrix $\boldsymbol{S}$ to satisfy $rank(\boldsymbol{S}) \leq r$. Unlike EHGNN-F(cond), we do not compute pooled hyperedge features, thereby avoiding the $\Theta(tF)$ cost of aggregating node features for each incidence.

In terms of expressivity, Equation 11 can be treated as a restricted low-rank parameterization of the incidence scoring function $\texttt{MLP}_\theta(.)$ used in EHGNN-F(cond), which trades expressive power for improved computational

and memory efficiency. The score computation still depends on node features $\boldsymbol{x}_v$, each hyperedge retains a learnable embedding $\boldsymbol{z}_e$, and the masking and straight-through estimator remain unchanged. In terms of activation memory, EHGNN-F(cond,LR) stores $\Theta(|V|r + tr)$ activations, corresponding to storing projected node features ($\boldsymbol{u}_v$) and edge factors ($\boldsymbol{z}_e$). Hence, the scorer activation-memory is reduced by roughly $\frac{tF + th}{|V|r + tr} \approx F/r$ when $t >> |V|$ and $F >> h$. The low-rank scorer can also regularize mask learning by tying incidence scores through shared node factors $\boldsymbol{u}_v$ and hyperedge factors $\boldsymbol{z}_e$. A more flexible MLP scorer can assign highly specific scores to many individual incidences from concatenated node–hyperedge features. When supervision is limited relative to the number of incidences, such flexibility can fit incidental or noisy scoring patterns. The low-rank form reduces the effective number of independent scoring patterns: incidences that share a node or hyperedge share part of the scorer. This coupling can improve generalization of the learned mask and helps explain why EHGNN-F(cond,LR) can sometimes outperform the more expressive EHGNN-F(cond).

**Low-rank feature-conditioned edge scoring: EHGNN-C(cond,LR).** Similar to EHGNN-F(cond,LR), we compute the latent hyperedge factor $\boldsymbol{z}_e$ and node factors $\boldsymbol{u}_v$ as in Equation 10. The projected node embeddings are then aggregated via mean-pooling to construct hyperedge embedding:

$$\boldsymbol{u}_e = \phi(\{\boldsymbol{u}_v \mid v \in e\}) \in \mathbb{R}^r. \tag{12}$$

Finally, instead of using `MLP`, we use a rank-$r$ bilinear form as a scorer:

$$s_e = \langle \boldsymbol{u}_e, \boldsymbol{z}_e \rangle, \qquad w_e = \sigma(s_e). \tag{13}$$

This produces hyperedge-level weights similar to original EHGNN-C (cond) but avoids the cost of evaluating an MLP on each hyperedge.

### 4.4 Training and Inference.

The sparsified incidence $\tilde{\boldsymbol{H}}$ is passed into any HGNN architecture (e.g., HyperGCN, UniGNN, etc.) and trained end-to-end with a task loss $\mathcal{L}_{\text{task}}$. This yields a sparse hypergraph $\tilde{\boldsymbol{H}}$ adapted to the node classification task. During inference, instead of stochastic sampling, we use deterministic top-$k$ (or top-$\kappa$) selection from $\boldsymbol{w}$ for stability and reproducibility.

**Degenerate hyperedges during sparsification.** Incidence-level masking may create degenerate hyperedges such as singleton hyperedges, empty hyperedges, and isolated nodes. Singleton hyperedges remain well-defined in the bipartite incidence representation; thus they are handled by the same message-passing rule as other retained hyperedges. Empty hyperedges simply have no retained incidences and therefore contribute no messages. In the implementation, message passing is carried out over the retained incidence list; if the retained list is empty, the layer returns a zero aggregate with the learned bias, and inverse degree normalizers that would otherwise divide by zero are set to zero. Thus isolated nodes receive no hypergraph message from the sparsified structure rather than producing invalid normalization values.

## 5 Theoretical Analysis of Mask Stability

In the following, we analyze the stability of the discrete sampler, not global optimization, recovery of an optimal subhypergraph, or downstream generalization. Since the logits often evolve gradually after early training, we study how small logit changes affect the stochastic masks produced by PL sampling. The theorems show that the expected number of mask flips is controlled by the magnitude of logit updates. Full proofs are provided in Appendix A.

**Theorem 5.1** (Epoch-to-epoch stability of fine-grained masks)**.** *Let $t = \sum_{e \in E} |e|$ be the number of incidence pairs and $k \in \{1, \ldots, t\}$ be a fixed incidence pair budget. Let $\boldsymbol{s}^{(\tau)}, \boldsymbol{s}^{(\tau+1)} \in \mathbb{R}^t$ denote the logit vectors at two consecutive epochs of EHGNN-F, and incidence weights*

$$\boldsymbol{w}^{(\tau)} := \sigma(\boldsymbol{s}^{(\tau)}), \quad \boldsymbol{w}^{(\tau+1)} := \sigma(\boldsymbol{s}^{(\tau+1)}).$$

Let $S^{(\tau)}$ and $S^{(\tau+1)}$ denote the $k$-subsets obtained by sampling from the Plackett–Luce distributions $\mathrm{PL}(\boldsymbol{w}^{(\tau)})$ and $\mathrm{PL}(\boldsymbol{w}^{(\tau+1)})$ respectively.

We assume that there exists $\alpha > 0$ such that for all epoch $\tau$ and all $i \in [t]$, $w^{(\tau)}(i) \geq \alpha$. Then the expected number of incidence mask flips between the two consecutive epochs satisfies

$$\mathbb{E}\Big[|S^{(\tau+1)}\triangle S^{(\tau)}|\Big] \leq \frac{k}{2\alpha} \ln \frac{t}{t-k} \, \|\boldsymbol{s}^{(\tau+1)} - \boldsymbol{s}^{(\tau)}\|_1.$$

**Theorem 5.2** (Epoch-to-epoch stability of coarse-grained masks)**.** *Let $m = |E|$ be the number of hyperedges and $\kappa \in \{1,\ldots,m\}$ be a fixed edge budget. At iteration $\tau$, let the edge logits $\boldsymbol{s}_e^{(\tau)} \in \mathbb{R}^m$ and edge weights*

$$w_e^{(\tau)}(e) := \sigma(s_e^{(\tau)}(e)) \in (0,1), \qquad e \in [m].$$

*EHGNN-C samples a permutation $\Gamma^{(\tau)} \sim \mathrm{PL}(\boldsymbol{w}_e^{(\tau)})$ and selects the first $\kappa$ edges*

$$T^{(\tau)} := \{\Gamma_1^{(\tau)}, \ldots, \Gamma_\kappa^{(\tau)}\} \subseteq [m].$$

*We assume that there exists $\alpha_c > 0$ such that $w_e^{(\tau)}(e) \geq \alpha_c$ for all $\tau, e$. Then the expected number of edge mask flips between consecutive epochs satisfies*

$$\mathbb{E}\Big[|T^{(\tau+1)}\triangle T^{(\tau)}|\Big] \leq \frac{\kappa}{2\alpha_c} \ln \frac{m}{m-\kappa} \, \|\boldsymbol{s}_e^{(\tau+1)} - \boldsymbol{s}_e^{(\tau)}\|_1.$$

**Immediate consequence.** If the retention logits enter a stationary regime, i.e., $\|\boldsymbol{s}^{(\tau+1)} - \boldsymbol{s}^{(\tau)}\|_1 \to 0$ for EHGNN-F or $\|\boldsymbol{s}_e^{(\tau+1)} - \boldsymbol{s}_e^{(\tau)}\|_1 \to 0$ for EHGNN-C, then Theorems 5.1–5.2 imply that the expected number of mask flips also vanishes. This is not a proof that gradient descent converges, but it gives a necessary consistency condition for joint training to settle on a stable learned subhypergraph. Thus, when the retention scores stabilize, the sampled sparse hypergraph also stabilizes in expectation.

**Discussion.** (i) The above results imply that if the logit vector changes slowly across epochs, the expected number of mask changes also becomes small. Thus, the stochastic sparsification procedure stabilizes during training. Operationally, this matters because stochastic masks add a sampler-induced source of variation to the HGNN loss and gradients: two consecutive epochs may train on different sparse hypergraphs. By tying expected mask changes to logit movement, the theorem controls this source of instability once the scorer stabilizes. This also motivates our deterministic top-$k$ or top-$\kappa$ inference rule, which removes residual sampling noise and evaluates the stable high-score subhypergraph. (ii) The assumption of PL weights being lower-bounded is mild in our setting, since the mask logits are produced by a learnable scoring function applied to node and hyperedge representations. When input features are bounded and model parameters remain bounded during training, the resulting logits are also bounded. In particular, if $s^{(\tau)}(i) \geq -B$ for all $i$, then $w^{(\tau)}(i) = \sigma(s^{(\tau)}(i)) \geq \sigma(-B)$. (iii) These results are sampler-stability guarantees. They do not imply optimization convergence, recovery of an optimal subhypergraph, generalization performance, or a direct performance improvement; rather, they explain why the learned discrete structure becomes empirically stable once the retention logits stabilize (see Appendix J).

**Theorem 5.3** (One-Swap Local Optimality Check)**.** *Let $U$ be the set of maskable units, namely incidences for EHGNN-F or hyperedges for EHGNN-C, and let $S^\star \subset U$ be a retained set of budget $B$. Define the one-swap neighborhood*

$$\mathcal{N}(S^\star) = \{(S^\star \setminus \{i\}) \cup \{j\} : i \in S^\star, \ j \in U \setminus S^\star\}.$$

*Fix trained HGNN parameters $\hat{\theta}$, and let*

$$G(S) = \mathcal{L}_{val}\big(f_{\hat{\theta}}(H \odot M_S, X), y_{val}\big).$$

*For tolerance $\epsilon \geq 0$, let $\rho_\epsilon(S^\star)$ be the fraction of one-swap neighbors $S' \in \mathcal{N}(S^\star)$ satisfying $G(S') \leq G(S^\star) - \epsilon$. If $Q$ one-swap neighbors are sampled uniformly with replacement and $B_Q$ of them are $\epsilon$-improving, then with probability at least $1 - \delta$,*

$$\rho_\epsilon(S^\star) \leq U_\delta(B_Q, Q),$$

where $U_\delta(B_Q, Q)$ is the one-sided Clopper–Pearson upper confidence bound. In particular, if $B_Q = 0$, then

$$\rho_\epsilon(S^\star) \leq 1 - \delta^{1/Q} \leq \frac{\log(1/\delta)}{Q}.$$

Theorem 5.3 does not establish global optimality. It only interprets the one-swap local optimality check statistically: if few sampled swaps reduce validation loss by more than $\epsilon$, then, with high confidence, only a small fraction of one-swap neighbors would provide such an improvement.

## 6 Experiments

We evaluate EdgeMask-HGNN on semi-supervised node classification task in the transductive setting as well as on the link prediction task. On the node classification task, we randomly split the nodes into training/validation/test samples using 50%/25%/25% splitting percentages (Chien et al., 2021). Unless stated otherwise, we have used HGNN (Feng et al., 2019) as a backbone in our implementation. We follow Chien et al. (2021) to set the hyperparameters of various base HGNN models. We average the results of 10 experiments using multiple random splits and initializations. All experiments are run on a system with 512 GB RAM and a single NVIDIA Tesla V100 GPU with 32 GB memory. All algorithms run for 1000 epochs except Trivago (2000 epochs) with a retention rate of $k = 50\%$ ($\kappa = 50\%$).

**Datasets.** The hypergraph datasets and their statistics are presented in Table 1. It includes recently proposed heterophilic benchmarks: Actor, Twitch, and Pokec from Li et al. (2025). The rest of the datasets

Table 1: Hypergraph dataset statistics used in the node-classification experiments. Dataset names shown in blue have edge homophily $\mathcal{H}_e \leq 0.5$.

| | Cora | Citeseer | Pubmed | Cora-CA | DBLP-CA | 20News | Mushroom | NTU2012 | ModelNet40 | Yelp | House | Walmart | Actor | Twitch | Pokec | Trivago |
|---|---|---|---|---|---|---|---|---|---|---|---|---|---|---|---|---|
| $|V|$ | 2708 | 3312 | 19717 | 2708 | 41302 | 16242 | 8124 | 2012 | 12311 | 50758 | 1290 | 88860 | 16255 | 16812 | 14998 | 172738 |
| $|E|$ | 1579 | 1079 | 7963 | 1072 | 22363 | 100 | 298 | 2012 | 12311 | 679302 | 341 | 69906 | 10164 | 2627 | 2406 | 233202 |
| Number of features | 1433 | 3703 | 500 | 1433 | 1425 | 100 | 22 | 100 | 100 | 1862 | 100 | 100 | 50 | 7 | 65 | 300 |
| Number of classes | 7 | 6 | 3 | 7 | 7 | 4 | 2 | 67 | 40 | 9 | 2 | 11 | 3 | 2 | 2 | 160 |
| Number of incidence pairs ($t$) | 4786 | 3453 | 34629 | 4585 | 99561 | 65451 | 40620 | 10060 | 61555 | 4523594 | 11843 | 460630 | 53372 | 16356 | 5502 | 726861 |
| Avg edge size ($\bar{e}_{avg}$) | 3.03 | 3.20 | 4.35 | 4.28 | 4.45 | 654.51 | 136.31 | 5.00 | 5.00 | 6.66 | 34.73 | 6.59 | 5.25 | 6.23 | 2.29 | 3.12 |
| Density ($\bar{e}_{avg}/|V|$) | 0.00112 | 0.00097 | 0.00022 | 0.00158 | 0.00011 | 0.04030 | 0.01678 | 0.00249 | 0.00041 | 0.00013 | 0.02692 | 0.00007 | 0.00032 | 0.00037 | 0.00015 | 0.00002 |
| Edge homophily ($\mathcal{H}_e$) | 0.75 | 0.68 | 0.78 | 0.78 | 0.87 | 0.61 | 0.94 | 0.79 | 0.87 | 0.29 | 0.49 | 0.60 | 0.46 | 0.49 | 0.45 | 0.98 |
| Node homophily ($\mathcal{H}_n$) | 0.76 | 0.68 | 0.76 | 0.76 | 0.86 | 0.53 | 0.91 | 0.77 | 0.85 | 0.24 | 0.51 | 0.50 | 0.48 | 0.50 | 0.45 | 0.98 |

Table 2: Accuracy ($\pm$ std) across different datasets for each algorithm at sparsity = 50%. OOM=Out-of-Memory. The House column reports the corrected label-independent evaluation described in Appendix H.2.

| Algorithms | 20news | ModelNet40 | Mushroom | NTU2012 | Actor | Citeseer | Cora-CA | DBLP-CA | |
|---|---|---|---|---|---|---|---|---|---|
| Full | $81.04 \pm 0.05$ | $94.72 \pm 0.05$ | $98.73 \pm 0.37$ | $88.03 \pm 0.26$ | $64.54 \pm 0.03$ | $68.19 \pm 0.39$ | $81.48 \pm 0.36$ | $90.77 \pm 0.07$ | |
| Random | $76.85 \pm 0.30$ | $95.91 \pm 0.33$ | $97.60 \pm 0.58$ | $86.92 \pm 0.36$ | $77.85 \pm 0.26$ | $67.32 \pm 1.27$ | $75.92 \pm 0.53$ | $86.26 \pm 0.32$ | |
| Edgedeg | $80.14 \pm 0.22$ | $95.18 \pm 0.23$ | $98.02 \pm 0.40$ | $86.32 \pm 0.78$ | $76.59 \pm 0.18$ | $67.27 \pm 0.31$ | $80.09 \pm 0.70$ | $88.76 \pm 0.27$ | |
| Spectral | $80.19 \pm 0.07$ | $94.80 \pm 0.07$ | $98.06 \pm 0.13$ | $85.88 \pm 0.31$ | $75.22 \pm 0.03$ | $67.05 \pm 0.59$ | $79.03 \pm 0.21$ | OOM | |
| HSL | $81.68 \pm 0.09$ | $79.93 \pm 2.99$ | $99.75 \pm 0.00$ | $80.58 \pm 0.41$ | $63.85 \pm 0.76$ | $63.37 \pm 1.06$ | $74.84 \pm 2.10$ | $90.31 \pm 0.39$ | |
| HSL w/ aug. | $81.74 \pm 0.37$ | $97.65 \pm 0.16$ | $100.00 \pm 0.00$ | $87.41 \pm 1.00$ | $63.74 \pm 0.69$ | $69.28 \pm 1.03$ | $81.34 \pm 0.74$ | $90.73 \pm 0.08$ | |
| EHGNN-F | $77.18 \pm 0.10$ | $96.32 \pm 0.12$ | $97.24 \pm 0.21$ | $87.48 \pm 0.47$ | $78.01 \pm 0.11$ | $66.76 \pm 0.45$ | $77.49 \pm 0.49$ | $87.01 \pm 0.14$ | |
| EHGNN-C | $79.08 \pm 0.22$ | $95.73 \pm 0.06$ | $93.59 \pm 0.50$ | $87.20 \pm 0.48$ | $76.72 \pm 0.10$ | $68.60 \pm 0.59$ | $77.02 \pm 0.32$ | $85.28 \pm 0.07$ | |
| EHGNN-F (cond) | $80.16 \pm 0.19$ | $95.76 \pm 0.10$ | $98.28 \pm 0.30$ | $88.11 \pm 0.09$ | $77.18 \pm 0.33$ | $67.71 \pm 0.52$ | $76.48 \pm 0.46$ | $86.45 \pm 0.10$ | |
| EHGNN-C (cond) | $78.11 \pm 0.98$ | $95.01 \pm 0.08$ | $97.97 \pm 0.33$ | $87.12 \pm 0.60$ | $77.01 \pm 0.33$ | $68.50 \pm 0.61$ | $76.16 \pm 0.56$ | $86.29 \pm 0.05$ | |
| EHGNN-F (cond,LR) | $76.94 \pm 0.39$ | $95.79 \pm 0.12$ | $98.23 \pm 0.39$ | $86.00 \pm 0.78$ | $77.98 \pm 0.22$ | $67.68 \pm 0.41$ | $77.02 \pm 0.58$ | $87.43 \pm 0.13$ | |
| EHGNN-C (cond,LR) | $79.51 \pm 0.55$ | $95.47 \pm 0.17$ | $96.57 \pm 0.30$ | $88.11 \pm 0.36$ | $76.73 \pm 0.42$ | $67.97 \pm 0.44$ | $77.73 \pm 0.63$ | $86.73 \pm 0.55$ | |
| | Cora | House | Pokec | PubMed | Twitch | Walmart | Yelp | Trivago | Avg. Rank |
| Full | $78.23 \pm 0.31$ | $47.31 \pm 0.14$ | $58.36 \pm 0.04$ | $86.50 \pm 0.05$ | $51.22 \pm 0.05$ | $95.36 \pm 0.01$ | $33.60 \pm 0.48$ | $61.39 \pm 1.15$ | |
| Random | $73.65 \pm 1.31$ | $50.03 \pm 1.21$ | $59.00 \pm 0.26$ | $86.72 \pm 0.24$ | $50.92 \pm 0.46$ | $96.97 \pm 0.16$ | $32.77 \pm 0.28$ | $47.14 \pm 1.52$ | 6.16 |
| Edgedeg | $74.62 \pm 1.42$ | $50.28 \pm 1.32$ | $58.71 \pm 0.14$ | $86.72 \pm 0.09$ | $50.86 \pm 0.38$ | $95.40 \pm 0.05$ | $32.36 \pm 0.57$ | $53.59 \pm 0.72$ | 5.97 |
| Spectral | $75.92 \pm 0.36$ | $48.73 \pm 0.71$ | $58.47 \pm 0.03$ | $86.47 \pm 0.01$ | $51.39 \pm 0.05$ | OOM | OOM | OOM | 6.92 |
| HSL | $70.56 \pm 0.37$ | $49.16 \pm 2.42$ | $57.81 \pm 0.71$ | $77.51 \pm 0.80$ | $51.15 \pm 0.99$ | $62.57 \pm 0.16$ | OOM | $68.17 \pm 2.16$ | 7.80 |
| HSL w/ aug. | $78.93 \pm 0.37$ | $49.54 \pm 4.15$ | $59.55 \pm 0.26$ | $87.43 \pm 0.17$ | $51.34 \pm 0.10$ | $61.71 \pm 0.39$ | OOM | $63.65 \pm 5.04$ | 3.07 |
| EHGNN-F | $75.72 \pm 0.22$ | $49.35 \pm 0.84$ | $58.53 \pm 0.03$ | $86.75 \pm 0.06$ | $50.84 \pm 0.04$ | $96.90 \pm 0.02$ | $33.36 \pm 0.26$ | $45.56 \pm 1.56$ | 5.38 |
| EHGNN-C | $75.60 \pm 0.28$ | $51.46 \pm 0.26$ | $59.15 \pm 0.05$ | $86.56 \pm 0.07$ | $51.44 \pm 0.08$ | $95.89 \pm 0.08$ | $31.98 \pm 0.23$ | $37.78 \pm 1.42$ | 5.72 |
| EHGNN-F (cond) | $74.71 \pm 0.44$ | $48.98 \pm 1.01$ | $59.14 \pm 0.08$ | $86.10 \pm 0.07$ | $50.58 \pm 0.08$ | $96.39 \pm 1.41$ | OOM | $40.08 \pm 0.29$ | 6.03 |
| EHGNN-C (cond) | $75.07 \pm 0.48$ | $50.59 \pm 0.47$ | $58.90 \pm 0.31$ | $86.43 \pm 0.13$ | $50.32 \pm 0.23$ | $95.20 \pm 0.35$ | OOM | $46.03 \pm 2.26$ | 7.00 |
| EHGNN-F (cond,LR) | $74.33 \pm 0.65$ | $52.51 \pm 0.56$ | $58.77 \pm 0.18$ | $86.97 \pm 0.04$ | $50.72 \pm 0.10$ | $96.65 \pm 0.22$ | $32.78 \pm 0.37$ | $51.45 \pm 1.14$ | 5.22 |
| EHGNN-C (cond,LR) | $76.13 \pm 0.60$ | $50.71 \pm 0.89$ | $58.93 \pm 0.06$ | $86.52 \pm 0.05$ | $51.26 \pm 0.20$ | $95.47 \pm 0.53$ | $33.03 \pm 0.21$ | $40.28 \pm 0.89$ | 5.09 |

are well-known in the hypergraph learning literature, originating from works such as Yadati et al. (2019); Chien et al. (2021) where they are discussed in detail.

**Baselines.** The baseline algorithms include i) *Full*: the HGNN model is trained on the entire hypergraph without any sparsification, ii) *Edgedeg*: $\kappa$ edges are sampled proportional to edge degree $deg(e) = |e|$. iii) *Random*: independently retains each incidence pair $(u, e)$ with probability equal to the target retention ratio $\rho$ (e.g., $\rho = 0.5$ for 50% retention), and iv) *Spectral*: We sample top-$\kappa$ hyperedges based on their effective resistance. We compute effective resistance as $R(e) := \sum_{v \in e} \left( L^{\dagger} \right)_{vv}$, where $L^{\dagger}$ indicates the Moore-Penrose pseudoinverse of the hypergraph Laplacian (Zhou et al., 2006). We additionally compare to (v) Hypergraph Structure Learning, HSL (Cai et al., 2022). To the best of our knowledge, there is no prior supervised hypergraph sparsifier that performs only pruning under an explicit global budget, and HSL is the closest task-aware hypergraph structure-learning baseline. The original HSL method combines pruning with connectivity augmentation/discovery, which changes the feasible set by adding incidences not present in the observed hypergraph. We therefore report both a pruning-oriented, budget-matched HSL-style comparator and HSL w/ aug., which includes the original residual-incidence augmentation step. Discussion on computational complexity, effectiveness on heterophilic hypergraphs, experiments with other HGNN backbones, scalability, model parameters, link prediction, low-label settings, House preprocessing, ablation studies, and empirical evaluation of mask stability can be found in the Appendix. Our source codes: `https://anonymous.4open.science/r/ehgnn-4E4D/`.

## 6.1 Experimental evaluation

**I. Effectiveness on node classification task.** Table 2 highlights the effectiveness of EdgeMask-HGNN.

*Supervised vs Unsupervised sparsifiers:* On most datasets, EHGNN variants consistently outperform un-supervised sparsification baselines. This highlights the importance of task-aware learning of sparsification masks, which can retain task-relevant incidences or hyperedges while discarding noisy ones. On large-scale datasets, spectral methods face memory issues due to the computation of the large matrix pseudoinverse. We evaluate HSL in two ways. The pruning-oriented HSL row gives a budget-matched adaptation over observed incidences, because EdgeMask-HGNN and the other sparsifiers select subhypergraphs from the given hypergraph. The HSL w/ aug. row includes the original residual-incidence augmentation step and therefore reflects a broader structure-learning setting that can add new incidences. The measured per-dataset ratios are 0.53 (20news), 0.43 (ModelNet40), 0.49 (Mushroom), 0.48 (NTU2012), 0.54 (Actor), 0.46 (Citeseer), 0.46 (Cora-CA), 0.58 (DBLP-CA), 0.45 (Cora), 0.36 (House), 0.57 (Pokec), 0.58 (PubMed), 0.52 (Twitch), 0.54 (Walmart), and 0.50 (Trivago); Yelp runs out of memory. Since these values are close to the nominal 50% budget, the HSL w/ aug. accuracy gains are not caused by systematically retaining a much denser hyperedge set.

*Full training vs Sparsification:* EdgeMask-HGNNs outperform full training on several benchmarks, such as Actor, ModelNet40, Pokec, PubMed, Twitch, and Walmart. This supports a conditional view: pruning can help when the removed incidences or hyperedges are noisy, redundant, or weakly related to the downstream task. On datasets such as 20news, Cora, Cora-CA, DBLP-CA, Mushroom, Yelp, and Trivago, Full training performs better or remains competitive, indicating that pruning can also introduce harmful approximation bias when many relations are useful. Note that, Appendix G discusses train–test gap analysis and controlled-noise experiments.

**Default variant and budget selection.** Unless otherwise stated, we use a 50% retention budget as a controlled operating point for comparing sparsifiers under the same compression level; it is not intended to be universally optimal. In practice, we recommend selecting the retention budget by validation accuracy over a small grid such as $\rho \in \{0.3, 0.5, 0.7, 0.9\}$. This is a validation-based model-selection procedure, not an automatic learned budget controller. A natural adaptive alternative would replace the hard constraint with a Lagrangian $L_0$-style penalty, e.g., optimizing $\mathcal{L}_{task} + \lambda \|\boldsymbol{M}\|_0$ or $\mathcal{L}_{task} + \lambda \|\boldsymbol{m}\|_0$ using differentiable hard-concrete style gates (Louizos et al., 2018); we leave such joint budget learning to future work. The Appendix retention sweep shows that homophilic datasets often prefer larger budgets. For variant choice, EHGNN-F is the default when incidence-level control is feasible, EHGNN-C is a lower-overhead alternative

Table 3: Practical starting points for EdgeMask-HGNN variant selection by incidence scale. The incidence count $t$ guides computational feasibility, while final model selection is validation-based.

| Incidence scale | Starting variant(s) | Rationale | Examples / caveat |
|---|---|---|---|
| Small $t < 20k$ | Full HGNN; validate EHGNN-C/F | Full training is usually inexpensive; sparsification is optional unless validation indicates useful pruning. | Cora, Cora-CA, Pokec, Twitch |
| Medium 20k–200k | EHGNN-C and EHGNN-F | Both variants are feasible; EHGNN-C has lower overhead, while EHGNN-F gives finer incidence-level control. | PubMed, Actor, ModelNet40, DBLP-CA |
| Large 200k–800k | EHGNN-F; EHGNN-F(cond,LR) | Incidence pruning can reduce activation cost; low-rank scoring controls feature-conditioned scorer overhead. | Walmart; Trivago is a high-homophily caveat |
| Very large $t > 800k$ | EHGNN-F or low-rank conditioned variants | Full MLP-conditioned scorers may become memory-limited; low-rank variants are safer when conditioning is needed. | Yelp |

Table 4: Node-classification cross-dataset correlation between accuracy gaps and homophily. For each dataset in Table 2, the gap is the mean test accuracy of the EdgeMask-HGNN variant minus the mean test accuracy of Full training at 50% retention; correlations are then computed across datasets using the homophily values in Table 1.

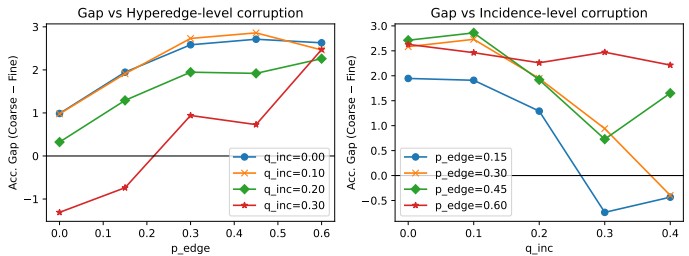

Figure 3: Classification performance on corrupted Cora.

| Metric | EHGNN-C | | EHGNN-F | |
|---|---|---|---|---|
| | Pearson | Spearman | Pearson | Spearman |
| $\mathcal{H}_e$ | -0.604 | **-0.607** | **-0.543** | **-0.519** |
| $\mathcal{H}_n$ | **-0.579** | **-0.617** | **-0.515** | **-0.531** |

for large incidence sets, and low-rank conditioned variants are useful when feature-conditioned scoring is desired but full MLP scoring is too memory intensive. All reported variant choices should be made by validation performance rather than post-hoc test-set selection.

**Practical variant-selection guidelines.** The incidence count $t$ is useful for choosing computationally feasible candidate variants, but it is not by itself a predictor of the best-performing method. Homophily, noise, and task structure can override this scale-based guideline, so Table 3 uses $t$ only to identify practical starting points; the final variant and retention budget should still be selected by validation performance.

*On heterophilic datasets:* We also analyze the node-classification performance gain of EdgeMask-HGNN relative to full training as a function of homophily. For each dataset in Table 2, we compute $\Delta_C = \text{Acc}(\text{EHGNN-C}) - \text{Acc}(\text{Full})$ and $\Delta_F = \text{Acc}(\text{EHGNN-F}) - \text{Acc}(\text{Full})$, using the mean test accuracies at 50% retention, and correlate these gaps with the edge and node homophily values in Table 1. Table 4 suggests that improvements tend to be larger in lower-homophily regimes, but the pattern is not universal and some gains are small. For instance, the Spearman correlation coefficient between $\Delta_C$ and edge homophily is $-0.61$, and that between $\Delta_F$ and edge homophily is $-0.52$. Figure 7 (Appendix) illustrates this trend.

*Coarse-grained vs Fine-grained Sparsification:* Although on the real-world datasets, generally we found fine-grained sparsification to perform better than coarse-grained sparsification, it is important to analyze under what conditions coarse-grained sparsification is more preferable than fine-grained or vice versa. To that end, we corrupted the hyperedges in Cora in two distinct yet controlled ways. Each hyperedge is assigned a target adversarial class based on the majority labels in that edge, and corrupted incidences are replaced by nodes

drawn from this class. (i) *Coarse/hyperedge-level corruption:* For each edge, with probability $p_{\text{edge}}$, all its nodes are replaced by nodes from its adversarial class. (ii) *fine/incidence-level corruption*: For each edge, with probability $1 - p_{\text{edge}}$, $q_{\text{inc}}$ proportion of its nodes are replaced by nodes from its adversarial class.

Figure 3 reveals a clear dependence of model performance on the granularity of corruption. (i) When $q_{\text{inc}}$ is small ($< 0.3$), increasing $p_{\text{edge}}$ leads to a monotonic increase in the performance gap in favor of coarse-grained sparsification, as entire hyperedges become uninformative and are best removed in their entirety. (ii) Conversely, when $p_{\text{edge}}$ is small, increasing $q_{\text{inc}}$ causes the gap to decrease and eventually become negative or close to 0. This suggests that fine-grained masking, which selectively removes corrupted incidences, is more beneficial when incidence-level corruption is more predominant and edge-level corruption is small. However, when $p_{\text{edge}}$ is large, many hyperedges are already fully corrupted, and coarse masking remains advantageous regardless of $q_{\text{inc}}$.

**II. Effectiveness on link prediction task.**  We also evaluate the applicability of EdgeMask-HGNN to link prediction by adapting it to **Hyper-SAGNN** (Zhang et al., 2020), a self-attention based architecture for hyperedge prediction. In the inductive setting, the model is trained on $\mathcal{E}_{train} \subset \mathcal{E}$ and evaluated on unseen hyperedges from $\mathcal{E}_{val}$ and $\mathcal{E}_{test}$. Training examples consist of positive edges $e \in \mathcal{E}_{train}$ and negative examples generated by corrupting one position of a positive hyperedge with another node of the same type.

Hyper-SAGNN computes node embeddings $\boldsymbol{h}_v^{(0)} = \text{Embedding}(v)$ and applies self-attention within each hyperedge to yield an edge representation $\boldsymbol{z}_e$, which is later used for link prediction. However, Hyper-SAGNN processes each hyperedge independently without exploiting global hypergraph structure. In order to enable global hypergraph-structure aware link prediction, we introduce a learnable sparsification module `EdgeMask-HGNN`$_{\Phi}()$ with parameters $\Phi$, which learns a binary selection variable $m_e$ that determines whether an incident hyperedge contributes to the contextualized node representations. During training, we use a straight-through estimator to enable gradient-based optimization of this discrete selection. We construct contextualized node embeddings as follows:

$$\boldsymbol{h}_v^{(1)} = \boldsymbol{h}_v^{(0)} + \sum_{e:v \in e} \boldsymbol{m}_e \boldsymbol{g}_e,$$

where $\boldsymbol{g}_e = \frac{1}{|e|} \sum_{u \in e} \boldsymbol{W} \boldsymbol{h}_u^{(0)}$ is an edge message and $\boldsymbol{W}$ is a learnable parameter. The updated embeddings $\boldsymbol{h}_v^{(1)}$ are then fed into Hyper-SAGNN so that the predictions for a candidate hyperedge can incorporate information from other hyperedges sharing nodes. The contextualized representations enabled by sparsifiers help control the impact of noise on message passing. A detailed discussion is presented in Appendix E.

Experiments using variants of EdgeMask-HGNN, as presented in Table 5, show large improvements in accuracy, AUC, and AUPR at 50% sparsification compared with the plain Hyper-SAGNN baseline. However, the adaptation also introduces contextual aggregation before Hyper-SAGNN. To separate the benefit of

Table 5: Inductive hyperedge link-prediction performance for Hyper-SAGNN and EdgeMask-HGNN contextual sparse aggregation variants at 50% retention. Results are mean $\pm$ standard deviation over five seeds. Metrics are accuracy, AUC, and AUPR under the same train/validation/test protocol used by Hyper-SAGNN. Table 12 in Appendix E reports the contextual aggregation ablation.

| Model | WordNet (user, relation, tail) $|V| = (40504, 18, 40551)$ | | | Drug (user, drug, reaction) $|V| = (12, 1076, 6398)$ | | | MovieLens (user, movie, tag) $|V| = (2113, 5908, 9079)$ | | |
|---|---|---|---|---|---|---|---|---|---|
| | **Acc** | **AUC** | **AUPR** | **Acc** | **AUC** | **AUPR** | **Acc** | **AUC** | **AUPR** |
| Hyper-SAGNN | $88.26 \pm 0.12$ | $87.72 \pm 0.42$ | $68.34 \pm 0.21$ | $93.97 \pm 0.02$ | $95.64 \pm 0.04$ | $89.02 \pm 0.05$ | $90.77 \pm 0.06$ | $91.00 \pm 0.05$ | $76.04 \pm 0.43$ |
| Random | $93.32 \pm 0.15$ | $95.94 \pm 0.15$ | $87.61 \pm 0.30$ | $96.49 \pm 0.02$ | $98.66 \pm 0.04$ | $95.72 \pm 0.06$ | $95.02 \pm 0.05$ | $97.27 \pm 0.06$ | $91.91 \pm 0.23$ |
| EdgeDeg | $93.17 \pm 0.28$ | $95.70 \pm 0.28$ | $87.05 \pm 0.88$ | $96.50 \pm 0.07$ | $98.65 \pm 0.05$ | $95.67 \pm 0.15$ | $94.97 \pm 0.03$ | $97.29 \pm 0.03$ | $91.72 \pm 0.09$ |
| Spectral | $91.16 \pm 0.17$ | $92.00 \pm 0.54$ | $79.14 \pm 1.01$ | $96.95 \pm 0.05$ | $98.71 \pm 0.04$ | $96.07 \pm 0.02$ | $93.73 \pm 0.16$ | $95.85 \pm 0.09$ | $87.97 \pm 0.29$ |
| EHGNN-F | $96.35 \pm 0.17$ | $96.86 \pm 0.50$ | $92.41 \pm 0.37$ | $96.96 \pm 0.07$ | $97.49 \pm 0.33$ | $94.96 \pm 0.13$ | $96.40 \pm 0.02$ | $97.13 \pm 0.36$ | $93.28 \pm 0.38$ |
| EHGNN-C | $93.14 \pm 0.34$ | $95.71 \pm 0.44$ | $87.02 \pm 1.12$ | $96.51 \pm 0.03$ | $98.65 \pm 0.04$ | $95.70 \pm 0.10$ | $94.97 \pm 0.03$ | $97.18 \pm 0.02$ | $91.55 \pm 0.02$ |
| EHGNN-F(cond) | $94.89 \pm 0.07$ | $97.07 \pm 0.27$ | $91.36 \pm 0.30$ | $97.25 \pm 0.19$ | $99.14 \pm 0.16$ | $97.01 \pm 0.29$ | $96.35 \pm 0.30$ | $98.35 \pm 0.29$ | $94.55 \pm 0.88$ |
| EHGNN-C(cond) | $93.79 \pm 0.31$ | $96.43 \pm 0.46$ | $88.85 \pm 1.10$ | $96.79 \pm 0.18$ | $98.93 \pm 0.17$ | $96.22 \pm 0.30$ | $95.65 \pm 0.14$ | $97.87 \pm 0.12$ | $93.29 \pm 0.27$ |
| EHGNN-F(cond,LR) | $\mathbf{98.74 \pm 0.08}$ | $\mathbf{99.68 \pm 0.04}$ | $\mathbf{99.01 \pm 0.13}$ | $\mathbf{98.25 \pm 0.05}$ | $\mathbf{99.67 \pm 0.00}$ | $\mathbf{98.72 \pm 0.01}$ | $\mathbf{98.38 \pm 0.05}$ | $\mathbf{99.66 \pm 0.01}$ | $\mathbf{98.57 \pm 0.08}$ |
| EHGNN-C(cond,LR) | $91.73 \pm 0.14$ | $92.43 \pm 0.82$ | $81.28 \pm 0.74$ | $95.76 \pm 0.03$ | $97.68 \pm 0.04$ | $93.68 \pm 0.09$ | $93.83 \pm 0.07$ | $95.66 \pm 0.10$ | $88.23 \pm 0.14$ |

contextualization from the benefit of sparsification, Appendix E reports ablations comparing plain Hyper-SAGNN, full contextual aggregation with no sparsification, random masked contextual aggregation, and learned masked contextual aggregation. These ablations show a nuanced picture: full-context aggregation is a strong no-sparsification control on WordNet/Drug/MovieLens, while learned masks are consistently stronger than random masked context and can outperform full context on dense GPS settings with a larger context budget. Thus, the link-prediction results support EdgeMask-HGNN as a learned budgeted contextualization mechanism, not a uniformly superior replacement for full-context aggregation.

**III. Accuracy/Runtime vs sparsity trade-off.** We analyze the trade-off between accuracy and sparsity in Figure 4a, and the impact of sparsity on end-to-end runtime (Training + Evaluation) in Figure 4b. On the x-axis, we plot % incidences retained, which is 100%-sparsity. As more incidences are retained, accuracy increases at the cost of higher runtime. Conversely, the more we sparsify, the lower the accuracy and runtime. This phenomenon mirrors that observed in graph sparsification (Das et al., 2025).

**IV. Runtime efficiency.** Figure 4c compares the average training time per epoch across several large-scale datasets. On Yelp, Walmart, and DBLP-CA, the Full HGNN consistently incurs the highest, while both EHGNN-F and EHGNN-C achieve substantially lower runtimes. This aligns with the fact that incidence-level sparsification reduces the message-passing workload, which dominates the cost on large hypergraphs.

**V. Ablation study: Impact of sparsity on homophilic hypergraphs.** In table 2, we observe that on datasets 20news, Cora-CA, DBLP-CA, Mushroom, and Trivago, Full training outperforms EdgeMask-HGNN. These hypergraphs have edge homophily ratio 0.53, 0.76, 0.86, 0.91, and 0.98, respectively. Here, we investigate whether increasing the budget (% incidences retained) improves the performance of EHGNN-F on these highly homophilic datasets.

Figure 5 shows the result with varying % of incidence retained by EHGNN-F. We observe a near-monotonic increase in accuracy as more incidences are retained. This suggests that highly homophilic hypergraphs have a high sensitivity to sparsification. For instance, even at a high retention level (0.9), EHGNN-F does not fully match the performance of Full training (red dashed lines), especially on datasets such as Trivago, where the performance gap remains significant. This indicates that, in strongly homophilic regimes, sparsification may discard weak but collectively important connections that contribute to label propagation.

This is a key limitation of sparsification in homophilic settings. Although moderate pruning yields competitive performance, a large fraction of incidences need to be retained to yield performance equivalent to full-training. This contrasts with heterophilic scenarios, where sparsification can be more beneficial by removing noisy or misleading connections.

## 7 Conclusion, Limitations and Future works

We introduced EdgeMask-HGNN, a task-aware sparsification framework that prunes task-irrelevant relations and often preserves or improves the predictive performance of HGNNs on downstream tasks. We proposed two learnable sparsification strategies: fine-grained masking and coarse-grained masking– both trained end-

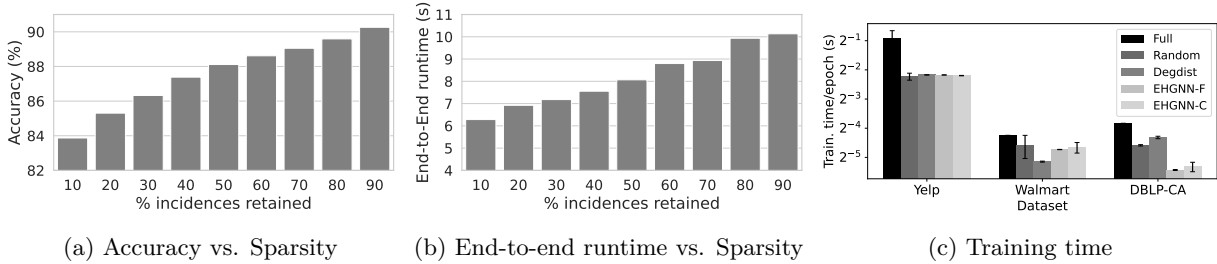

(a) Accuracy vs. Sparsity     (b) End-to-end runtime vs. Sparsity     (c) Training time

Figure 4: (a-b) Impact of sparsity on EHGNN-F's performance (DBLP-CA dataset) (c) Training time comparison of various sparsifiers.

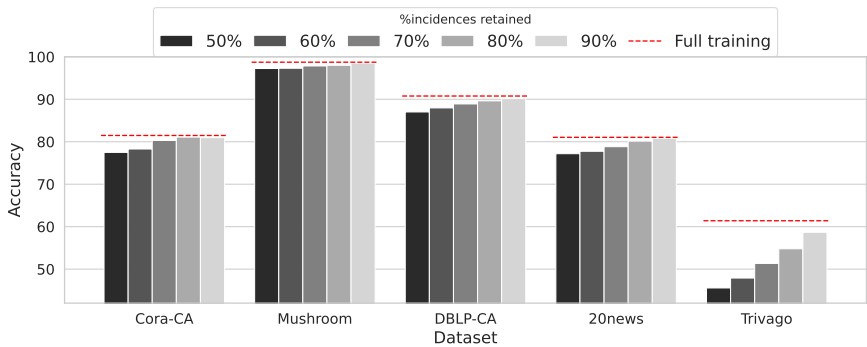

Figure 5: Impact of %retained incidences on EHGNN-F for homophilic hypergraphs.

to-end using feedback from downstream tasks. Furthermore, EdgeMask-HGNN is theoretically grounded in terms of the mask stability of the learned sparsifiers.

Extensive experiments across diverse and challenging node classification benchmarks suggest EdgeMask-HGNN is generally effective, while most effective on heterophilic hypergraphs. In particular, the fine-grained variants not only improve accuracy over full hypergraph training in many cases, but also achieve slightly smaller training time on large hypergraphs. Finally, EdgeMask-HGNN is adaptable to other downstream tasks such as link prediction. Our revised ablations show that full-context aggregation is a strong no-sparsification control, while learned masks outperform random masked context and can be preferable in dense settings where context selection is useful.

*Limitations.* EdgeMask-HGNN learns its sparsification mask from a downstream supervised objective. Our limited-label experiments show that the method can remain useful with very few labels, but under severe label scarcity it does not always outperform full hypergraph training on every dataset. Fully unsupervised settings such as node clustering would require replacing the supervised task loss with a proxy objective, such as incidence reconstruction or contrastive learning; we leave this extension to future work. EdgeMask-HGNN also requires a retention-budget hyperparameter; learning this budget jointly, for example through a Lagrangian $L_0$-regularized objective, is another direction for future work. The additional computational overhead introduced by feature-conditioned variants also causes scalability challenges on large-scale hypergraphs. In the future, we plan to address these limitations.

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

## Appendix

## A   Theoretical Analysis

We analyze the stability of the stochastic masks produced by EHGNN-F and EHGNN-C during training. In particular, we study how changes in the logit parameters affect the sampled incidence sets (for EHGNN-F) and sampled edges (for EHGNN-C) between consecutive epochs.

### A.1   Stability of Fine-grained Stochastic Masks

We analyze the stochastic fine mask used in EHGNN-F during training, where $k$ incidences are sampled without replacement from $t$ candidate incidences using probabilities proportional to weights $w_{v,e} := \sigma(s_{v,e})$.

Let $t = \sum_{e \in E} |e|$ denote the number of incidence pairs and $k$ denotes the number of retained incidence pairs after sparsification. At iteration $\tau$, let the logits $\boldsymbol{s}^{(\tau)} \in \mathbb{R}^t$ and the weights

$$w^{(\tau)}(i) := \sigma(s^{(\tau)}(i)) \in (0,1), \quad i \in [t].$$

For a fixed $k \in \{1, \ldots, t\}$, sampling $k$ incidence pairs without replacement with probabilities proportional to weights is equivalent to:

1. Sampling a permutation $\Pi^{(\tau)} \sim \mathrm{PL}(\boldsymbol{w}^{(\tau)})$ under the Plackett–Luce (PL) distribution (Luce et al., 1959).

2. Then selecting the first k indices

$$S^{(\tau)} := \{\Pi_1^{(\tau)}, \ldots, \Pi_k^{(\tau)}\} \subseteq [t], \quad |S^{(\tau)}| = k.$$

The selected set $S^{(\tau)}$ uniquely determines the incidence mask produced by EHGNN-F. We measure stability of fine-grained stochastic masks via the symmetric difference:

$$|S^{(\tau+1)} \triangle S^{(\tau)}|.$$

This equals the number of incidences whose inclusion state changes between epochs.

**Assumption A.1** (Uniformly lower-bounded weights)**.** There exists $\alpha > 0$ such that for all $\tau$ and all $i \in [t]$,

$$w^{(\tau)}(i) \geq \alpha.$$

This assumption holds, for instance, if logits are uniformly bounded. In particular, since $\sigma(x) > 0$, a uniform lower bound $s^{(\tau)}(i) \geq -B$ implies the following uniform weight lower bound

$$w^{(\tau)}(i) \geq \sigma(-B) := \alpha.$$

First, we formalize how perturbations in the weights $w^{(\tau)}$ at epoch $\tau$ affect the distribution of the first $k$ positions of the Plackett–Luce (PL) permutation.

**Lemma A.2** (Single-step conditional stability of PL weights)**.** *Let $R \subseteq [t]$ with $|R| = r$. Let us define categorical distributions on $R$:*

$$u(i) = \frac{w(i)}{\sum_{j \in R} w(j)}, \quad v(i) = \frac{w'(i)}{\sum_{j \in R} w'(j)}.$$

*If $w(i), w'(i) \geq \alpha$ for all $i$, then the total variation distance*

$$\mathrm{TV}(u,v) \leq \frac{1}{r\alpha} \|\boldsymbol{w} - \boldsymbol{w}'\|_1.$$

*Proof.* Let
$$P_R := \sum_{j \in R} w(j), \quad Q_R := \sum_{j \in R} w'(j).$$

Since $w(j) \geq \alpha$, we have
$$P_R \geq r\alpha, \quad Q_R \geq r\alpha.$$

By definition of TV,
$$\mathrm{TV}(u, v) = \frac{1}{2} \sum_{i \in R} \left| \frac{w(i)}{P_R} - \frac{w'(i)}{Q_R} \right|.$$

Insert and subtract $\frac{w(i)}{Q_R}$:
$$\left| \frac{w(i)}{P_R} - \frac{w'(i)}{Q_R} \right| \leq \left| \frac{w(i)}{P_R} - \frac{w(i)}{Q_R} \right| + \left| \frac{w(i) - w'(i)}{Q_R} \right|.$$

Summing over $i \in R$ yields
$$\sum_{i \in R} \left| \frac{w(i)}{P_R} - \frac{w'(i)}{Q_R} \right| \leq \frac{|Q_R - P_R|}{Q_R} + \frac{1}{Q_R} \sum_{i \in R} |w(i) - w'(i)|.$$

Since
$$|Q_R - P_R| = \left| \sum_{j \in R} (w'(j) - w(j)) \right| \leq \|\boldsymbol{w} - \boldsymbol{w}'\|_1,$$

and $Q_R \geq r\alpha$, we obtain
$$\mathrm{TV}(u, v) \leq \frac{1}{r\alpha} \|\boldsymbol{w} - \boldsymbol{w}'\|_1.$$

$\square$

**Lemma A.3** (Total variation bound for the first $k$ positions of the Plackett–Luce permutation)**.** *Let* $\boldsymbol{w}, \boldsymbol{w}' \in \mathbb{R}^t$ *satisfy* $w(i), w'(i) \geq \alpha > 0 \quad \forall i \in [t]$. *Let* $\Pi^{(w)}$ *and* $\Pi^{(w')}$ *be permutations sampled from the Plackett–Luce distributions with base weights $w$ and $w'$ respectively. Let us denote the first $k$ positions by* $\Pi^{(w)}_{1:k}$, $\Pi^{(w')}_{1:k}$. *Then*
$$\mathrm{TV}\big(\Pi^{(w)}_{1:k}, \Pi^{(w')}_{1:k}\big) \leq \frac{1}{\alpha} \ln \frac{t}{t - k} \|\boldsymbol{w} - \boldsymbol{w}'\|_1.$$

*Proof.* The PL process samples sequentially.

At step $r$, given previously selected elements $\Pi_{1:r-1}$, the next element is drawn from the remaining set $R_r$ of size $|R_r| = t - r + 1$. At step $r$ of the Plackett–Luce process, let us denote
$$\mathrm{TV}_r := \mathrm{TV}\big(u_{R_r}, v_{R_r}\big),$$

where $u_{R_r}$ and $v_{R_r}$ are the conditional categorical distributions restricted to $R_r$.

By Lemma A.2, for any realization of the remaining set $R_r$
$$\mathrm{TV}_r \leq \frac{1}{(t - r + 1)\alpha} \|\boldsymbol{w} - \boldsymbol{w}'\|_1.$$

We now construct a maximal coupling of the two PL processes, coupling the draws at each step.

Let $E_r$ be the event that the two processes differ for the first time at step $r$. Using maximal coupling at each step of the sequential PL sampling process, the probability that the two permutations first disagree at step

$r$, conditioned on agreement up to step $r-1$, is at most the total variation distance between the conditional distributions at that step:

$$\mathbb{P}(E_r \mid \text{agreement up to } r-1) \leq \text{TV}_r,$$

Therefore, by a union bound over the first $k$ steps,

$$\mathbb{P}\big(\Pi_{1:k}^{(w)} \neq \Pi_{1:k}^{(w')}\big) \leq \sum_{r=1}^{k} \text{TV}_r = \sum_{r=1}^{k} \frac{1}{(t-r+1)\alpha} \|\boldsymbol{w} - \boldsymbol{w}'\|_1.$$

Since

$$\sum_{r=1}^{k} \frac{1}{t-r+1} = H_t - H_{t-k},$$

we obtain

$$\mathbb{P}\big(\Pi_{1:k}^{(w)} \neq \Pi_{1:k}^{(w')}\big) \leq \frac{H_t - H_{t-k}}{\alpha} \|\boldsymbol{w} - \boldsymbol{w}'\|_1.$$

Here $H_t := \sum_{i=1}^{t} \frac{1}{i}$. Since total variation distance equals the minimal disagreement probability under coupling,

$$\text{TV}\big(\Pi_{1:k}^{(w)}, \Pi_{1:k}^{(w')}\big) \leq \frac{H_t - H_{t-k}}{\alpha} \|\boldsymbol{w} - \boldsymbol{w}'\|_1.$$

As $H_t - H_{t-k} \leq \ln \frac{t}{t-k}$, we have

$$\text{TV}\big(\Pi_{1:k}^{(w)}, \Pi_{1:k}^{(w')}\big) \leq \frac{1}{\alpha} \ln \frac{t}{t-k} \|\boldsymbol{w} - \boldsymbol{w}'\|_1.$$

$$\square$$

We now state and prove a data processing property of total variation distance, which states that total variation distance cannot increase under deterministic mappings. We will apply this property of total variation distance to pass from permutation prefixes to subsets.

**Lemma A.4** (Data processing inequality for TV). *Let $X$ and $Y$ be random variables with distributions $\mu$ and $\nu$ respectively. Let $f$ be any deterministic measurable function. Then the distributions of $f(X)$ and $f(Y)$ satisfy*

$$\text{TV}\big(\mathcal{L}(f(X)), \mathcal{L}(f(Y))\big) \leq \text{TV}(\mu, \nu).$$

*Proof.* Let $\mu_f := \mathcal{L}(f(X))$ and $\nu_f := \mathcal{L}(f(Y))$ denote the distributions of the transformed random variables $f(X)$ and $f(Y)$ respectively. Recall that the total variation distance between two probability measures $\mu$ and $\nu$ is defined as

$$\text{TV}(\mu, \nu) = \sup_A |\mu(A) - \nu(A)|.$$

Let $B$ be any measurable subset of the codomain of $f$. Then the events $\{f(X) \in B\}$ and $\{X \in f^{-1}(B)\}$ are identical. Therefore

$$\mu_f(B) = \mathbb{P}(f(X) \in B) = \mu(f^{-1}(B)), \qquad \nu_f(B) = \mathbb{P}(f(Y) \in B) = \nu(f^{-1}(B)).$$

Hence

$$|\mu_f(B) - \nu_f(B)| = |\mu(f^{-1}(B)) - \nu(f^{-1}(B))|.$$

Since $f^{-1}(B)$ is a measurable subset of the domain,

$$|\mu(f^{-1}(B)) - \nu(f^{-1}(B))| \leq \sup_A |\mu(A) - \nu(A)| = \text{TV}(\mu, \nu).$$

Taking the supremum over all measurable sets $B$ yields

$$\text{TV}(\mu_f, \nu_f) = \sup_B |\mu_f(B) - \nu_f(B)| \leq \text{TV}(\mu, \nu).$$

$\square$

In our setting, the mapping $f$ sends the prefix permutation $\Pi_{1:k}$ to the selected subset $S$. Hence, the total variation distance between the distributions of $S$ is bounded by that of the prefixes:

$$\text{TV}(S(\boldsymbol{w}), S(\boldsymbol{w}')) \leq \text{TV}(\Pi_{1:k}^{(w)}, \Pi_{1:k}^{(w')}).$$

We now bound the expected number of incidence flips between two consecutive training iterations.

**Theorem 5.1** (Epoch-to-epoch stability of fine-grained masks). *Let $s^{(\tau)}, s^{(\tau+1)} \in \mathbb{R}^t$ denote the logit vectors at two consecutive epochs, and let*

$$\boldsymbol{w}^{(\tau)} := \sigma(\boldsymbol{s}^{(\tau)}), \qquad \boldsymbol{w}^{(\tau+1)} := \sigma(\boldsymbol{s}^{(\tau+1)}).$$

*Let $S^{(\tau)}$ and $S^{(\tau+1)}$ denote the $k$-subsets obtained by sampling from the Plackett–Luce distributions $\text{PL}(\boldsymbol{w}^{(\tau)})$ and $\text{PL}(\boldsymbol{w}^{(\tau+1)})$ respectively.*

*Under Assumption A.1, the expected number of incidence mask flips between the two consecutive epochs satisfies*

$$\mathbb{E}\left[|S^{(\tau+1)} \triangle S^{(\tau)}|\right] \leq \frac{k}{2\alpha} \ln \frac{t}{t-k} \|\boldsymbol{s}^{(\tau+1)} - \boldsymbol{s}^{(\tau)}\|_1.$$

*Proof.* Since $|S^{(\tau)}| = |S^{(\tau+1)}| = k$, we have

$$|S^{(\tau+1)} \triangle S^{(\tau)}| \leq 2k \cdot \mathbf{1}\{S^{(\tau+1)} \neq S^{(\tau)}\}.$$

Taking expectations gives

$$\mathbb{E}\left[|S^{(\tau+1)} \triangle S^{(\tau)}|\right] \leq 2k\,\mathbb{P}(S^{(\tau+1)} \neq S^{(\tau)}).$$

Under maximal coupling,

$$\mathbb{P}(S^{(\tau+1)} \neq S^{(\tau)}) = \text{TV}(S(\boldsymbol{w}^{(\tau+1)}), S(\boldsymbol{w}^{(\tau)})).$$

By Lemmas A.4 and A.3,

$$\text{TV}(S(\boldsymbol{w}^{(\tau+1)}), S(\boldsymbol{w}^{(\tau)})) \leq \text{TV}(\Pi_{1:k}^{(w^{(\tau+1)})}, \Pi_{1:k}^{(w^{(\tau)})}) \leq \frac{1}{\alpha} \ln \frac{t}{t-k} \|\boldsymbol{w}^{(\tau+1)} - \boldsymbol{w}^{(\tau)}\|_1.$$

Thus

$$\mathbb{E}\left[|S^{(\tau+1)} \triangle S^{(\tau)}|\right] \leq 2k \cdot \frac{1}{\alpha} \ln \frac{t}{t-k} \|\boldsymbol{w}^{(\tau+1)} - \boldsymbol{w}^{(\tau)}\|_1.$$

Since $\sigma'(x) = \sigma(x)(1 - \sigma(x)) \leq \frac{1}{4}$, the sigmoid map is 1/4-Lipschitz. Hence

$$\|\boldsymbol{w}^{(\tau+1)} - \boldsymbol{w}^{(\tau)}\|_1 \leq \frac{1}{4}\|\boldsymbol{s}^{(\tau+1)} - \boldsymbol{s}^{(\tau)}\|_1.$$

Substituting yields

$$\mathbb{E}\left[|S^{(\tau+1)} \triangle S^{(\tau)}|\right] \leq \frac{k}{2\alpha} \ln \frac{t}{t-k} \|\boldsymbol{s}^{(\tau+1)} - \boldsymbol{s}^{(\tau)}\|_1.$$

$\square$

Thus, the expected number of incidence flips between epochs is controlled by the change in the logit vector. We now show that an analogous stability guarantee holds for the coarse-grained mask used in EHGNN-C.

## A.2 Stability of Coarse-grained Stochastic Masks

**Theorem 5.2** (Epoch-to-epoch stability of coarse-grained masks)**.** *Let $m = |E|$ be the number of hyperedges and fix an edge budget $\kappa \in \{1, \ldots, m\}$. At iteration $\tau$, let edge logits $s_e^{(\tau)} \in \mathbb{R}^m$ and edge weights*

$$w_e^{(\tau)}(e) := \sigma(s_e^{(\tau)}(e)) \in (0,1), \qquad e \in [m].$$

*EHGNN-C samples a permutation $\Gamma^{(\tau)} \sim \mathrm{PL}(\boldsymbol{w}_e^{(\tau)})$ and selects the first $\kappa$ edges*

$$T^{(\tau)} := \{\Gamma_1^{(\tau)}, \ldots, \Gamma_\kappa^{(\tau)}\} \subseteq [m].$$

*Assume there exists $\alpha_c > 0$ such that $w_e^{(\tau)}(e) \geq \alpha_c$ for all $\tau, e$. Then the expected number of edge-mask flips between consecutive epochs satisfies*

$$\mathbb{E}\Big[|T^{(\tau+1)} \triangle T^{(\tau)}|\Big] \leq \frac{\kappa}{2\alpha_c} \ln \frac{m}{m-\kappa} \|\boldsymbol{s}_e^{(\tau+1)} - \boldsymbol{s}_e^{(\tau)}\|_1.$$

*Proof.* The result follows from the same argument as Theorem 5.1 for the fine-grained mask. In EHGNN-C the stochastic mask is generated by sampling a $\kappa$-subset of hyperedges from a Plackett–Luce distribution with weights $\boldsymbol{w}_e^{(\tau)} = \sigma(\boldsymbol{s}_e^{(\tau)})$.

Applying the total variation bound for the first $\kappa$ positions of the Plackett–Luce permutation (Lemma A.3) together with the data processing property (Lemma A.4) yields

$$\mathbb{P}\Big(T^{(\tau+1)} \neq T^{(\tau)}\Big) \leq \frac{1}{\alpha_c} \ln \frac{m}{m-\kappa} \|\boldsymbol{w}_e^{(\tau+1)} - \boldsymbol{w}_e^{(\tau)}\|_1.$$

Since $|T^{(\tau)}| = |T^{(\tau+1)}| = \kappa$, we have

$$|T^{(\tau+1)} \triangle T^{(\tau)}| \leq 2\kappa \, \mathbf{1}\{T^{(\tau+1)} \neq T^{(\tau)}\}.$$

Taking expectations and using the fact that the sigmoid is $1/4$-Lipschitz gives

$$\|\boldsymbol{w}_e^{(\tau+1)} - \boldsymbol{w}_e^{(\tau)}\|_1 \leq \frac{1}{4}\|\boldsymbol{s}_e^{(\tau+1)} - \boldsymbol{s}_e^{(\tau)}\|_1,$$

which yields the stated bound. $\qquad\square$

## A.3 One-Swap Local Optimality Check

**Theorem 5.3** (One-Swap Local Optimality Check)**.** *Let $U$ be the set of maskable units, namely incidences for EHGNN-F or hyperedges for EHGNN-C, and let $S^\star \subset U$ be a retained set of budget $B$. Define the one-swap neighborhood*

$$\mathcal{N}(S^\star) = \{(S^\star \setminus \{i\}) \cup \{j\} : i \in S^\star, \ j \in U \setminus S^\star\}.$$

*Fix trained HGNN parameters $\hat{\theta}$, and let*

$$G(S) = \mathcal{L}_{val}\big(f_{\hat{\theta}}(H \odot M_S, X), y_{val}\big).$$

*For tolerance $\epsilon \geq 0$, let $\rho_\epsilon(S^\star)$ be the fraction of one-swap neighbors $S' \in \mathcal{N}(S^\star)$ satisfying $G(S') \leq G(S^\star) - \epsilon$. If $Q$ one-swap neighbors are sampled uniformly with replacement and $B_Q$ of them are $\epsilon$-improving, then with probability at least $1 - \delta$,*

$$\rho_\epsilon(S^\star) \leq U_\delta(B_Q, Q),$$

*where $U_\delta(B_Q, Q)$ is the one-sided Clopper–Pearson upper confidence bound. In particular, if $B_Q = 0$, then*

$$\rho_\epsilon(S^\star) \leq 1 - \delta^{1/Q} \leq \frac{\log(1/\delta)}{Q}.$$

*Proof.* Condition on the learned mask $S^\star$ and the fixed trained parameters $\hat{\theta}$. A uniformly sampled one-swap neighbor is $\epsilon$-improving with probability $\rho_\epsilon(S^\star)$. Since the $Q$ sampled neighbors are drawn independently with replacement,

$$B_Q \sim \text{Binomial}(Q, \rho_\epsilon(S^\star)).$$

Inverting the binomial tail gives the one-sided Clopper–Pearson upper confidence bound $U_\delta(B_Q, Q)$, so with probability at least $1 - \delta$, $\rho_\epsilon(S^\star) \leq U_\delta(B_Q, Q)$. If $B_Q = 0$, then

$$\Pr(B_Q = 0) = (1 - \rho_\epsilon(S^\star))^Q.$$

The event $B_Q = 0$ has probability at least $\delta$ whenever $(1 - \rho_\epsilon(S^\star))^Q \geq \delta$, which implies

$$\rho_\epsilon(S^\star) \leq 1 - \delta^{1/Q}.$$

Finally, $1 - e^{-x} \leq x$ with $x = \log(1/\delta)/Q$ gives $1 - \delta^{1/Q} \leq \log(1/\delta)/Q$. $\qquad\square$

## B  Computational Complexity

We compare the per-layer computational complexity of full HGNNs with our fine-grained (EHGNN-F) and coarse-grained (EHGNN-C) sparsification variants. Let $n, m, t, F$, and $k$ denote the number of nodes, number of hyperedges, number of node-hyperedge incidence pairs in $\boldsymbol{H}$, feature dimension, and number of node-hyperedge incidence pairs in $\tilde{\boldsymbol{H}}$ respectively. In addition, let $h$ denote the hidden layer size of the MLP, and let $\kappa$ denote the number of hyperedges retained after sparsification, meaning, $\kappa \approx m.k/t$, where $t = \sum_{e \in E} |e|$. Table 6 shows that EHGNN-F reduces activation and computation overhead via incidence-level sparsification, but incurs a higher parameter cost. EHGNN-C reduces parameter overhead by scoring hyperedges via a shared MLP over pooled node features, sacrificing fine-grained control for scalability. Both variants are more scalable than full HGNN.

**Full HGNN.** Node-to-edge and edge-to-node aggregations per layer is typically done via sparse matrix–dense matrix multiplication (SpMM). In particular, node-to-edge aggregation is done via $\boldsymbol{H}^T \boldsymbol{X} \in \mathbb{R}^{m \times F}$ to construct hyperedge embedding, while edge-to-node aggregation is done via $\boldsymbol{H}(\boldsymbol{H}^T \boldsymbol{X}) \in \mathbb{R}^{n \times F}$ to construct embedding of the nodes in the next layer. Both aggregations have a time complexity $\mathcal{O}(tF)$. The space complexity to hold the intermediate results is $\mathcal{O}(tF)$. This is because there are $t$ non-zero entries in $\boldsymbol{H}$, for each we need to conduct $F$ elementwise multiply-add operations. There is no additional parameter overhead involved in Full HGNN training.

Table 6: Time and space (activation) complexity, and parameter overhead comparison. Here $F =$ feature dimension, $m = |E|$ is the original number of hyperedges, $t = \sum_{e \in E} |e|$ is the original incidence size, $k \approx t.\kappa/m$ indicates the reduced incidence size after sparsification, and $\kappa$ indicates the number of hyperedges after sparsification. Low-rank variants use rank $r \ll F$.

| Method | Time | Space/Activation | Param. overhead |
|---|---|---|---|
| Full HGNN | $\mathcal{O}(tF)$ | $\mathcal{O}(tF)$ | None |
| EHGNN-F | $\mathcal{O}(t \log k + kF)$ | $\mathcal{O}(t + kF)$ | $\mathcal{O}(t)$ |
| EHGNN-F (cond) | $\mathcal{O}(t \log k + kF + tF)$ | $\mathcal{O}(t + kF)$ | $\mathcal{O}(F)$ |
| EHGNN-F (cond, LR) | $\mathcal{O}(nFr + t \log k + kF + tr)$ | $\mathcal{O}(t + kF)$ | $\mathcal{O}(Fr + mr)$ |
| EHGNN-C | $\mathcal{O}(m \log \kappa + kF)$ | $\mathcal{O}(m + kF)$ | $\mathcal{O}(m)$ |
| EHGNN-C (cond) | $\mathcal{O}(tF + m \log \kappa + kF + mF)$ | $\mathcal{O}(m + kF)$ | $\mathcal{O}(F)$ |
| EHGNN-C (cond, LR) | $\mathcal{O}(nFr + m \log \kappa + kF + tr + mr)$ | $\mathcal{O}(m + kF)$ | $\mathcal{O}(Fr + mr)$ |

**Fine-grained EdgeMask-HGNN variants.** *Time complexity:* EHGNN-F incurs computational cost in two main stages: sparsification and message passing. During sparsification, the model computes sigmoid scores for all $t$ incidence logits in $\mathcal{O}(t)$ time. To select the top-$k$ incidences, it uses PL sampling, costing $\mathcal{O}(t \log k)$. Once the top-$k$ mask is applied, message passing is executed over the reduced incidence matrix containing $k$ incidence pairs. Each such pair involves computations over feature vectors of dimension $F$, resulting in a total message passing cost of $\mathcal{O}(kF)$. Summing these components, the overall time complexity per forward pass is: $\mathcal{O}(t \log k + kF)$.

In EHGNN-F(cond), computing incidence scores requires first aggregating node features via $\boldsymbol{H}^T \boldsymbol{X}$ to obtain hyperedge embeddings $\hat{X}_e$, concatenating with node features $\boldsymbol{X}_v$, and then evaluating a shared scorer on each original incidence pair $(v, e)$. Aggregation costs $\Theta(tF)$, Concatenation costs $\Theta(2tF)$, and evaluating the MLP scorer with hidden width $h$ for all incidences costs $\Theta(tFh)$. Since $h$ is a constant, the overall cost of computing scores $s_{v,e}$ is $\mathcal{O}(tF)$. Together with the PL sampling cost and message passing cost, the overall time complexity of EHGNN-F(cond) is $\mathcal{O}(t \log k + kF + tF)$.

In EHGNN-F(cond,LR), the node factor construction costs $\mathcal{O}(nFr)$, since it involves multiplying matrices of dimensions $n \times F$ and $F \times r$. Incidence scorer, which computes $r$-dimensional dot product costs $\Theta(tr)$. The top-$k$ selection cost and message passing costs remain the same as EHGNN-F. This yields total runtime complexity of $\mathcal{O}(nFr + t \log k + kF + tr)$.

*Space/Activation complexity:* The space complexity of EHGNN-F consists of both the memory required for storing scores $s_{v,e}$ and intermediate activations stored during HGNN message passing. First, the model learns a scalar mask logit $s_{v,e}$ for every incidence pair (v,e), totaling $\mathcal{O}(t)$ persistent memory. During forward propagation, all $t$ scores are passed through a sigmoid and retained in memory for use in the straight-through estimator, contributing an additional $\mathcal{O}(t)$ temporary memory. The model then selects the top-$k$ incidences and performs message passing only over this pruned subset, requiring storage of $\mathcal{O}(kF)$ for the selected node or edge features. Thus, the total space complexity is: $\mathcal{O}(t + kF)$. This becomes prohibitive in large, dense hypergraphs where $t \gg n, m$; however, it is still more efficient than Full HGNN's space complexity $\mathcal{O}(tF)$.

In EHGNN-F(cond), the activation cost remains $\mathcal{O}(t + kF)$, because the pooled embeddings $\hat{\boldsymbol{X}}_e$ are intermediate tensors that do not persist once the scorers are computed. Similarly, in EHGNN-F(cond,LR) the activation cost remains $\mathcal{O}(t + kF)$.

*Parameter overhead:* The parameter overhead in EHGNN-F originates from its fine-grained masks. For each incidence pair, the model maintains a dedicated scalar parameter. This leads to a total parameter count of: $\mathcal{O}(t)$.

However, in EHGNN-F(cond), the parameter overhead originates from the MLP. A 2-layer MLP with $h$ hidden layers costs $\mathcal{O}(hF)$. Assuming $h$ as constant, the overall parameter overhead of EHGNN-F(cond) is $\mathcal{O}(F)$.

In EHGNN-F(cond,LR), the parameter overhead comes from node feature projection matrix $\boldsymbol{W}_x \in \mathbb{R}^{F \times r}$ and hyperedge factor $Z \in \mathbb{R}^{m \times r}$, incurring a total overhead of $\mathcal{O}(Fr + mr)$.

**Coarse-grained EdgeMask-HGNN variants.** We present their complexity in Table 6.

## C   Additional Experiments

### C.1   Effectiveness on heterophilic hypergraphs.

Li et al. (2025) proposed a synthetic dataset containing hypergraphs with various homophily ratios. We evaluate Full triaing, EHGNN-F and EHGNN-C on this dataset to understand the effectiveness of EdgeMask-HGNN on synthetically generated hypergraphs with controlled homophily ratio. The results with 50% sparsification are presented in Figure 6.

On highly heterophilic hypergraphs (e.g., homophily ratio = 0.3 and 0.4), where connected nodes often have dissimilar labels, EHGNN-C and EHGNN-F outperforms Full training. This suggests that sparsification

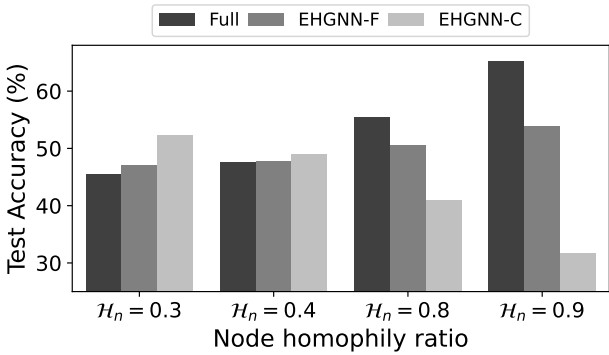

Figure 6: Performance of EdgeMask-HGNN on synthetic hypergraphs with the same number of nodes but different node-homophily ratios.

prunes cross-class connections that are harmful for label propagation in highly heterophilic settings. However, as the hypergraph becomes more homophillic, sparsification hurts the performance.

The performance of EHGNN-F becomes better than that of EHGNN-C as hypergraphs become more homophilic (e.g., homophily ratio = 0.8 and 0.9). This suggests that across homophily ratios, EHGNN-F better maintains its performance by selectively pruning incidences that adversely affect the downstream node classification accuracy.

Furthermore, we have analyzed their performance gain relative to Full training by examining the homophily ratios across various hypergraph datasets. Figure 7 indicates a negative correlation, which suggests that the proposed models perform better in low homophily (high heterophily) regimes than in high homophily regimes. The Spearman correlation between the accuracy improvement over Full and edge homophily is $-0.607$ for EHGNN-C and $-0.519$ for EHGNN-F.

## C.2 Adaptability to existing HGNN backbones.

EdgeMask-HGNN is model-agnostic and easily adaptable to different hypergraph architectures. To demonstrate this, we have employed EHGNN-F into existing architectures such as AllSetTransformer (Chien et al., 2021), ED-HNN (Wang et al., 2023), and a heterophily-specific architecture– HyperUFG (Li et al., 2025). The performance comparison is presented in Table 7. We observe that EHGNN-F achieves comparable and sometimes better performance across these three HGNN backbones. This indicates that the sparsification mechanism is flexible and can act as a plug-in module without significantly degrading the existing HGNNs' representational power.

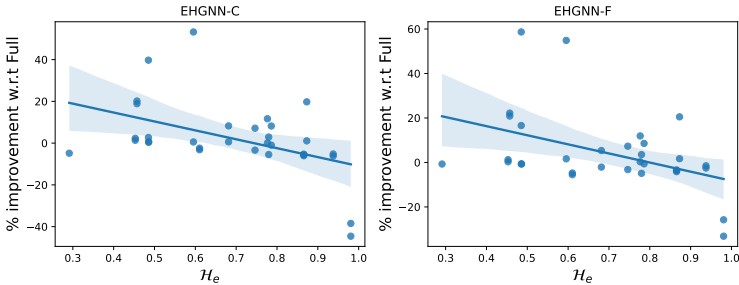

Figure 7: Correlation between Edge homophily vs performance gain of EHGNN-C and EHGNN-F. EdgeMask-HGNN variants offer more accuracy gain near low homophily/high heterophily regimes.

Table 7: Comparing various state-of-the-art HGNNs and their EHGNN-F enhanced counterparts (sparsity = 50%).

| Models | 20news | Actor | Citeseer | Cora | Cora-CA | DBLP-CA | House | ModelNet40 |
|---|---|---|---|---|---|---|---|---|
| AllSetTrans. | $82.23 \pm 0.26$ | $67.24 \pm 0.54$ | $\mathbf{67.37 \pm 0.92}$ | $\mathbf{76.43 \pm 0.45}$ | $\mathbf{81.71 \pm 1.23}$ | $\mathbf{91.11 \pm 0.06}$ | $\mathbf{100.00 \pm 0.00}$ | $\mathbf{97.94 \pm 0.15}$ |
| AllSetTrans.+EHGNN-F | $\mathbf{82.30 \pm 0.15}$ | $\mathbf{85.41 \pm 0.29}$ | $66.59 \pm 0.93$ | $73.77 \pm 1.13$ | $77.28 \pm 1.59$ | $88.48 \pm 0.21$ | $98.33 \pm 2.35$ | $97.71 \pm 0.08$ |
| ED-GNN | $\mathbf{82.36 \pm 0.16}$ | $67.73 \pm 0.10$ | $\mathbf{69.42 \pm 0.50}$ | $\mathbf{78.14 \pm 0.50}$ | $\mathbf{82.36 \pm 0.65}$ | $\mathbf{91.44 \pm 0.09}$ | $93.31 \pm 4.74$ | $97.36 \pm 0.12$ |
| ED-GNN+EHGNN-F | $80.66 \pm 0.08$ | $\mathbf{83.10 \pm 0.07}$ | $69.20 \pm 0.76$ | $75.45 \pm 0.78$ | $78.97 \pm 0.85$ | $89.66 \pm 0.18$ | $\mathbf{97.89 \pm 1.08}$ | $\mathbf{97.63 \pm 0.08}$ |
| HyperUFG | $\mathbf{81.74 \pm 0.24}$ | $75.94 \pm 1.73$ | $69.90 \pm 0.62$ | $\mathbf{74.12 \pm 0.46}$ | $\mathbf{74.24 \pm 0.50}$ | $86.53 \pm 0.08$ | $\mathbf{100.00 \pm 0.00}$ | $91.35 \pm 0.10$ |
| HyperUFG+EHGNN-F | $81.71 \pm 0.21$ | $\mathbf{80.27 \pm 0.89}$ | $\mathbf{69.93 \pm 0.48}$ | $74.03 \pm 0.32$ | $74.21 \pm 0.31$ | $\mathbf{86.66 \pm 0.06}$ | $\mathbf{100.00 \pm 0.00}$ | $\mathbf{91.82 \pm 0.21}$ |
| | Mushroom | NTU | Pokec | PubMed | Twitch | Walmart | Yelp | |
| AllSetTrans. | $\mathbf{100.00 \pm 0.00}$ | $88.23 \pm 0.43$ | $59.25 \pm 0.35$ | $87.90 \pm 0.42$ | $\mathbf{50.85 \pm 0.27}$ | $99.79 \pm 0.04$ | OOM | |
| AllSetTrans.+EHGNN-F | $99.98 \pm 0.04$ | $\mathbf{89.03 \pm 0.43}$ | $\mathbf{59.64 \pm 0.63}$ | $\mathbf{87.91 \pm 0.30}$ | $50.52 \pm 0.05$ | $\mathbf{99.98 \pm 0.01}$ | $\mathbf{31.01 \pm 0.58}$ | |
| ED-GNN | $\mathbf{99.82 \pm 0.07}$ | $\mathbf{89.38 \pm 0.41}$ | $58.94 \pm 0.67$ | $\mathbf{88.02 \pm 0.25}$ | $50.14 \pm 0.04$ | $99.34 \pm 0.17$ | $30.49 \pm 1.69$ | |
| ED-GNN+EHGNN-F | $99.28 \pm 0.14$ | $89.15 \pm 0.46$ | $\mathbf{59.00 \pm 0.44}$ | $87.98 \pm 0.21$ | $\mathbf{50.20 \pm 0.14}$ | $\mathbf{99.69 \pm 0.08}$ | $\mathbf{31.51 \pm 0.42}$ | |
| HyperUFG | $89.31 \pm 3.51$ | $69.54 \pm 0.76$ | $55.78 \pm 1.01$ | $86.22 \pm 0.50$ | $49.36 \pm 0.21$ | $\mathbf{100.00 \pm 0.00}$ | $29.17 \pm 0.94$ | |
| HyperUFG+EHGNN-F | $\mathbf{89.39 \pm 4.85}$ | $\mathbf{71.13 \pm 0.82}$ | $\mathbf{56.03 \pm 0.83}$ | $\mathbf{86.30 \pm 0.39}$ | $\mathbf{49.50 \pm 0.41}$ | $\mathbf{100.00 \pm 0.00}$ | $\mathbf{29.54 \pm 0.71}$ | |

### C.3 Exact vs. Approximate effective resistance sparsification.

In addition to the exact effective-resistance-based sparsifier, which requires a dense pseudoinverse of the hypergraph Laplacian, we implemented an approximate spectral sparsifier using a Hutchinson-type random projection estimator for $\mathrm{diag}(L^+)$. Specifically, we sample vectors $g$ from Rademacher distribution, solve $(L + \lambda I)z = g$ via Conjugate Gradient, and approximate $\mathrm{diag}(L^+) \approx \mathbb{E}[z \odot g]$. These approximate resistances are then aggregated per hyperedge as in the exact method. This follows the standard line of random-projection-based spectral sparsification and avoids forming the pseudoinverse explicitly.

In practice, this approximation yields some loss in accuracy compared to full training while being significantly more scalable. Table 8 shows that on DBLP-CA it reduces memory consumption than exact sparsification, while enables training on Yelp and Walmart.

### C.4 Scalability

Table 9 reports peak GPU memory across methods at the same sparsity budget.

*On small-scale datasets (e.g., NTU2012, Cora-CA):* On small datasets, all methods exhibit comparable peak memory since the full incidence structure already fits comfortably in GPU memory. In this regime, sparsification yields only marginal gains, and differences across methods are small (typically within ∼0.02–0.05 GB). Notably, HSL incurs substantially higher memory overhead (e.g., 0.23 GB vs. 0.08 GB on NTU2012).

*On moderately large hypergraphs (e.g., DBLP-CA, Walmart):* As the incidence count increases, clear differences emerge. On DBLP-CA and Walmart, activation memory begins to dominate, and sparsification becomes critical. Random and degree-based pruning achieve the largest memory reductions. In contrast, EHGNN-F achieves consistent memory reductions compared to Full (e.g., 1.12 GB on DBLP-CA, 2.44 GB on Walmart) while preserving strong predictive performance. Finally, HSL shows substantially higher memory usage (e.g., 2.65 GB on DBLP-CA and 7.39 GB on Walmart), indicating poor scalability.

*On large-scale hypergraphs (Yelp, Trivago):* At large scale, memory behavior becomes more heterogeneous. On Yelp, Full already requires 20.02 GB, and most learnable sparsifiers provide only modest reductions (e.g., EHGNN-F: 18.45 GB, EHGNN-C: 18.71 GB). Feature-conditioned variants introduce additional overhead due to per-incidence feature aggregation and scoring, leading to increased memory or even OOM (EHGNN-

Table 8: Comparison (Accuracy, peak memory) between exact and approximate effective resistance based sparsification.

| | DBLP-CA | Yelp | Walmart |
|---|---|---|---|
| Spectral (exact Effective resistance) | OOM | OOM | OOM |
| Spectral (approximate Effective resistance) | 88.79, 1.15 GB | 32.88, 16.02 GB | 94.76, 2.23 GB |

Table 9: Mean peak GPU memory usage (in GB) during training. **Bold**=best, ↓=%reduction w.r.t. Full.

| Algorithms | 20news | ModelNet40 | Mushroom | NTU2012 | Actor | Citeseer | Cora-CA | DBLP-CA |
|---|---|---|---|---|---|---|---|---|
| Full | 0.44 | 0.41 | 0.26 | 0.08 | 0.31 | 0.18 | 0.10 | 1.29 |
| Random | **0.31 (29% ↓)** | **0.28 (31% ↓)** | **0.18 (31% ↓)** | **0.06 (24% ↓)** | **0.29 (6% ↓)** | **0.17 (5% ↓)** | **0.09 (11% ↓)** | **1.06 (18% ↓)** |
| Edgedeg | 0.36 (18% ↓) | 0.29 (29% ↓) | 0.23 (10% ↓) | 0.06 (24% ↓) | 0.32 (2% ↑) | 0.18 (3% ↓) | 0.10 (5% ↓) | 1.17 (10% ↓) |
| Spectral | 0.39 (12% ↓) | 0.29 (29% ↓) | 0.24 (6% ↓) | 0.06 (24% ↓) | 0.32 (4% ↑) | 0.18 (3% ↓) | 0.10 (5% ↓) | OOM |
| HSL | 1.15 (161% ↑) | 1.08 (161% ↑) | 0.70 (175% ↑) | 0.23 (178% ↑) | 0.82 (164% ↑) | 0.30 (65% ↑) | 0.21 (105% ↑) | 2.65 (104% ↑) |
| HSL w/ aug. | 1.16 (163% ↑) | 1.09 (165% ↑) | 0.70 (171% ↑) | 0.23 (190% ↑) | 0.83 (169% ↑) | 0.30 (69% ↑) | 0.21 (113% ↑) | 2.66 (106% ↑) |
| EHGNN-F | 0.41 (7% ↓) | 0.35 (14% ↓) | 0.24 (6% ↓) | 0.08 (9% ↓) | 0.34 (10% ↑) | 0.18 (4% ↓) | 0.09 (8% ↓) | 1.12 (14% ↓) |
| EHGNN-C | 0.46 (5% ↑) | 0.37 (12% ↓) | 0.28 (11% ↑) | 0.08 (9% ↓) | 0.36 (14% ↑) | 0.18 (4% ↓) | 0.09 (8% ↓) | 1.18 (9% ↓) |
| EHGNN-F(cond) | 0.47 (6% ↑) | 0.41 (1% ↓) | 0.25 (2% ↓) | 0.08 (1% ↑) | 0.37 (18% ↑) | 0.35 (88% ↑) | 0.16 (60% ↑) | 2.71 (109% ↑) |
| EHGNN-C(cond) | 0.47 (8% ↑) | 0.37 (10% ↓) | 0.27 (5% ↑) | 0.08 (8% ↓) | 0.35 (13% ↑) | 0.20 (9% ↑) | 0.10 (3% ↓) | 1.25 (3% ↓) |
| EHGNN-F (cond,LR) | 0.41 (7% ↓) | 0.35 (14% ↓) | 0.24 (6% ↓) | 0.08 (9% ↓) | 0.34 (10% ↑) | 0.18 (3% ↓) | 0.10 (7% ↓) | 1.13 (13% ↓) |
| EHGNN-C (cond,LR) | 0.45 (2% ↑) | 0.37 (11% ↓) | 0.25 (1% ↓) | 0.08 (9% ↓) | 0.36 (14% ↑) | 0.18 (4% ↓) | 0.09 (9% ↓) | 1.18 (9% ↓) |
| | Cora | House | Pokec | PubMed | Twitch | Walmart | Yelp | Trivago |
| Full | 0.10 | 0.08 | 0.19 | 0.43 | 0.25 | 2.81 | 20.02 | 6.41 |
| Random | **0.09 (11% ↓)** | **0.05 (31% ↓)** | **0.18 (8% ↓)** | **0.36 (18% ↓)** | **0.21 (13% ↓)** | **1.86 (34% ↓)** | **11.14 (44% ↓)** | **4.69 (27% ↓)** |
| Edgedeg | 0.10 (8% ↓) | 0.06 (21% ↓) | 0.19 (5% ↓) | 0.39 (10% ↓) | 0.22 (10% ↓) | 2.23 (20% ↓) | 15.92 (20% ↓) | 5.25 (18% ↓) |
| Spectral | 0.10 (9% ↓) | 0.06 (16% ↓) | 0.19 (4% ↓) | 0.39 (9% ↓) | 0.23 (7% ↓) | OOM | OOM | OOM |
| HSL | 0.33 (211% ↑) | 0.23 (211% ↑) | 0.37 (90% ↑) | 0.92 (113% ↑) | 0.52 (112% ↑) | 7.39 (163% ↑) | OOM | 13.35 (108% ↑) |
| HSL w/ aug. | 0.33 (226% ↑) | 0.24 (195% ↑) | 0.37 (96% ↑) | 0.93 (116% ↑) | 0.53 (111% ↑) | 7.46 (166% ↑) | OOM | 13.46 (110% ↑) |
| EHGNN-F | 0.09 (9% ↓) | 0.07 (5% ↓) | 0.18 (8% ↓) | 0.38 (13% ↓) | 0.22 (9% ↓) | 2.44 (13% ↓) | 18.45 (8% ↓) | 6.15 (4% ↓) |
| EHGNN-C | 0.10 (9% ↓) | 0.07 (1% ↓) | 0.18 (5% ↓) | 0.39 (11% ↓) | 0.22 (12% ↓) | 2.49 (11% ↓) | 18.71 (7% ↓) | 6.67 (4% ↑) |
| EHGNN-F(cond) | 0.17 (61% ↑) | 0.07 (3% ↓) | 0.18 (6% ↓) | 0.51 (18% ↑) | 0.23 (8% ↓) | 2.53 (10% ↓) | OOM | 4.74 (26% ↓) |
| EHGNN-C(cond) | 0.10 (2% ↓) | 0.08 (5% ↑) | 0.19 (5% ↓) | 0.40 (7% ↓) | 0.22 (12% ↓) | 2.58 (8% ↓) | OOM | 7.07 (10% ↑) |
| EHGNN-F (cond,LR) | 0.10 (9% ↓) | 0.07 (4% ↓) | 0.18 (7% ↓) | 0.38 (11% ↓) | 0.23 (9% ↓) | 2.46 (12% ↓) | 18.64 (7% ↓) | 6.23 (3% ↓) |
| EHGNN-C (cond,LR) | 0.10 (8% ↓) | 0.07 (6% ↓) | 0.18 (5% ↓) | 0.38 (12% ↓) | 0.22 (12% ↓) | 2.42 (14% ↓) | 18.83 (6% ↓) | 6.68 (4% ↑) |

F(cond), EHGNN-C(cond)). In contrast, simpler sparsifiers (Random) achieve the largest reductions (down to 11.14 GB), though at a significant accuracy cost. On Trivago, which is large but less extreme, EHGNN variants consistently reduce memory relative to Full (e.g., 6.15 GB vs. 6.41 GB), with feature-conditioned variants remaining competitive unless additional overhead dominates.

Overall, peak GPU memory is primarily governed by hypergraph scale, particularly the total incidence count $t$. Unsupervised sparsifiers, such as random and degree-based sparsifiers, have the lowest memory footprint, since they do not need to perform additional computation during training and benefit from fewer retained incidences during message passing. However, as observed in Table 2, their performance is often superseded by EdgeMask-HGNN variants. EHGNN-F provides a favorable trade-off, achieving consistent memory reductions while maintaining strong performance. EHGNN-C offers similar benefits, but is more sensitive to hyperedge structure.

# D    Model Parameter Sizes

We report the model parameter sizes in Table 10 for a complete understanding of the parameter overhead, which was discussed earlier theoretically. Recall that EHGNN-C and EHGNN-F have parameter overhead of $\mathcal{O}(m)$ and $\mathcal{O}(t)$, respectively, where $m$ is the number of hyperedges and $t = \sum_{e \in E} |e|$ is the number of node–hyperedge pairs. Since $m << t$ in most of the datasets, we observe that EHGNN-C has a smaller parameter overhead than EHGNN-F in general.

EHGNN-F has a higher parameter overhead than Full, Random, Degdist, and Spectral due to the fact that it needs to learn the mask conditioned on the supervision signal from the downstream task, requiring additional parameters. While EHGNN-F introduces a higher parameter overhead, its memory usage ($\mathcal{O}(t + kF)$) is dominated by sparsified message passing layers. The substantial reduction in active node–hyperedge incidences ($k \ll t$) leads to a much smaller activation footprint. Thus, despite having more parameters, EHGNN-F generally consumes less memory than the Full training (see in earlier Table 9).

Table 10: Number of model parameters (integers) across datasets for each algorithm at sparsity = 50%. OOM=Out-of-Memory.

| Algorithms | 20news | ModelNet40 | Mushroom | NTU2012 | Actor | Citeseer | Cora-CA | DBLP-CA |
|---|---|---|---|---|---|---|---|---|
| Full | 53764 | 72745 | 12802 | 86083 | 27651 | 1899526 | 737799 | 733190 |
| Random | 53764 | 72745 | 12802 | 86083 | 27651 | 1899526 | 737799 | 733190 |
| Edgedeg | 53764 | 72745 | 12802 | 86083 | 27651 | 1899526 | 737799 | 733190 |
| Spectral | 53764 | 72745 | 12802 | 86083 | 27651 | 1899526 | 737799 | OOM |
| HSL | 88718 | 91123 | 78448 | 92813 | 82153 | 557238 | 262203 | 261098 |
| EHGNN-F | 119215 | 134300 | 53422 | 96143 | 81023 | 1902979 | 742384 | 832751 |
| EHGNN-C | 53864 | 85056 | 13100 | 88095 | 37815 | 1900605 | 738871 | 755553 |
| EHGNN-F (cond) | 60229 | 79210 | 14275 | 92548 | 30916 | 2136583 | 829576 | 824455 |
| EHGNN-C (cond) | 57029 | 76010 | 13571 | 89348 | 29316 | 2018087 | 783720 | 778855 |
| EHGNN-F (cond, LR) | 54564 | 122389 | 14082 | 94531 | 68507 | 1918654 | 747819 | 828342 |
| EHGNN-C (cond, LR) | 54564 | 122389 | 14082 | 94531 | 68507 | 1918654 | 747819 | 828342 |
| | Cora | House | Pokec | PubMed | Twitch | Walmart | Yelp | Trivago |
| Full | 737799 | 2562 | 34818 | 258051 | 5122 | 11787 | 958473 | 498848 |
| Random | 737799 | 2562 | 34818 | 258051 | 5122 | 11787 | 958473 | 498848 |
| Edgedeg | 737799 | 2562 | 34818 | 258051 | 5122 | 11787 | 958473 | 498848 |
| Spectral | 737799 | 2562 | 34818 | 258051 | 5122 | OOM | OOM | OOM |
| HSL | 651963 | 75848 | 84038 | 140653 | 76498 | 77603 | OOM | 124858 |
| EHGNN-F | 742585 | 14405 | 40320 | 292680 | 21478 | 472417 | 5482067 | 1225709 |
| EHGNN-C | 739378 | 2903 | 37224 | 266014 | 7749 | 81693 | 1637775 | 732050 |
| EHGNN-F (cond) | 829576 | 2755 | 39043 | 290116 | 5635 | 12556 | OOM | 255457 |
| EHGNN-C (cond) | 783720 | 2691 | 36963 | 274116 | 5411 | 12204 | OOM | 508513 |
| EHGNN-F (cond, LR) | 749847 | 3934 | 44702 | 291903 | 15658 | 291455 | 3683129 | 1432856 |
| EHGNN-C (cond, LR) | 749847 | 3934 | 44702 | 291903 | 15658 | 291455 | 3683129 | 1432856 |

## E EdgeMask-HGNN on Inductive Link Prediction Task

While the main experiments evaluate EdgeMask-HGNN on node classification task, we also investigate its adaptability on link prediction task. To that end, we adapt **Hyper-SAGNN** (Zhang et al., 2020), a self-attention-based architecture designed for this task.

**Link prediction task.** Let $\mathcal{V}$ denote the set of nodes and let $\mathcal{E} = \{e_1, \ldots, e_m\}$ be the set of observed hyperedges, where each hyperedge $e \subseteq \mathcal{V}$ represents a higher-order relation among multiple nodes. The goal of hyperedge prediction is to learn a scoring function

$$f_\theta : 2^{\mathcal{V}} \to [0, 1],$$

that estimates the likelihood that a candidate node set forms a valid hyperedge. In this paper, we focus on a simpler variant of link prediction where all edges have the same cardinality. Thus, in this paper, the scoring function we intend to learn is the following:

$$f_\theta : [\mathcal{V}]^k \to [0, 1], \tag{14}$$

where $[\mathcal{V}]^k$ indicates the set of all k-sets of $\mathcal{V}$. Formally, given a candidate hyperedge $e = \{v_1, \ldots, v_k\}$, the model predicts

$$\hat{y}_e = f_\theta(e), \tag{15}$$

where $\hat{y}_e$ is the probability that $e$ corresponds to a valid higher-order relation.

**Inductive setting.** In the inductive link prediction setting, the model is trained using a subset of observed hyperedges $\mathcal{E}_{train} \subset \mathcal{E}$, while validation and test hyperedges are drawn from $\mathcal{E}_{val}$ and $\mathcal{E}_{test}$. Importantly, node embeddings must generalize to previously unseen hyperedges composed of nodes from $\mathcal{V}$.

To train the model, we construct a binary classification dataset consisting of positive and negative hyperedges. Positive examples correspond to observed hyperedges $e \in \mathcal{E}_{train}$, while negative examples are generated by corrupting one position of a positive hyperedge with another node of the same type and rejecting corrupted tuples that already appear in the observed hyperedge set. Each example is assigned a label

$$y_e = \begin{cases} 1 & e \in \mathcal{E}_{train}, \\ 0 & \text{otherwise.} \end{cases} \tag{16}$$

The model parameters are optimized using the binary cross-entropy loss

$$L = \frac{1}{|\mathcal{B}|} \sum_{e \in \mathcal{B}} \ell(\hat{y}_e, y_e), \tag{17}$$

where $\mathcal{B}$ denotes a minibatch of candidate hyperedges.

**Hyper-SAGNN backbone.** Given a candidate hyperedge (tuple) $e = (v_1, \ldots, v_k)$, each node $v_i$ is first mapped to an initial embedding $\boldsymbol{h}_{v_i}^{(0)}$. These embeddings are then processed through two parallel transformations.

First, a position-wise feed-forward network produces the *static embedding*

$$\boldsymbol{h}_{v_i}^{stat} = \tanh(W_s^\top \boldsymbol{h}_{v_i}^{(0)}), \tag{18}$$

which depends only on the individual node $v_i$ and is independent of the other nodes in the hyperedge.

Second, a multi-head self-attention layer is applied over the set $\{\boldsymbol{h}_{v_1}^{(0)}, \ldots, \boldsymbol{h}_{v_k}^{(0)}\}$ to produce the *dynamic embedding*

$$\boldsymbol{h}_{v_i}^{dyn} = \text{Attention}(\boldsymbol{h}_{v_i}^{(0)}, \{\boldsymbol{h}_{v_j}^{(0)}\}_{j=1}^k), \tag{19}$$

which captures interactions among all nodes within the hyperedge.

Thus, $\boldsymbol{h}_{v_i}^{stat}$ represents node-specific information that is invariant across hyperedges, while $\boldsymbol{h}_{v_i}^{dyn}$ encodes context-dependent information that varies with the composition of the hyperedge.

Finally, the resulting edge representation is computed as

$$\boldsymbol{z}_e = \frac{1}{|e|} \sum_{v_i \in e} (\boldsymbol{h}_{v_i}^{dyn} - \boldsymbol{h}_{v_i}^{stat})^2, \tag{20}$$

which is passed to a classification head to predict the validity of the hyperedge. Hyper-SAGNN treats each hyperedge independently; thus, the predictions depend only on the nodes contained in the target hyperedge.

**EdgeMask-HGNN adaptation** To incorporate EdgeMask-HGNN variants, we introduce a learnable sparsification module that assigns an importance score $\boldsymbol{m}_e \in [0, 1]$ to each hyperedge. Unlike the baseline Hyper-SAGNN model, these scores are used to construct contextualized node embeddings prior to the edge scoring stage.

For each hyperedge $e$, we compute an edge-level message

$$\boldsymbol{g}_e = \frac{1}{|e|} \sum_{u \in e} W h_u^{(0)}, \tag{21}$$

and update node embeddings using masked hypergraph aggregation

$$\boldsymbol{h}_v^{(1)} = \boldsymbol{h}_v^{(0)} + \sum_{e:v \in e} \boldsymbol{m}_e \boldsymbol{g}_e. \tag{22}$$

For a node $v$, its contextualized embeddings $\boldsymbol{h}_v^{(1)}$ are then used as inputs to Hyper-SAGNN:

$$\hat{y}_e = f_\theta(\{\boldsymbol{h}_v^{(1)} : v \in e\}), \tag{23}$$

Table 11: Inductive hyperedge link-prediction performance for Hyper-SAGNN and EdgeMask-HGNN contextual sparse aggregation variants at 50% retention. Results are mean $\pm$ standard deviation over five seeds. Metrics are accuracy, AUC, and AUPR under the same train/validation/test protocol used by Hyper-SAGNN. The contextualization ablations in Table 12 isolate full-context, random-context, and learned-context aggregation.

| Model | WordNet (user, relation, tail) $|V| = (40504, 18, 40551)$ | | | Drug (user, drug, reaction) $|V| = (12, 1076, 6398)$ | | | MovieLens (user, movie, tag) $|V| = (2113, 5908, 9079)$ | | |
|---|---|---|---|---|---|---|---|---|---|
| | **Acc** | **AUC** | **AUPR** | **Acc** | **AUC** | **AUPR** | **Acc** | **AUC** | **AUPR** |
| Hyper-SAGNN | $88.26 \pm 0.12$ | $87.72 \pm 0.42$ | $68.34 \pm 0.21$ | $93.97 \pm 0.02$ | $95.64 \pm 0.04$ | $89.02 \pm 0.05$ | $90.77 \pm 0.06$ | $91.00 \pm 0.05$ | $76.04 \pm 0.43$ |
| Random | $93.32 \pm 0.15$ | $95.94 \pm 0.15$ | $87.61 \pm 0.30$ | $96.49 \pm 0.02$ | $98.66 \pm 0.04$ | $95.72 \pm 0.06$ | $95.02 \pm 0.05$ | $97.27 \pm 0.06$ | $91.91 \pm 0.23$ |
| EdgeDeg | $93.17 \pm 0.28$ | $95.70 \pm 0.28$ | $87.05 \pm 0.88$ | $96.50 \pm 0.07$ | $98.65 \pm 0.05$ | $95.67 \pm 0.15$ | $94.97 \pm 0.03$ | $97.29 \pm 0.03$ | $91.72 \pm 0.09$ |
| Spectral | $91.16 \pm 0.17$ | $92.00 \pm 0.54$ | $79.14 \pm 1.01$ | $96.95 \pm 0.05$ | $98.71 \pm 0.04$ | $96.07 \pm 0.02$ | $93.73 \pm 0.16$ | $95.85 \pm 0.09$ | $87.97 \pm 0.29$ |
| EHGNN-F | $96.35 \pm 0.17$ | $96.86 \pm 0.50$ | $92.41 \pm 0.37$ | $96.96 \pm 0.07$ | $97.49 \pm 0.33$ | $94.96 \pm 0.13$ | $96.40 \pm 0.02$ | $97.13 \pm 0.36$ | $93.28 \pm 0.38$ |
| EHGNN-C | $93.14 \pm 0.34$ | $95.71 \pm 0.44$ | $87.02 \pm 1.12$ | $96.51 \pm 0.03$ | $98.65 \pm 0.04$ | $95.70 \pm 0.10$ | $94.97 \pm 0.03$ | $97.18 \pm 0.02$ | $91.55 \pm 0.02$ |
| EHGNN-F(cond) | $94.89 \pm 0.07$ | $97.07 \pm 0.27$ | $91.36 \pm 0.30$ | $97.25 \pm 0.19$ | $99.14 \pm 0.16$ | $97.01 \pm 0.29$ | $96.35 \pm 0.30$ | $98.35 \pm 0.29$ | $94.55 \pm 0.88$ |
| EHGNN-C(cond) | $93.79 \pm 0.31$ | $96.43 \pm 0.46$ | $88.85 \pm 1.10$ | $96.79 \pm 0.18$ | $98.93 \pm 0.17$ | $96.22 \pm 0.30$ | $95.65 \pm 0.14$ | $97.87 \pm 0.12$ | $93.29 \pm 0.27$ |
| EHGNN-F(cond,LR) | $\mathbf{98.74 \pm 0.08}$ | $\mathbf{99.68 \pm 0.04}$ | $\mathbf{99.01 \pm 0.13}$ | $\mathbf{98.25 \pm 0.05}$ | $\mathbf{99.67 \pm 0.00}$ | $\mathbf{98.72 \pm 0.01}$ | $\mathbf{98.38 \pm 0.05}$ | $\mathbf{99.66 \pm 0.01}$ | $\mathbf{98.57 \pm 0.08}$ |
| EHGNN-C(cond,LR) | $91.73 \pm 0.14$ | $92.43 \pm 0.82$ | $81.28 \pm 0.74$ | $95.76 \pm 0.03$ | $97.68 \pm 0.04$ | $93.68 \pm 0.09$ | $93.83 \pm 0.07$ | $95.66 \pm 0.10$ | $88.23 \pm 0.14$ |

where $f_\theta$ denotes the Hyper-SAGNN encoder.

This modification introduces a lightweight form of structural reasoning into Hyper-SAGNN: predictions for an edge now depend on other hyperedges sharing nodes through the masked aggregation step. Next, we discuss the computation of mask $\boldsymbol{m}_e$.

**Mask computation and Training objective.** The mask is obtained from the underlying EdgeMask-HGNN sparsifier. We first discuss the case of coarse-grained sparsifier. Let the model parameters of the sparsifier be $\Phi$. Then the global edge mask $\boldsymbol{m}_e$ is obtained as

$$\boldsymbol{m}_e = \texttt{EdgeMask-HGNN}_\Phi(\mathcal{B}, \boldsymbol{X} = \boldsymbol{X}_\mathcal{B}), \tag{24}$$

where $\boldsymbol{X}_\mathcal{B} = (\boldsymbol{h}_v^{(0)})_{v \in \mathcal{B}}$ are the node features of the nodes in batch $\mathcal{B}$. The model is trained end-to-end using the loss $L$ defined in Equation (17), where the gradients propagate through both the Hyper-SAGNN encoder and the sparsification module.

When the sparsifier is fine-grained, we obtain $\boldsymbol{m}_e$ from aggregating learned incidence masks $\boldsymbol{m}_{v,e}$:

$$\boldsymbol{m}_{v,e} = \texttt{EdgeMask-HGNN}_\Phi(\mathcal{B}, \boldsymbol{X} = \boldsymbol{X}_\mathcal{B}), \quad \boldsymbol{m}_e = \frac{1}{|e|} \sum_{v \in e} \boldsymbol{m}_{v,e} \tag{25}$$

**Experimental results and discussion.** Table 11 summarizes the performance of the baseline Hyper-SAGNN model and the EHGNN-enhanced variant across three datasets at 50% sparsity.

The results demonstrate that contextual sparse aggregation can significantly enhance Hyper-SAGNN in inductive link prediction tasks. For instance, on WordNet, EHGNN-F(cond,LR) increases AUC from $87.72 \pm 0.42$ to $99.68 \pm 0.04$, while AUPR improves from $68.34 \pm 0.21$ to $99.01 \pm 0.13$. Similarly, on the Drug dataset, AUC increases from $95.64 \pm 0.04$ to $99.67 \pm 0.00$. To separate the benefit of contextual aggregation from the benefit of learned sparsification, we additionally ran the WordNet ablation in Table 12. The full-context model applies the same contextual aggregation but uses an all-ones mask, i.e., no sparsification. The ablation shows that much of the WordNet improvement comes from adding structural context to Hyper-SAGNN; learned masking still improves substantially over plain Hyper-SAGNN and random masked context, but is slightly below full-context aggregation on WordNet. We therefore interpret the link-prediction results as evidence for learned budgeted contextualization, where the best choice between full and sparse context depends on the dataset and the usefulness of context selection.

We also evaluate the GPS link-prediction dataset used by Hyper-SAGNN (Zhang et al., 2020). GPS is a 3-uniform heterogeneous hypergraph whose hyperedges are triples of the form (user, location, activity),

Table 12: WordNet link-prediction ablation isolating contextual aggregation from sparsification. Results are mean ± standard deviation over five seeds. "FullContext" uses the same contextual aggregation as the sparse variants but with an all-ones mask.

| Model | Acc | AUC | AUPR |
|---|---|---|---|
| Hyper-SAGNN | $88.32 \pm 0.09$ | $87.90 \pm 0.20$ | $68.66 \pm 0.30$ |
| FullContext | $\mathbf{99.35 \pm 0.03}$ | $\mathbf{99.94 \pm 0.01}$ | $\mathbf{99.75 \pm 0.01}$ |
| Random masked context | $93.28 \pm 0.11$ | $95.81 \pm 0.27$ | $87.36 \pm 0.56$ |
| EHGNN-F(cond,LR) | $98.75 \pm 0.06$ | $99.67 \pm 0.02$ | $98.99 \pm 0.07$ |

Table 13: GPS context-budget and context-noise ablation. Results are mean ± standard deviation over ten seeds. GPS has dense contextual overlap because its third node type has only five values. EHGNN-F(cond,LR) uses a 75% context budget; FullContext uses all context.

| Noise | FullContext | | Random | | EHGNN-F(cond,LR) | |
|---|---|---|---|---|---|---|
| | AUC | AUPR | AUC | AUPR | AUC | AUPR |
| 0.0 | $84.58 \pm 5.03$ | $54.12 \pm 8.88$ | $84.58 \pm 4.11$ | $54.97 \pm 6.68$ | $\mathbf{85.27 \pm 3.73}$ | $\mathbf{55.81 \pm 6.05}$ |
| 0.2 | $84.80 \pm 4.32$ | $54.89 \pm 8.15$ | $84.02 \pm 4.68$ | $54.34 \pm 9.31$ | $\mathbf{86.21 \pm 1.79}$ | $\mathbf{58.31 \pm 4.77}$ |
| 0.4 | $83.06 \pm 3.88$ | $50.78 \pm 6.92$ | $84.12 \pm 2.49$ | $54.02 \pm 5.63$ | $\mathbf{84.36 \pm 2.41}$ | $\mathbf{54.61 \pm 4.84}$ |
| 0.8 | $81.89 \pm 5.60$ | $51.25 \pm 10.97$ | $81.67 \pm 5.73$ | $50.59 \pm 10.83$ | $\mathbf{84.73 \pm 3.49}$ | $\mathbf{55.25 \pm 6.54}$ |
| 1.5 | $82.57 \pm 4.89$ | $50.94 \pm 9.67$ | $82.98 \pm 5.24$ | $52.20 \pm 9.63$ | $\mathbf{84.33 \pm 2.98}$ | $\mathbf{55.46 \pm 6.51}$ |
| 3.0 | $81.79 \pm 4.85$ | $50.07 \pm 8.56$ | $82.19 \pm 4.81$ | $51.21 \pm 8.36$ | $\mathbf{83.33 \pm 3.35}$ | $\mathbf{53.53 \pm 5.51}$ |

representing a user participating in an activity at a location. The dataset contains 146 users, 70 locations, 5 activities, and 1,436 observed hyperedges. The small number of activity nodes makes GPS a dense context-overlap setting: many candidate triples share activity nodes, so full contextual aggregation can repeatedly propagate messages through a small shared set of nodes.

**Effect of sparsification on link prediction.** The key benefit of the proposed adaptation is that it introduces structural context into a model that originally treats hyperedges independently. As a result, predictions for a candidate hyperedge can incorporate information from other related hyperedges through the aggregation step. The ablations clarify that contextualization is itself a strong architectural change. FullContext is therefore a natural no-sparsification control, not a separate contribution. Learned masks are useful as a controllable way to restrict context under a budget: they outperform random masked context on WordNet/Drug/MovieLens at matched retention, and Table 13 shows that on dense GPS they can outperform FullContext in mean AUC/AUPR under both clean and noisy context. We therefore do not claim uniform dominance of either FullContext or EHGNN; instead, FullContext is preferable when most context is helpful, while EHGNN is preferable when learned context selection reduces redundant or misleading aggregation.

# F Statistical Significance Tests

In order to evaluate whether the proposed EdgeMask variants are meaningfully superior to full hypergraph training baseline, we conduct paired (related) samples t-tests comparing Full vs. EHGNN-C and Full vs. EHGNN-F across all datasets in Table 14. Here, $* = p < 0.05$, $** = p < 0.01$, and $- = p > 0.05$ indicate statistical significance levels. All Edgemask-HGNN variants operate under a 50% sparsification budget.

First, sparsification does not universally degrade performance. On several datasets, particularly Actor, ModelNet40, Walmart, Pokec, PubMed, and Twitch, at least one EdgeMask-HGNN variant significantly improves over the full hypergraph model despite using a 50% sparsification budget. This indicates that dense hypergraphs can contain redundant or noisy high-order relations, and that principled sparsification can improve generalization by acting as a structural regularizer.

Table 14: Statistical test by conducting paired (related) samples t-tests: Full vs. EHGNN-C and Full vs. EHGNN-F. Here $* = p < 0.05, ** = p < 0.01$ and $- = p > 0.05$ indicates no statistical significance. Edgemask variants are run with 50% sparsification.

| | | EHGNN-C | | | | EHGNN-F | | | | |
|---|---|---|---|---|---|---|---|---|---|---|
| Dataset | Full | Mean Acc. | $\Delta$ | p-value | Sig. | Mean Acc. | $\Delta$ | p-value | Sig. | Better Method |
| 20news | 81.04 | 79.08 | -1.96 | 2.58e-05 | ** | 77.18 | -3.86 | 3.42e-07 | ** | Full |
| Actor | 64.54 | 76.72 | **+12.19** | 2.16e-09 | ** | 78.01 | **+13.47** | 1.33e-09 | ** | EHGNN-F |
| Citeseer | 68.19 | 68.60 | **+0.41** | 1.46e-01 | - | 66.76 | -1.43 | 1.27e-02 | * | EHGNN-C |
| Cora | 78.23 | 75.60 | -2.63 | 3.38e-04 | ** | 75.72 | -2.51 | 8.48e-05 | ** | Full |
| Cora-CA | 81.48 | 77.02 | -4.46 | 1.92e-05 | ** | 77.49 | -3.99 | 4.43e-05 | ** | Full |
| DBLP-CA | 90.77 | 85.28 | -5.49 | 5.14e-08 | ** | 87.01 | -3.75 | 7.30e-07 | ** | Full |
| House | 73.87 | 75.85 | **+1.98** | 7.03e-04 | ** | 86.13 | **+12.26** | 5.15e-07 | ** | EHGNN-F |
| ModelNet40 | 94.72 | 95.73 | **+1.01** | 1.68e-05 | ** | 96.32 | **+1.60** | 1.74e-05 | ** | EHGNN-F |
| Mushroom | 98.73 | 93.59 | -5.14 | 3.20e-05 | ** | 97.24 | -1.49 | 4.57e-05 | ** | Full |
| NTU2012 | 88.03 | 87.20 | -0.83 | 8.15e-03 | ** | 87.48 | -0.56 | 7.29e-02 | - | Full |
| Pokec | 58.36 | 59.15 | **+0.79** | 4.37e-07 | ** | 58.53 | **+0.17** | 1.40e-03 | ** | EHGNN-C |
| PubMed | 86.50 | 86.56 | **+0.06** | 2.31e-01 | - | 86.75 | **+0.25** | 7.02e-03 | ** | EHGNN-F |
| Twitch | 51.22 | 51.44 | **+0.22** | 9.97e-03 | ** | 50.84 | -0.38 | 1.10e-05 | ** | EHGNN-C |
| Trivago | 61.39 | 37.78 | -23.61 | 2.16e-05 | ** | 45.56 | -15.83 | 5.33e-05 | ** | Full |
| Walmart | 95.36 | 95.89 | **+0.53** | 7.30e-05 | ** | 96.90 | **+1.55** | 9.77e-09 | ** | EHGNN-F |
| Yelp | 33.60 | 31.98 | -1.62 | 1.14e-03 | ** | 33.36 | -0.24 | 2.89e-01 | - | Full |

Table 15: Train–test accuracy-gap analysis. Values are mean $\pm$ standard deviation over five runs. Acc. gap is train accuracy minus test accuracy.

| | Full | | EHGNN-F | | EHGNN-C | |
|---|---|---|---|---|---|---|
| Dataset | Test acc. | Gap | Test acc. | Gap | Test acc. | Gap |
| Actor | $64.54 \pm 0.03$ | $0.96 \pm 0.03$ | $\mathbf{77.96} \pm 0.05$ | $0.85 \pm 0.06$ | $76.72 \pm 0.09$ | $1.22 \pm 0.43$ |
| Twitch | $51.22 \pm 0.04$ | $0.89 \pm 0.05$ | $50.81 \pm 0.06$ | $1.47 \pm 0.08$ | $\mathbf{51.44} \pm 0.07$ | $-1.88 \pm 0.14$ |
| Pokec | $58.36 \pm 0.03$ | $1.45 \pm 0.03$ | $58.54 \pm 0.04$ | $1.79 \pm 0.03$ | $\mathbf{59.15} \pm 0.05$ | $-0.23 \pm 0.07$ |
| Walmart | $95.36 \pm 0.01$ | $-0.02 \pm 0.02$ | $\mathbf{96.93} \pm 0.03$ | $0.05 \pm 0.03$ | $95.89 \pm 0.07$ | $0.03 \pm 0.07$ |
| ModelNet40 | $94.72 \pm 0.04$ | $0.73 \pm 0.12$ | $\mathbf{96.28} \pm 0.09$ | $0.63 \pm 0.14$ | $95.73 \pm 0.06$ | $-1.67 \pm 0.08$ |
| Cora-CA | $\mathbf{81.48} \pm 0.32$ | $18.27 \pm 0.31$ | $77.49 \pm 0.43$ | $21.99 \pm 0.45$ | $77.02 \pm 0.29$ | $22.69 \pm 0.27$ |

Second, datasets with strong homophily, such as Cora, Cora-CA, CiteSeer, and DBLP-CA, show significant performance drops after sparsification. Here, full connectivity provides a useful neighborhood signal, and removing hyperedges disrupts information flow.

# G  Empirical Analysis of Generalization and Noise Robustness

The main constrained objective in Section 3 is an empirical risk minimization problem under a mask budget. This does not by itself imply that sparsification always improves test performance. We therefore add two analyses that directly address the motivation: a train–test gap analysis on real datasets and a controlled noisy-hyperedge experiment where the corrupted structure is known.

**Train–test gap analysis.**  Table 15 reports test accuracy and the accuracy gap between train and test performance. The results support a conditional interpretation rather than a universal gap-reduction claim. On Actor and ModelNet40, learned sparsification improves test accuracy while keeping the gap comparable or smaller for EHGNN-F. On Pokec and Twitch, differences are modest and mixed. On Cora-CA, Full HGNN remains stronger and the gap increases under sparsification, consistent with the view that pruning can introduce harmful approximation bias when many homophilic relations are useful.

**Controlled noisy-hyperedge experiment.**  The train–test gap analysis cannot identify which relations are harmful. We therefore corrupt Cora by replacing a fraction $\eta$ of original hyperedges with mixed-label hyperedges. For each corrupted hyperedge, nodes are sampled from classes different from the original majority class, making the corrupted relation a task-adversarial message-passing channel. Table 16 shows that Full HGNN degrades as $\eta$ increases, while EHGNN-F and EHGNN-C are more robust at high noise rates. At $\eta = 0.4$, EHGNN-F and EHGNN-C achieve 69.31 and 69.78 test accuracy, respectively, compared with 64.93

Table 16: Controlled noisy-hyperedge experiment on Cora. Values are mean ± standard deviation over five runs. The corrupted-edge removal column is reported for EHGNN-C because it performs whole-hyperedge masking.

| Noise $\eta$ | Full test acc. | Full gap | EHGNN-F test acc. | EHGNN-F gap | EHGNN-C test acc. | EHGNN-C gap | EHGNN-C corrupt. removal |
|---|---|---|---|---|---|---|---|
| 0.0 | 77.46 ± 0.22 | 19.22 ± 0.24 | 75.66 ± 1.15 | 21.82 ± 1.16 | 75.36 ± 0.28 | 20.77 ± 0.92 | – |
| 0.1 | 74.12 ± 0.41 | 21.39 ± 0.41 | **74.30** ± 1.45 | 22.57 ± 1.40 | 73.91 ± 0.68 | 21.76 ± 0.62 | 0.632 ± 0.042 |
| 0.2 | 72.58 ± 0.47 | 21.60 ± 0.41 | **73.41** ± 1.02 | 22.94 ± 0.98 | 72.73 ± 0.50 | 23.37 ± 0.97 | 0.663 ± 0.023 |
| 0.3 | 68.33 ± 0.44 | 23.87 ± 0.35 | 71.05 ± 0.71 | 24.18 ± 0.94 | **71.43** ± 0.65 | 23.25 ± 1.36 | 0.620 ± 0.024 |
| 0.4 | 64.93 ± 0.52 | 27.40 ± 0.47 | 69.31 ± 0.83 | 25.92 ± 1.04 | **69.78** ± 0.80 | 25.73 ± 0.91 | 0.634 ± 0.031 |
| 0.5 | 65.02 ± 0.28 | 25.48 ± 0.30 | **69.96** ± 0.95 | 24.65 ± 1.24 | 68.42 ± 0.87 | 26.44 ± 1.01 | 0.606 ± 0.018 |

Table 17: Brier-loss decomposition for the high-noise Cora settings. Lower values are better. Bias and variance are empirical proxies computed from the five-run prediction ensemble and are not formal guarantees.

| Noise $\eta$ | Method | Brier error | Bias proxy | Variance proxy |
|---|---|---|---|---|
| 0.4 | Full | 0.4436 | 0.4423 | 0.0014 |
| 0.4 | EHGNN-F | 0.4122 | **0.3843** | 0.0279 |
| 0.4 | EHGNN-C | **0.4069** | 0.3934 | 0.0135 |
| 0.5 | Full | 0.4527 | 0.4514 | 0.0013 |
| 0.5 | EHGNN-F | **0.4072** | **0.3817** | 0.0255 |
| 0.5 | EHGNN-C | 0.4252 | 0.4121 | 0.0131 |

for Full HGNN. Since EHGNN-C masks whole hyperedges, we can also measure corrupted-edge removal recall. Uniform random selection at 50% retention removes roughly 50% of corrupted hyperedges, whereas EHGNN-C removes about 61–66% across nonzero noise rates, indicating that the learned mask preferentially suppresses corrupted structure.

**Brier-loss decomposition.** We additionally log train/validation/test losses and report a Brier-loss bias/variance-style decomposition. For a fixed split and $R$ random initializations or sampling seeds, let $\mathbf{p}_v^{(r)} \in \Delta^{C-1}$ be the predicted class-probability vector for test node $v$ in run $r$, let $\mathbf{y}_v \in \{0,1\}^C$ be its one-hot label vector, and let $\bar{\mathbf{p}}_v = \frac{1}{R} \sum_{r=1}^{R} \mathbf{p}_v^{(r)}$. Then

$$\frac{1}{R} \sum_{r=1}^{R} \|\mathbf{p}_v^{(r)} - \mathbf{y}_v\|_2^2 = \|\bar{\mathbf{p}}_v - \mathbf{y}_v\|_2^2 + \frac{1}{R} \sum_{r=1}^{R} \|\mathbf{p}_v^{(r)} - \bar{\mathbf{p}}_v\|_2^2.$$

Averaging over test nodes yields an empirical decomposition of test Brier error into a bias proxy and a variance proxy. We use this only as an empirical indicator, not as a formal guarantee. Across noise levels $\{0.4, 0.5\}$, we observe that learned sparsification improves high-noise Brier error mainly by reducing the dominant bias-proxy component; the variance proxy is not reduced.

# H  Low-Label and Dataset Audit Experiments

## H.1  Sensitivity to label scarcity

Because EdgeMask-HGNN learns the sparsifier from downstream supervision, we evaluated a low-label setting on representative datasets using 1%, 5%, 10%, and 20% labeled nodes for training, with 25% validation nodes and the remaining nodes used for testing. Table 18 reports mean test accuracy over five runs. The results show that label scarcity does not make the learned sparsifier collapse into random pruning. On Actor, EHGNN-F improves over Full by 10.25 points and over Random by 1.42 points even with only 1% labels, and it remains 10–13 points above Full across all four low-label settings. On ModelNet40, EHGNN-F improves over both Full and Random at every label ratio; its gains over Full are 0.66, 1.21, 1.25, and 1.34 points at 1%, 5%, 10%, and 20% labels, respectively, and its gains over Random are 0.50, 0.41, 0.25, and 0.24 points. On Cora, Full training remains strongest, but the best EdgeMask-HGNN variant improves over Random by 11.72, 5.34, 4.35, and 2.86 points at 1%, 5%, 10%, and 20% labels, respectively. Thus, low-label supervision does not invalidate the method or reduce it to random pruning. We still avoid claiming uniform dominance

Table 18: Low-label node classification results for HGNN with 50% retention. Values are mean test accuracy ± standard deviation over five runs. House is omitted from this table because the public preprocessing is analyzed separately in Appendix H.2.

| Dataset | Train labels | Full | Random | EHGNN-F | EHGNN-C |
|---------|-------------|------|--------|---------|---------|
| Actor | 1% | 58.80 ± 0.06 | 67.63 ± 0.50 | **69.05 ± 0.10** | 67.80 ± 0.30 |
| Actor | 5% | 62.32 ± 0.03 | 74.53 ± 0.55 | **75.70 ± 0.04** | 74.45 ± 0.10 |
| Actor | 10% | 63.36 ± 0.03 | 76.00 ± 0.37 | **76.45 ± 0.05** | 75.29 ± 0.12 |
| Actor | 20% | 64.20 ± 0.02 | 77.06 ± 0.21 | **77.07 ± 0.07** | 76.19 ± 0.04 |
| Cora | 1% | **47.61 ± 0.29** | 34.33 ± 5.34 | 45.70 ± 0.45 | 46.05 ± 0.21 |
| Cora | 5% | **64.82 ± 0.61** | 58.47 ± 1.16 | 61.88 ± 0.20 | 63.81 ± 0.38 |
| Cora | 10% | **71.77 ± 0.15** | 64.66 ± 0.68 | 67.93 ± 0.37 | 69.01 ± 0.25 |
| Cora | 20% | **75.99 ± 0.27** | 70.19 ± 0.94 | 73.05 ± 0.18 | 72.75 ± 0.38 |
| ModelNet40 | 1% | 85.92 ± 0.05 | 86.07 ± 0.95 | **86.57 ± 0.03** | 85.55 ± 0.08 |
| ModelNet40 | 5% | 93.59 ± 0.09 | 94.39 ± 0.08 | **94.80 ± 0.04** | 93.58 ± 0.14 |
| ModelNet40 | 10% | 94.15 ± 0.03 | 95.15 ± 0.16 | **95.40 ± 0.08** | 94.13 ± 0.17 |
| ModelNet40 | 20% | 94.45 ± 0.05 | 95.56 ± 0.22 | **95.79 ± 0.03** | 95.17 ± 0.06 |

Table 19: Comparison among models on House dataset with label-independent random 64-dimensional node features (50% train / 25% validation / 25% test split, and 50% retention). Values are mean test accuracy ± standard deviation over five runs.

| Method | Accuracy | Δ vs Full |
|--------|----------|-----------|
| Full | 47.31 ± 0.14 | – |
| Random | 50.03 ± 1.21 | +2.72 |
| EdgeDeg | 50.28 ± 1.32 | +2.97 |
| Spectral | 48.73 ± 0.71 | +1.42 |
| HSL | 49.16 ± 2.42 | +1.86 |
| HSL w/ aug. | 49.54 ± 4.15 | +2.23 |
| EHGNN-F | 49.35 ± 0.84 | +2.04 |
| EHGNN-C | 51.46 ± 0.26 | +4.15 |
| EHGNN-F (cond.) | 48.98 ± 1.01 | +1.67 |
| EHGNN-C (cond.) | 50.59 ± 0.47 | +3.28 |
| EHGNN-F (cond, LR) | **52.51 ± 0.56** | +5.20 |
| EHGNN-C (cond, LR) | 50.71 ± 0.89 | +3.41 |

over Full, since the Cora results show that Full training can remain strongest when the full hypergraph is already reliable.

## H.2 label-independent evaluation of House dataset

To test whether sparsification still has value without label-derived features, we evaluate House with label-independent random node features: node features $X \in \mathbb{R}^{1290 \times 64}$ are sampled from a standard normal distribution. Table 19 reports the comparison across the methods. We observe that, several sparsified variants modestly improve over Full training, and EHGNN-F(cond,LR) gives the best mean accuracy.

## I Ablation Studies

**Choice of sampler.** Table 20 compares the default weighted sampling without replacement strategy used in EHGNN-F against two alternatives: deterministic top-$k$ based on the scores $\boldsymbol{s}$ and independent Bernoulli sampling with $\ell_2$ regularization. The three variants are generally competitive, but EHGNN-F achieves

Table 20: Comparison (accuracy ± std) of EHGNN-F, deterministic top-k EHGNN-F, and EHGNN-F w/ Bernoulli sampling + L2 reg. across datasets.

| Model | 20news | ModelNet40 | Mushroom | NTU2012 | Actor | Citeseer | Cora-CA | DBLP-CA |
|---|---|---|---|---|---|---|---|---|
| EHGNN-F w/ deterministic top-k | 76.64 ± 0.05 | 96.23 ± 0.03 | **97.34 ± 0.33** | 86.08 ± 0.34 | 77.69 ± 0.05 | **68.91 ± 0.22** | 75.81 ± 0.51 | 86.03 ± 0.12 |
| EHGNN-F w/ Bern. sampling + $\ell_2$ | 77.04 ± 0.09 | 96.23 ± 0.11 | **97.34 ± 0.34** | 87.00 ± 0.67 | **77.99 ± 0.12** | 67.03 ± 0.53 | 77.02 ± 0.34 | 86.88 ± 0.10 |
| EHGNN-F | **77.18 ± 0.10** | **96.32 ± 0.11** | 97.24 ± 0.21 | **87.44 ± 0.36** | 77.90 ± 0.05 | 66.76 ± 0.45 | **77.61 ± 0.34** | **87.00 ± 0.10** |
| | Cora | House | Pokec | PubMed | Twitch | Walmart | Yelp | Trivago |
| EHGNN-F w/ deterministic top-k | **76.37 ± 0.18** | **86.01 ± 0.26** | **58.61 ± 0.04** | 86.66 ± 0.06 | 50.73 ± 0.03 | 96.92 ± 0.01 | 33.13 ± 0.16 | 45.22 ± 1.65 |
| EHGNN-F w/ Bern. sampling + $\ell_2$ | 75.66 ± 0.69 | 84.40 ± 1.09 | 58.52 ± 0.09 | **86.91 ± 0.13** | **50.94 ± 0.09** | 96.74 ± 0.08 | **33.33 ± 0.24** | 44.18 ± 1.90 |
| EHGNN-F | 75.72 ± 0.22 | 85.82 ± 0.96 | 58.56 ± 0.03 | 86.76 ± 0.05 | 50.82 ± 0.05 | **96.92 ± 0.02** | 33.29 ± 0.22 | **46.75 ± 0.71** |

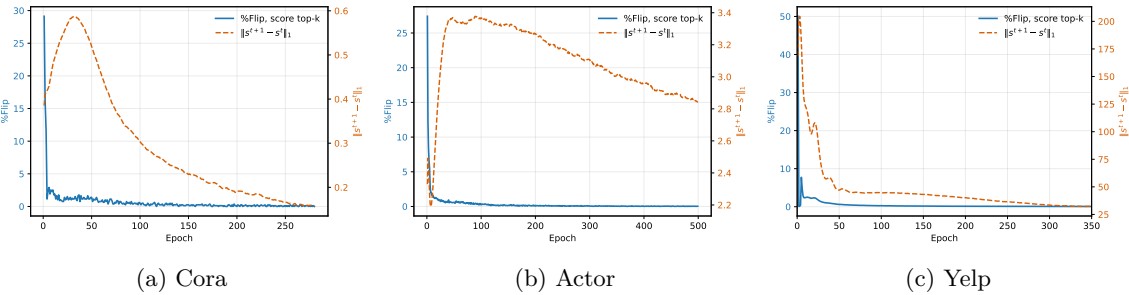

| (a) Cora | (b) Actor | (c) Yelp |
|---|---|---|

Figure 8: EHGNN-C mask stability at 50% retention. Blue curves show the percentage of hyperedge masks flipped between consecutive epochs, and orange curves show the normalized score-change term $\|\boldsymbol{s}_e^{(t)} - \boldsymbol{s}_e^{(t-1)}\|_1/|E|$ that appears in the stability bound.

the most consistent performance across datasets, supporting our choice of budget-constrained stochastic sampling.

## J Empirical Convergence and Stability of Learned Masks

A central concern in learnable sparsification is whether the discrete masks $\hat{m}_{v,e}$ stabilize during training. Here, "stability" refers to the sampler-level behavior analyzed in Section 4 and Appendix A; it is not a guarantee of optimization convergence or recovery of an optimal subhypergraph. Since the forward pass uses a hard budgeted selection and backpropagation uses a straight-through estimator (STE), unstable surrogate updates could otherwise lead to erratic mask behavior.

To measure empirical stability and connect the observed pattern to Theorems 5.1–5.2, we track both mask flips and the epoch-to-epoch score-change term appearing in the bound. For EHGNN-C, mask flips are hyperedges whose hard mask value changes between consecutive epochs:

$$\% \text{ Flip}^{(t)} = \frac{100}{|E|} \sum_{e \in E} \mathbf{1}\big[\hat{m}_e^{(t)} \neq \hat{m}_e^{(t-1)}\big],$$

where $\hat{m}_e^{(t)} \in \{0,1\}$ denotes whether hyperedge $e$ is kept by the sparsifier at epoch $t$. We also track the normalized score movement $\|\boldsymbol{s}_e^{(t)} - \boldsymbol{s}_e^{(t-1)}\|_1/|E|$, where $\boldsymbol{s}_e^{(t)}$ is the vector of EHGNN-C hyperedge scores at epoch $t$. If the learned sparsifier is unstable, we expect frequent flips throughout training (a higher %Flip).

Figures 8a, 8b, and 8c show the %Flip together with the normalized score-change term for three datasets (Cora, Actor, and Yelp). Across all three datasets, mask flips are concentrated in early epochs when score movement is largest, and the flip rate becomes negligible once score updates stabilize. This empirical pattern is consistent with the stability mechanism in Theorems 5.1–5.2; it is not intended as a tight numerical verification of the bound constants.

Table 21: Controlled Cora edge-corruption recovery evaluation at 50% retention. Accuracy and mask-quality metrics are percentages.

| Method | Acc. | Clean prec. ↑ | Removal recall ↑ | Retained corrupt. ↓ | Score AUPRC ↑ |
|---|---|---|---|---|---|
| Random | $82.27 \pm 0.90$ | $54.54 \pm 1.47$ | $50.49 \pm 0.62$ | $45.46 \pm 1.47$ | $55.15 \pm 2.95$ |
| Degree | $\mathbf{85.92} \pm 0.09$ | $54.37 \pm 2.92$ | $50.33 \pm 2.18$ | $45.63 \pm 2.92$ | $54.03 \pm 2.41$ |
| EHGNN-C | $82.52 \pm 1.15$ | $\mathbf{60.16} \pm 0.74$ | $\mathbf{56.59} \pm 1.52$ | $\mathbf{39.84} \pm 0.74$ | $\mathbf{62.99} \pm 1.87$ |

Table 22: Frozen-mask retraining evaluation. *Joint EdgeMask* is the original jointly trained EHGNN-F model; *Learned frozen* retrains a fresh HGNN on its deterministic learned subhypergraph.

| Method | Cora Acc. | Actor Acc. |
|---|---|---|
| Full HGNN | $\mathbf{77.99} \pm 0.30$ | $74.34 \pm 0.57$ |
| Random frozen | $75.18 \pm 0.89$ | $77.65 \pm 0.95$ |
| Degree frozen | $76.56 \pm 1.12$ | $76.47 \pm 0.96$ |
| Learned frozen | $76.17 \pm 0.67$ | $\mathbf{77.78} \pm 0.67$ |
| Joint EdgeMask | $76.51 \pm 1.03$ | $77.75 \pm 0.73$ |

## J.1 Mask Quality Evaluation

The stability result in Section 4 explains sampler-level behavior, but it does not by itself show that the selected subhypergraph is task-relevant or locally strong. We therefore add three mask-quality evaluations. These evaluations use the same 50% retention budget and report mean ± standard deviation over five seeds. They are not intended as global optimality or generalization guarantees.

**Controlled corruption recovery.** We first evaluate whether learned scores identify known corrupted structure. Starting from Cora, each hyperedge is assigned an adversarial class based on its majority label. Under edge-level corruption, with probability $p_{\mathrm{edge}} = 0.45$, all incidences in a hyperedge are replaced by nodes from the adversarial class. Since the corrupted hyperedges are known, we evaluate clean-retention precision, corruption-removal recall, retained-corruption rate, and score AUPRC for clean hyperedges. Table 21 shows that EHGNN-C gives the strongest mask-quality scores, even though Degree has the highest classification accuracy in this particular corrupted setting. Thus the learned coarse mask is not merely an arbitrary regularizer; it preferentially removes known corrupted hyperedges.

**Frozen-mask retraining.** To separate the selected structure from co-adaptation with the jointly trained HGNN weights, we train EHGNN-F, extract its deterministic top-$k$ mask, discard the trained HGNN weights, and retrain a fresh HGNN on the frozen learned subhypergraph. Table 22 shows a nuanced result. On Actor, the learned frozen mask remains competitive with the jointly trained model and improves over Full and Degree. On Cora, where Full training is already strong, the learned frozen mask remains competitive with other 50% frozen masks but does not outperform Full. This supports the claim that the learned subhypergraph can remain predictive after reinitialization, not that it always dominates other baselines.

**One-swap local optimality check.** Finally, we evaluate whether the learned deterministic mask admits many simple local improvements. For each trained model, we sample $Q = 1000$ one-swap neighbors by replacing one retained mask unit with one unretained unit, keep the trained HGNN weights fixed, and re-evaluate validation loss. Because many zero-threshold swaps lead only to numerically tiny changes, Table 23 reports swaps that reduce validation loss by more than $\epsilon = 10^{-3}$. For EHGNN-F, only 3.03% of sampled swaps improve validation loss by this margin; for EHGNN-C, the fraction is 8.07%. The Clopper–Pearson bounds, corresponding to Theorem 5.3 with $\delta = 0.05$, show that only a small fraction of one-swap neighbors are likely to provide such improvements. This does not establish global optimality, but it suggests that the learned masks have few obvious local replacements under the validation objective.

Table 23: One-swap local mask optimality check on Cora. We sample $Q = 1000$ one-swap mask replacements and report the fraction that reduce validation loss by more than $\epsilon = 10^{-3}$, together with the 95% Clopper–Pearson upper confidence bound on that fraction.

| Model | Improving swaps ↓ | 95% upper bound ↓ | Best loss improvement |
|---|---|---|---|
| EHGNN-F | $3.03 \pm 0.67\%$ | $4.08 \pm 0.76\%$ | $0.0037 \pm 0.0008$ |
| EHGNN-C | $8.07 \pm 1.93\%$ | $9.62 \pm 2.08\%$ | $0.0043 \pm 0.0002$ |

