# OpenReview forum: "EdgeMask-HGNN: Learning to Sparsify Hypergraphs in Hypergraph Neural Networks"
_TMLR — Decision pending for TMLR_

### Review · Reviewer_tSxH · 2026-04-15

**Summary Of Contributions:**

This paper presents a method of learnable sparsification to select task-relevant hyperedges for hypergraph neural networks. It argues that not all the hyperedges are useful for the downstream task. Thus, masking some edges improves the performance. The manuscript includes two masking strategies and scalable variants for both of them.


## Strengths

1. The proposed methodology can be trivially adapted to many hypergraph models.
2. The sparsification's stability is theoretically guaranteed.
3. The sparcification improves training time.
4. Several variants of the model presented with scalability considerations.


## Weaknesses

1. The paper is missing evidences that hypergraph sparcification introduces fundamentally new challenges beyond graphs.
2. Weak empirical evaluation results, including baseline modifications, inconsistent performance, lack of statistical validation
3. The proposed methodology demonstrate reduced effectiveness on homophilic hypergraphs.

**Audience:**

Yes

**Audience Explanation:**

The topic of hyper-graph sparcification will surely be interesting for the graph learning community.

**Broader Impact Concerns:**

No ethical concerns. No need for the Broader Impact Statement.

**Claims And Evidence:**

No

**Claims Explanation:**

TL;DR: The presented results are not fully clear or convincing.

The paper presents 3 claims:

1. $\text{&#x2753;}$ "Unlike unsupervised methods (e.g., random, degree-based, and spectral sparsification) that rely on fixed hypergraph structures sampled a priori, EdgeMask-HGNN dynamically selects a subset of hyperedges during training that helps the downstream HGNN perform better on the learning task." I disagree with the usage of "dynamically selects". From the text, the mask seems to be learned once and then fixed. The learned mask is static. No dynamic selection happening during the inference. The results in section 6 do not unambiguously show that using EdgeMask is better for the downstream performance or that it is better than a random mask for node tasks. However, it does improve the downstream performance for link prediction.

2. $\text{&#x2705;}$ "We theoretically analyze the stability of the learned sparsifier. To be precise, we show that as the retention logits evolve across epochs, the expected number of changes in the sampled masks is bounded by the magnitude of logit updates."
The theory is presented in Section 5 and extended in Appendix A.

3. $\text{&#x2753;}$ Empirical evaluation and analysis show that EdgeMask-HGNN is more effective than alternative sparsifiers such as random, degree-based, and spectral sparsifiers, as well as full hypergraph training on heterophilic hypergraphs. On more homophilic datasets, its performance is often comparable to strong baselines."
The results in section 6 do not show unambiguously that using EdgeMask is better for the downstream performance or that it is better than a random mask for node tasks. It does perform well for link prediction tasks.

**Requested Changes:**

## Critical

The manuscript is missing a discussion with formal or empirical arguments on why sparsifying a hypergraph is a fundamentally new challenge and "extending
these ideas [supervised sparsification for graphs] to hypergraphs is nontrivial". Argumentation on Page 3 is slightly handwavy.
Could you clarify what are the exact fundamental issues that arise when one attempts to extend supervised sparsification from graphs to hypergraphs?

The proposed sparsification approach is relying on a fixed sparcification hyperparameter. This is an additional hyperparameter that requires setting. The manuscript mentions that for some homophilic datasets a higher budget is needed to achieve better performance. However, there are no guidelines to selecting this budget for a given dataset. Moreover, almost all the experiments are conducted at a sparsification rate of k = 50%. Why was this rate selected for the given datasets?
Did you explore any adaptive procedures to set sparcification budget automatically?

Theorems 5.1 and 5.2 bound the "expected number of mask flips between epochs". While this may be an interesting theoretical result, the manuscript is not discussing why it is relevant. The stability does not seem to directly translate to convergence. What is the theoretical connection between this results and sparsification training convergence?
Can you specify how bounded mask stability translates to an improvement in the model's performance?

For the baseline choice, I cannot understand the decision behind removing augmentation for HSL baseline. This decision intentionally weakens the baseline for no apparent solid reason.
Why is this considered a fair comparison? Can you provide results with included augmentation?

Table 2. Is confidence interval a standard deviation? Are the presented results statistically significant comparing to the best baseline? Could you perform a t-test or Wilcoxon test on the runs? Without statistical testing results, your method clearly outperforms the random baseline in only 3 out of 16 cases (NTU2012, Citeseer, House).

Table 2. In 6 cases out of 16 the proposed model performs worse than the full model. In three additional cases, the sparse graph performance falls within the confidence interval of the full model. That undermines the original motivation that "pruning irrelevant pairs enhances the performance".
Are the results statistically significant comparing to the full model?

Table 3. What task this table is referring to? Node classification? What dataset does it use? How exactly the correlation was computed?

Table 4. Could you discuss why low-rank decomposition introduces such a performance boost? No need for new experiments, just add your best hypothesis that can explain the result.

Page 12. The presented model "relies on supervised signals and may not
generalize as effectively in limited-label settings or fully unsupervised settings." Could you explain why you arrived at this conclusion? This limitation does not directly follow from the methodology where your method is an end-to-end preprocessing step for a downstream task. Could you demonstrate this limitation? Possibly by making an experiment in an unsupervised setting (e.g., node clustering)?

Could you make a graph classification experiment to further support the point that your approach can be utilized with any downstream task?



## Minor

In the problem statement, I recommend adding a sentence that a hyperedge is defined as a subset of nodes. For readability. Also, I suggest replacing in the text "#incidence pairs" with "number of incidence pairs" and "#hyperedges" with "number of hyperedges".

While the overall masking for hypergraphs is original, the proposed limiting of a parameter with sigmoid in equation 1 has already been introduced in the literature. I believe one of the first cases was presented by Trask et al. 2018. Please add an appropriate citation to equation 1. You may either use this citation or any suitable paper citation from the sparsification domain.

Page 7. "The proofs are in the Appendix." It would improve the quality if, in this and other cases, the appendix was specified with a reference.

Table 4. Are you following the same experiment protocol as previous tables? Why don't you report the confidence intervals here? While I can see that for this task the performance gap is considerable, please make the result presentation consistent.

Tables 2,3,4. Generally it is hard to understand what experiment a table is referring to from the caption. Please ensure the captions clearly present the main details without requiring reference to the text.


## Literature

1. Trask, Andrew, et al. "Neural arithmetic logic units." NeurIPS 2018.

---

> ### Author Response · Authors · 2026-05-20
> **Author response (I)**
>
> Thank you for your constructive comments. In the following we address the changes suggested.
>
> **On Weaknesses / Unconvincing Claims**
>
> > "I disagree with the usage of "dynamically selects". From the text, the mask seems to be learned once and then fixed. The learned mask is static. No dynamic selection happening during the inference."
> - We have revised that line in the introduction.
>
> > "The results in section 6 do not show unambiguously that using EdgeMask is better for the downstream performance or that it is better than a random mask for node tasks."
> - We have carefully revised the manuscript to avoid that claim. The claim is that **EdgeMask-HGNN is not always better than Full or Random, but it is better in settings where task-aware selection is useful**. For instance, in Section 6, Table 2, EdgeMask variants improve over Full on several benchmarks, and Appendix F, Table 14 reports the paired tests. In addition, Appendix H.1, Table 18 shows that EHGNN-F performs better than both Full and Random in low-label regimes on Actor and ModelNet40. Finally, Appendix G, Table 16 shows that under controlled noisy-hyperedge corruption, learned masks outperform Full and remove corrupted hyperedges at a higher rate than random pruning. Thus our claim is not that EdgeMask is uniformly superior, but that learned budgeted sparsification provides a task-aware alternative to random pruning and is beneficial under identifiable structural/noise regimes.
>
> **On Recommended Changes**
>
> > 1. Fundamental challenges to hypergraph sparsification
>
> We added a more concrete discussion of why supervised hypergraph sparsification is not a direct extension of supervised graph sparsification (Section 2). The main issue is not only that hyperedges have cardinality larger than two; the optimization object itself changes.
>
> In a graph, pruning an edge removes one pairwise relation. In a hypergraph, there are at least two natural but different maskable units: whole hyperedges and individual node-hyperedge incidences. These choices are not interchangeable. Hyperedge pruning deletes an entire higher-order relation, whereas incidence pruning edits the membership of a relation that may still remain in the hypergraph. As a result, the two operations change hyperedge cardinalities, node degrees, and the two-stage node-to-hyperedge-to-node normalization in different ways.
>
> We also added a formal clique-expansion argument that explains why this becomes a modeling challenge. For a size-$r$ hyperedge, clique expansion creates $\binom{r}{2}$ pairwise graph edges. Deleting the hyperedge corresponds to deleting all $\binom{r}{2}$ clique-expanded edges. Deleting one incidence corresponds to deleting only the $r-1$ pairwise edges involving the removed node while retaining the pairwise edges among the remaining members. A generic graph-edge mask on the clique expansion permits many other pairwise subsets that correspond to neither a valid hyperedge deletion nor a valid incidence edit. Therefore a graph sparsifier would need additional consistency constraints to ensure that its pairwise pruning pattern maps back to a coherent hypergraph operation. Without those constraints, it may retain arbitrary fragments of a higher-order relation that no longer correspond to any masked incidence matrix. This is why clique-expanded graph pruning is not an adequate substitute for directly learning hyperedge- or incidence-level masks (Section 2).

---

> ### Author Response · Authors · 2026-05-20
> **Author response (II)**
>
> > 2. Fixed sparsification budget and budget selection
>
> We now answer both budget questions explicitly (Section 6, Table 3; Figure 5). The 50% retention rate was selected as a common controlled compression point, not because it is optimal for every dataset. Using the same nominal budget lets us compare learned masks with Random, EdgeDeg, Spectral, and HSL-style sparsifiers without confounding the comparison by giving one method a larger retained subhypergraph. For budget choice in practice, we added a validation-based selection procedure over $\\{0.3, 0.5, 0.7, 0.9\\}$. We did not propose or evaluate an automatic adaptive budget controller in this revision; instead, the recommended practical procedure is to select both the budget and the variant by validation performance. We also added a concrete future-work path: replace the hard budget with a Lagrangian $L\_0$-style penalty, e.g., $\mathcal{L}\_{task}+\lambda\|M\|\_0$ or $\mathcal{L}\_{task}+\lambda\|m\|\_0$, using differentiable hard-concrete style gates, so that the budget can be learned jointly. We also added a practical variant-selection table based on incidence scale $t$ and point to the retention sweep showing that homophilic datasets often need higher retention (Figure 5).
>
> > 3. Relevance of Theorems 5.1 and 5.2
>
> We agree the previous wording over-positioned the theorem. The revised version states that the theorem explains sampler stability only: once logits change slowly, the expected number of mask flips becomes small (Section 5). It does not imply performance improvement, convergence of training, or recovery of an optimal mask. To connect the theorem with the empirical figure, we revised Figure 8 to overlay \%Flip with $\|\mathbf{s}^{t+1}-\mathbf{s}^{t}\|_1$; the plot shows that flips concentrate when score movement is large early in training and become negligible as score updates stabilize (Section 5 and Appendix J, Figure 8).
>
> > 4. HSL augmentation
>
> We clarified that our main HSL comparison is a pruning-oriented HSL adaptation over observed incidences (Section 6, Table 2). We also implemented the original residual-incidence augmentation step and report it as `HSL w/ aug.` in the accuracy and memory tables (Section 6, Table 2; Appendix C, Table 9).
>
> > 5-6. Statistical significance and comparison to Full/Random
>
> We added paired tests versus Full for EHGNN-F and EHGNN-C and softened claims where gains are small or where Full remains better (Appendix F, Table 14). We no longer claim universal improvement over Full or Random (Abstract, Section 6, Table 2; Section 7).
>
> > 7. Table/caption clarity
>
> We revised the current Table 5 caption to make the link-prediction protocol clearer, including the 50% retention setting, the reported metrics, and the appendix location of the multi-seed mean $\pm$ standard-deviation contextualization ablations. We also replaced visible # shorthand with clearer wording in the relevant text, added the requested hyperedge definition in the problem statement, cited bounded sigmoid gates near the first sigmoid mask equation, and replaced generic proof pointers with explicit appendix references (Sections 3-6 and Appendix A).
>
> > 8. Performance boost in low-rank variants
>
> We added this explanation in Section 4.3, in the paragraph immediately after the EHGNN-F(cond,LR) low-rank score definition. The revised text states that the low-rank scorer can regularize mask learning because it ties incidence scores through shared node and hyperedge factors. A more flexible MLP scorer can assign highly specific scores to many individual incidences, which can fit incidental or noisy scoring patterns when supervision is limited relative to the number of incidences. The low-rank form reduces the effective number of independent scoring patterns because incidences that share a node or hyperedge also share part of the scorer. This coupling can improve generalization of the learned mask and helps explain why the low-rank conditioned variant sometimes outperforms the more expressive MLP scorer.

---

> ### Author Response · Authors · 2026-05-20
> **Author response (III)**
>
> > 9. Limited-label setting
>
> We revised the limitation paragraph that discusses limited-label and fully unsupervised settings. EdgeMask-HGNN learns its mask from the downstream supervised loss, so the amount and type of supervision directly affect the mask-learning signal. We therefore added low-label experiments in Appendix H.1, Table 18. On Actor and ModelNet40, EHGNN-F outperforms both Full and Random at every label ratio, including the 1% setting. On Cora, Full training remains strongest, but both EHGNN-F and EHGNN-C stay consistently above Random at the same 50% retention budget. Thus, the limited-label evidence is more nuanced than the original wording: with very few labels, EdgeMask-HGNN does not always outperform Full training, but it still learns masks that are stronger than random pruning.
>
> Representative 1% label results from Appendix H.1, Table 18 are presented below:
>
> | Dataset | Full | Random | EHGNN-F | EHGNN-C |
> |---|---:|---:|---:|---:|
> | Actor | 58.80 ± 0.06 | 67.63 ± 0.50 | **69.05 ± 0.10** | 67.80 ± 0.30 |
> | Cora | **47.61 ± 0.29** | 34.33 ± 5.34 | 45.70 ± 0.45 | 46.05 ± 0.21 |
> | ModelNet40 | 85.92 ± 0.05 | 86.07 ± 0.95 | **86.57 ± 0.03** | 85.55 ± 0.08 |
>
> For fully unsupervised settings such as node clustering, the supervised task loss used by EdgeMask-HGNN is unavailable. Thus, task loss would have to be replaced by a proxy objective, such as incidence reconstruction or contrastive learning. Since the learned mask would then be optimized for that proxy rather than for labels, we treat unsupervised EdgeMask-HGNN as a separate extension to our work (Section 7).
>
> > 10. Graph classification
>
> We appreciate the suggestion. We did not add graph classification because it is outside the scope of the claims made in this paper.  EdgeMask-HGNN assumes that the input is already a hypergraph. For a graph-classification benchmark, we would first need to decide how to convert each graph into a hypergraph and how to pool node representations into a graph-level prediction. The result would therefore depend heavily on those extra design choices, while making it harder to isolate the effectiveness of sparsification.
>
> ---
> **Please let us know if you have any further questions/comments**.

---

> > ### Comment · Reviewer_tSxH · 2026-06-06
> > **Response to Authors**
> >
> > Thank you for your updated work. A lot of my comments are addressed.
> > I apologize for the late response.
> >
> > ### Updated claims
> >
> > 1. $\text{&#x2705;}$ "Figure 1 shows that under the same sparsification budget, embeddings learned by EdgeMask-HGNN yield better class separation than the random
> > sparsifier."
> > Yes, it is shown on the figure 1.
> >
> >
> > 2. $\text{&#x2705;}$ "We theoretically analyze the stability of the learned sparsifier. To be precise, we show that as the
> > retention logits evolve across epochs; the expected number of changes in the sampled masks is bounded by
> > the magnitude of logit updates. This result characterizes the stability of the sampling process." This theory is presented in the section 5.
> >
> > 3. $\text{&#x274C;}$ "Empirical evaluation and analysis show that EdgeMask-HGNN is more effective than alternative sparsifiers such as random, degree-based, and spectral sparsifiers, as well as full hypergraph training on heterophilic
> > hypergraphs. On more homophilic datasets, its performance is often comparable to strong baselines." Presented results are not conclusive for this claim. Moreover, EdgeMask-HGNN is not better than full training according to Appendix F.
> >
> > 4. $\text{&#x2705;}$ "For link prediction, EdgeMask-HGNN enabled contextual aggregation improves over plain Hyper-SAGNN and
> > random masked context; comparisons with a full-context control show that the best choice between full and
> > learned budgeted context is dataset-dependent." Section 6.1 provides experimental results.
> >
> > 5. $\text{&#x2753;}$ "EdgeMask-HGNN is simple by design and easily adaptable to different HGNN backbones." While section 4 does not contradict this claim, there is not supporting evidence presented.
> >
> >
> >
> >
> > ### Some comments
> > On page 4. Since the masking is meant to remove the relationship between nodes, the third option could be subdividing the hyperedge. For regular graphs, the edge connects only two nodes. Thus, such masking is not applicable, but for the hypergraphs, it is possible.
> >
> > The Section 5 is still largely irrelevant for the manuscript. Local optimality is a weak property.
> >
> > There are still no statistical tests to show if the results are significant. For link predictions the gap is clear, but for node-prediction your model does not show any clear advantage.
> > To put it bluntly, the paper still does not answer, "Why should I use this model for node-classification instead of a random sparcification?"
> >
> > There is an Appendix F exploring pairwise comparisons of Full vs. EHGNN-C and Full vs. EHGNN-F. This is a good result. Why no comparison with baselines? One of your claims on updated page 2 is "EdgeMask-HGNN is more effective than alternative sparsifiers such as random, degree-based, and spectral sparsifiers, as well as full hypergraph training on heterophilic
> > hypergraphs." So, are the results from Table 2 statistically significant, comparing to the baselines?
> >
> >
> > You should not write "Our limited-label experiments show..." in the conclusion, because the main body introduces no experiments for limited-label setting.
> > This reads as incorrect, since one reads the paper from top to bottom.
> >
> > **Graph classification** You understood me wrong. EdgeMask-HGNN assumes that the input is a hypergraph. So a graph-classification should look like this: hypergraph on the input, class label on the output. No need to convert a regular graph into a hypergraph. The pooling could be any of your choice.
> > However, there is no need for those additional experiments.

---

### Review · Reviewer_EYAy · 2026-04-24

**Summary Of Contributions:**

## Summary

The paper proposes a learnable hypergraph sparsifier that keeps a fixed
fraction of hyperedges (EHGNN-C) or of individual node–hyperedge links
(EHGNN-F). Scores are assigned to candidates (free logits or predicted
from features), and a random subset of the target size is drawn with
probability proportional to the scores. Gradients pass through the
discrete sampling via a straight-through estimator. Two theorems bound
the expected epoch-to-epoch change in the sampled subset. Experiments
cover node classification on 16 benchmarks (including the heterophilic
datasets Actor, Twitch, Pokec) at 50% sparsity, plus link prediction.

## Main weakness: the problem formulation is incomplete

Section 3 frames the contribution as "budget-constrained task-aware
sparsification" but the optimization problem actually written is

        θ* = arg min_θ  L_task(f_θ(s; H̃, X), y_L),

with no budget constraint, no sparsifier variable, and no joint
minimization. H̃ is described only as "obtained from a learnable
sparsifier" and never enters the arg min. This is ordinary empirical
risk minimization with a fixed masked incidence matrix — the defining
aspect of the paper (the budget, the discreteness, the joint
learning) is absent from the formal statement.

The correct formulation would be a constrained joint optimization,
e.g.

        (θ*, M̂*) = arg min_{θ, M̂}  L_task(f_θ(s; H ⊙ M̂, X), y_L)
        subject to   ‖M̂‖_0 ≤ k   (or  Σ_e m̂_e ≤ κ).

This single missing sentence cascades through the rest of the paper:

- The relationship between EHGNN-F, EHGNN-C, random, and spectral
  baselines becomes unclear, because all of them are then just
  different approximate solvers for the same constrained program.
- The theorems are about stability of a specific sampler, not about
  the optimization problem being solved. Without the optimization
  problem written down, the reader cannot judge whether stability of
  the sampler is the right quantity to bound.
- The connection to the large continuous-weight / graph-signal-
  processing (GSP) literature on graph learning is invisible.
  Learning the structure of a graph (or hypergraph) from data under
  smoothness or sparsity priors has been studied for over a decade.
  None of the following is cited:
    - V. Kalofolias, "How to Learn a Graph from Smooth Signals,"
      AISTATS 2016, PMLR v51:920–929.
    - X. Dong, D. Thanou, P. Frossard & P. Vandergheynst, "Learning
      Laplacian Matrix in Smooth Graph Signal Representations,"
      IEEE Trans. Signal Process., vol. 64, no. 23, pp. 6160–6173,
      2016. doi:10.1109/TSP.2016.2602809
    - S. Segarra, A. G. Marques, G. Mateos & A. Ribeiro, "Network
      Topology Inference from Spectral Templates," IEEE Trans.
      Signal Inf. Process. over Netw., vol. 3, no. 3, pp. 467–483,
      2017. doi:10.1109/TSIPN.2017.2731051
    - G. Mateos, S. Segarra, A. G. Marques & A. Ribeiro,
      "Connecting the Dots: Identifying Network Structure via
      Graph Signal Processing," IEEE Signal Process. Mag., vol. 36,
      no. 3, pp. 16–43, 2019. doi:10.1109/MSP.2018.2890143
    - X. Dong, D. Thanou, M. Rabbat & P. Frossard, "Learning Graphs
      From Data: A Signal Representation Perspective," IEEE Signal
      Process. Mag., vol. 36, no. 3, pp. 44–63, 2019.
      doi:10.1109/MSP.2018.2887284
  In particular, the GTVMin (generalized total variation
  minimization) view connects network connectivity directly to
  estimation error:
    - Y. SarcheshmehPour, Y. Tian, L. Zhang & A. Jung, "Clustered
      Federated Learning via Generalized Total Variation
      Minimization," IEEE Trans. Signal Process., vol. 71,
      pp. 4240–4256, 2023. doi:10.1109/TSP.2023.3322848
  shows that estimation error under GTVMin scales with a
  connectivity-based quantity of the underlying graph. This
  suggests a principled, network-driven alternative to learning
  retention logits by supervision: sample or retain hyperedges
  according to structural statistics tied to estimation error,
  e.g., a hypergraph extension of the clustering coefficient, or a
  Laplacian / effective-resistance score on the hypergraph itself.
- The motivational premise ("unnecessary hyperedges hurt accuracy")
  is never derived from the formulation. I cannot find a
  bias–variance decomposition or a train-vs-test gap measurement
  anywhere in the paper.

It would be helpful to write the constrained joint optimization explicitly
in Section 3, derive (or at least sketch) why the minimizer should prefer
task-aligned hyperedges, and position every baseline and every
EHGNN variant as an approximate solver of that program.

## Other weaknesses

- Empirical margins are small. On Twitch and Pokec, the best EHGNN
  variant improves the best baseline within standard deviations.
  Full training beats every EHGNN variant on 7 of 16 datasets
  (Cora, 20news, Mushroom, Cora-CA, DBLP-CA, Yelp, Trivago).
- Performance on House (100.00 ± 0.00 for EHGNN-F(cond)) is
  unexplained and might be due to a label-leakage / evaluation bug.
- Unclear how Theorems 5.1–5.2 are useful for convergence, recovery,
  or generalization guarantees.
- Theory and experiments are not well connected: Fig. 8 tracks %Flip
  over epochs but never overlays ‖s^{t+1} − s^{t}‖_1, so the bound
  is not actually tested.
- Motivational claims appear without citation or measurement:
  "HGNNs prone to overfitting", "real-world hypergraphs are often
  noisy and densely connected", "inevitably contain task-irrelevant
  or even task-adversarial higher-order relations".

**Audience:**

Yes

**Audience Explanation:**

Yes, budget-constrained supervised sparsification of hypergraphs is somewhat
underexplored.

**Claims And Evidence:**

No

**Claims Explanation:**

C1 (effective on heterophilic hypergraphs): supported only on
  Actor. Twitch / Pokec gains are within noise.
- C2 (comparable on homophilic datasets): supported but weak (ties
  with Random on many datasets; Full wins on 7 of 16).
- C3 (superior link prediction): in the appendix, not verifiable
  from the main text.
- C4 (theoretical stability): derivations are correct but the
  contribution is overstated and disconnected from the experiments.

**Requested Changes:**

address above concerns

---

> ### Author Response · Authors · 2026-05-20
> **Response (part 1)**
>
> Thank you for your time and comments. In the following, we address the issues that were highlighted.
>
> > Incomplete problem formulation
>
> Thank you for highlighting this issue. In the revision, we rewrote the problem statement as a constrained joint optimization and connected it directly to the method and baselines (Section 3).
>
> For incidence-level sparsification, the revision now states:
>
> `(theta*, M*) = argmin_{theta,M} L_task(f_theta(s; H ⊙ M, X), y_L)`
>
> subject to `M_{v,e}=0` whenever `H_{v,e}=0` and `||M||_0 <= k`. The support constraint makes the incidence budget count retained observed node-hyperedge incidences, and the notation now parallels the hyperedge-level constraint `||m||_0 <= kappa`.
>
> For hyperedge-level sparsification, it analogously optimizes over an edge mask with `||m||_0 <= kappa`.
>
> We also added an intuition for why the supervised objective can favor task-aligned relations: under a fixed budget, retaining an incidence or hyperedge is useful only insofar as it reduces the downstream supervised loss, and the straight-through estimator routes this task loss back to the corresponding mask scores. This is an optimization intuition, not a guarantee of test improvement (Section 3).
>
> Finally, we now position each method relative to this program. EHGNN-F and its conditioned variants approximate the incidence-level program, EHGNN-C and its conditioned variants approximate the hyperedge-level program, Random/EdgeDeg/Spectral are task-agnostic feasible-mask comparators, whereas the HSL is a task-aware pruning comparator over observed incidences. We report `HSL w/ aug.` separately as it changes the feasible set by adding incidences (Sections 3 and 6).
>
> > Missing graph signal processing / graph-learning literature
>
> We added the suggested GSP graph-learning references and discussed them in Section 2. These methods typically infer weighted graph structure under smoothness, sparsity, or spectral constraints. In contrast, budgeted hypergraph sparsification starts from an observed hypergraph and selects a discrete budget-feasible subhypergraph using downstream supervision. Nevertheless, GSP-style structural priors could be a useful future direction for hypergraph sparsification.
>
> > Theoretical analysis
>
> We revised the theory discussion to say exactly what is proved: bounded epoch-to-epoch mask flips under bounded logit changes (Section 5). We removed implications that the theorem establishes task-level convergence, optimality, recovery, or generalization. We also revised Figure 8 to overlay the mask flip rate with the score-change term $\|\mathbf{s}^{t+1}-\mathbf{s}^{t}\|_1$ used in the bound, so the empirical pattern now directly reflects the theorem's stability mechanism while avoiding any claim of tight numerical verification (Appendix J, Figure 8).

---

> ### Author Response · Authors · 2026-05-20
> **Response (part 2)**
>
> > Motivational claims and empirical margins
>
> Thank you for pointing this out. We agree that the original motivation was too broad. We revised it to a conditional claim: **sparsification can help when removing noisy or task-irrelevant relations reduces harmful message passing more than it increases approximation bias from discarding useful relations.** Conversely, in highly homophilic settings, pruning can remove weak but collectively useful relations and hurt performance.
>
> To support our claim, we added a bias/generalization tradeoff discussion after the constrained formulation (Section 3), a controlled noisy-hyperedge experiment where corrupted hyperedges are known (Appendix G, Table 16), train-vs-test accuracy-gap analysis (Appendix G, Table 15), and a Brier-loss decomposition (Appendix G, Table 17).
>
> **Controlled noisy-hyperedge experiment.** We corrupt Cora by replacing a fraction $\eta$ of original hyperedges with mixed-label hyperedges, making those relations task-adversarial for message passing. Values are mean ± std over five runs. The corrupted-edge removal column is reported for EHGNN-C because it performs whole-hyperedge masking.
>
> | Noise | Full test acc | Full acc gap | EHGNN-F test acc | EHGNN-F acc gap | EHGNN-C test acc | EHGNN-C acc gap | EHGNN-C corrupted-edge removal recall |
> |---:|---:|---:|---:|---:|---:|---:|---:|
> | 0.0 | 77.46 ± 0.22 | 19.22 ± 0.24 | 75.66 ± 1.15 | 21.82 ± 1.16 | 75.36 ± 0.28 | 20.77 ± 0.92 | - |
> | 0.1 | 74.12 ± 0.41 | 21.39 ± 0.41 | **74.30 ± 1.45** | 22.57 ± 1.40 | 73.91 ± 0.68 | 21.76 ± 0.62 | 0.632 ± 0.042 |
> | 0.2 | 72.58 ± 0.47 | 21.60 ± 0.41 | **73.41 ± 1.02** | 22.94 ± 0.98 | 72.73 ± 0.50 | 23.37 ± 0.97 | 0.663 ± 0.023 |
> | 0.3 | 68.33 ± 0.44 | 23.87 ± 0.35 | 71.05 ± 0.71 | 24.18 ± 0.94 | **71.43 ± 0.65** | 23.25 ± 1.36 | 0.620 ± 0.024 |
> | 0.4 | 64.93 ± 0.52 | 27.40 ± 0.47 | 69.31 ± 0.83 | 25.92 ± 1.04 | **69.78 ± 0.80** | 25.73 ± 0.91 | 0.634 ± 0.031 |
> | 0.5 | 65.02 ± 0.28 | 25.48 ± 0.30 | **69.96 ± 0.95** | 24.65 ± 1.24 | 68.42 ± 0.87 | 26.44 ± 1.01 | 0.606 ± 0.018 |
>
> As $\eta$ increases, Full HGNN's test accuracy drops from 77.46 to about 65, while EHGNN-F and EHGNN-C are more robust at high noise rates. At 50% retention, uniform random removal removes about 49-51% of corrupted hyperedges; EHGNN-C removes about 61-66% across nonzero noise rates. Thus, the learned mask preferentially suppresses corrupted structure rather than pruning uniformly.
>
> **Train-test gap analysis.** In the following table, we present the train-test gap analysis. Acc gap is train accuracy minus test accuracy. Negative gaps indicate cases where the averaged test accuracy is slightly higher than the train accuracy.
>
> | Dataset | Full test acc | Full acc gap | EHGNN-F test acc | EHGNN-F acc gap | EHGNN-C test acc | EHGNN-C acc gap |
> |---|---:|---:|---:|---:|---:|---:|
> | Actor | 64.54 ± 0.03 | 0.96 ± 0.03 | **77.96 ± 0.05** | 0.85 ± 0.06 | 76.72 ± 0.09 | 1.22 ± 0.43 |
> | Twitch | 51.22 ± 0.04 | 0.89 ± 0.05 | 50.81 ± 0.06 | 1.47 ± 0.08 | **51.44 ± 0.07** | -1.88 ± 0.14 |
> | Pokec | 58.36 ± 0.03 | 1.45 ± 0.03 | 58.54 ± 0.04 | 1.79 ± 0.03 | **59.15 ± 0.05** | -0.23 ± 0.07 |
> | Walmart | 95.36 ± 0.01 | -0.02 ± 0.02 | **96.93 ± 0.03** | 0.05 ± 0.03 | 95.89 ± 0.07 | 0.03 ± 0.07 |
> | ModelNet40 | 94.72 ± 0.04 | 0.73 ± 0.12 | **96.28 ± 0.09** | 0.63 ± 0.14 | 95.73 ± 0.06 | -1.67 ± 0.08 |
> | Cora-CA | **81.48 ± 0.32** | 18.27 ± 0.31 | 77.49 ± 0.43 | 21.99 ± 0.45 | 77.02 ± 0.29 | 22.69 ± 0.27 |
>
> This table is not meant to show universal gap reduction. On Actor and ModelNet40, learned sparsification improves test accuracy while keeping the gap comparable or smaller for EHGNN-F. On Pokec and Twitch, differences are modest and mixed. On Cora-CA, Full HGNN remains stronger, and the gap increases under sparsification, illustrating the limitation that pruning can introduce bias when many homophilic relations are useful.
>
> **Brier decomposition and takeaway.** For a fixed train/validation/test split and multiple random initialization/sampling seeds, Appendix G, Table 17 reports a Brier-loss bias/variance-style decomposition. We use it only as an empirical decomposition, not as a formal guarantee. In the noisy-hyperedge grid, learned sparsification improves high-noise Brier error mainly by reducing the dominant bias-proxy component; the variance proxy is not uniformly smaller. Thus, our claim is narrower: **task-aware sparsification is useful when the benefit from suppressing noisy or task-irrelevant relations outweighs the approximation bias introduced by pruning.**

---

> ### Author Response · Authors · 2026-05-20
> **Response (part 3)**
>
> > House dataset
>
> Thank you for pointing out this issue. We traced the issue to the House preprocessing used in the public Cornell/AllSet benchmark [1]. In that preprocessing, node labels are converted into one-hot node features and then perturbed with Gaussian noise controlled by feature noise. Our submitted House runs used feature noise=0, and as House has two classes, each node feature is exactly a two-dimensional one-hot encoding of its class. As a result, the class label can be recovered directly from the largest feature entry for every House node. The near-perfect original House accuracy was therefore due to label leakage through the node features.
>
> In the revision, we replaced the original House results in the main accuracy table with a corrected rerun. We replaced the label-derived node features with random 64-dimensional node features and evaluated the same baseline methods used in the main table. Appendix H.2, Table 19 reports the full corrected comparison.
>
> | Method | Accuracy | $\Delta$ vs Full |
> |---|---:|---:|
> | Full | 47.31 ± 0.14 | - |
> | Random | 50.03 ± 1.21 | +2.72 |
> | EdgeDeg | 50.28 ± 1.32 | +2.97 |
> | Spectral | 48.73 ± 0.71 | +1.42 |
> | HSL | 49.16 ± 2.42 | +1.86 |
> | HSL w/ aug. | 49.54 ± 4.15 | +2.23 |
> | EHGNN-F | 49.35 ± 0.84 | +2.04 |
> | EHGNN-C | 51.46 ± 0.26 | +4.15 |
> | EHGNN-F(cond.) | 48.98 ± 1.01 | +1.67 |
> | EHGNN-C(cond.) | 50.59 ± 0.47 | +3.28 |
> | EHGNN-F(cond,LR) | **52.51 ± 0.56** | **+5.20** |
> | EHGNN-C(cond,LR) | 50.71 ± 0.89 | +3.41 |
>
> These results show that several sparsified models improve modestly over Full, with EHGNN-F(cond,LR) giving the best mean accuracy. The manuscript now uses only the corrected House results and does not use the original label-derived House row to support any empirical claim (Section 6, Table 2; Appendix H.2, Table 19).
>
> Reference(s):
>
> [1] Eli Chien, Chao Pan, Jianhao Peng, and Olgica Milenkovic. "You are AllSet: A Multiset Function Framework for Hypergraph Neural Networks." ICLR, 2022.
>
> ---
> **Please let us know if you have any further questions.**

---

### Review · Reviewer_a2kT · 2026-05-07

**Summary Of Contributions:**

The paper proposes **EdgeMask-HGNN**, a supervised hypergraph sparsification framework for HGNNs. The main idea is to learn a task-aware sparse subhypergraph under an explicit budget, rather than treating the observed hypergraph as fixed. The authors introduce two masking granularities: **fine-grained incidence-level masking** and **coarse-grained hyperedge-level masking**, both trained end-to-end with downstream supervision using weighted sampling without replacement and a straight-through estimator. They also introduce feature-conditioned and low-rank variants to improve generalization/scalability. The paper evaluates the method on semi-supervised node classification and link prediction, with emphasis on heterophilic hypergraphs. The paper addresses a real limitation of HGNNs: many constructed hypergraphs contain noisy or task-irrelevant higher-order relations. The experimental section is reasonably broad, and the distinction between incidence-level and hyperedge-level sparsification is meaningful.

---

## Strengths


S1. The paper correctly identifies that many real-world hypergraphs are constructed heuristically and may contain task-irrelevant or even harmful hyperedges. Treating the hypergraph as fixed is indeed a limitation for HGNNs, especially in heterophilic settings.

S2. The distinction between incidence-level masking and hyperedge-level masking is important for hypergraphs. Unlike ordinary graphs, a hyperedge can be corrupted partially or entirely, so fine-grained and coarse-grained pruning address different noise patterns. The controlled corruption experiment on Cora is a useful way to illustrate this distinction.

S3. The paper evaluates on many node-classification datasets, includes heterophilic benchmarks, reports homophily correlations, and includes link-prediction experiments. The node-classification table shows that EdgeMask-HGNN can outperform full training on several heterophilic/noisy datasets, while the authors also acknowledge that full training remains better on several homophilic datasets.


S4. The paper studies sparsity-performance tradeoffs, homophilic regimes, sampler choice, mask stability, and some runtime behavior. The mask-flip analysis is a good sanity check: the learned masks reportedly stabilize after early epochs, which partly addresses concerns about stochastic mask instability.


S5. The paper does not oversell the method as universally superior. It explicitly reports that full training remains stronger on highly homophilic datasets and that increasing the retention budget helps but may still not fully close the gap.

---

## Weaknesses

W1. The core mechanism—learnable edge/incidence masks, straight-through estimation, top-k or sampling-based sparsification—is conceptually close to existing supervised graph sparsification, learned edge dropout, hard attention, and graph structure learning methods. The extension to hypergraphs is nontrivial because of incidence structure, but the method itself appears relatively direct. The paper would be stronger if it more sharply distinguished itself from learned graph sparsification, Gumbel-top-k/Concrete masks, hard-concrete gates, and related task-aware pruning methods.


W2. The stability theorem essentially says that if logits change slowly, sampled masks change slowly. This is mathematically reasonable but not very informative about the central questions: whether the learned mask is optimal, whether it generalizes, whether the biased STE produces a good estimator, or whether the selected subhypergraph is task-relevant. The authors acknowledge that the result does not guarantee optimization, recovery of an optimal subhypergraph, or generalization performance, which limits its value.


W3. Several comparisons are potentially confounded. For example, HSL is modified by disabling augmentation, but HSL was originally designed as a structure-learning method with pruning and augmentation. Disabling part of the method may make the comparison less representative. Similarly, the spectral baseline appears expensive and sometimes OOM, but it is not clear whether the implementation is optimized or whether approximate spectral sparsification alternatives were considered.



W4. The link-prediction section reports very large gains, especially for EHGNN-F(cond,LR), e.g. WordNet AUC increasing from 87.92 to 99.64 and AUPR from 68.62 to 99.05. However, the proposed adaptation adds contextualized node embeddings using masked hypergraph aggregation before Hyper-SAGNN. This is not just sparsification; it also changes the model by injecting global hypergraph structure into a baseline that processes hyperedges independently. A fairer ablation would compare against the same contextual aggregation with no sparsification, full mask, random mask, and learned mask. Otherwise, the gains may be due to adding message passing/contextualization rather than EdgeMask itself.


W5. The paper reports EHGNN-F, EHGNN-C, conditioned versions, and low-rank conditioned versions. No single variant consistently dominates. In practice, this raises a model-selection issue: if the best variant is chosen post hoc per dataset, the reported gains may overstate robustness. The paper should specify a principled variant-selection protocol or recommend one default model.


W6. Because the method is supervised and learns the sparsifier using task labels, it may be sensitive to label scarcity. The paper uses a 50/25/25 split for node classification, which is relatively generous. A stronger evaluation would include low-label settings, such as 1%, 5%, 10%, and 20% labeled nodes, especially since the paper itself notes that supervised sparsification may not generalize well under limited labels.


W7. Incidence-level masking can create singleton hyperedges, empty hyperedges, isolated nodes, or invalid degree-normalization cases depending on the HGNN backbone. The paper should explicitly state how these cases are handled during training and inference. This is important for reproducibility and for understanding whether the method is robust or relies on implicit implementation details.

**Audience:**

Yes

**Audience Explanation:**

Yes. The paper would likely interest a subset of the TMLR audience, particularly researchers working on hypergraph neural networks, graph sparsification, graph structure learning, and heterophilic graph learning. The problem is relevant because many real-world hypergraphs are noisy or heuristically constructed, and task-aware sparsification is a plausible way to improve robustness and efficiency. However, the interest is likely to be somewhat specialized, since the paper focuses mainly on HGNNs and does not yet establish broader implications for general graph learning or structure learning.

**Broader Impact Concerns:**

I do not see major negative broader-impact concerns specific to this work. The paper mainly proposes a technical method for sparsifying hypergraphs in HGNNs. However, there are some moderate concerns worth noting. Since the sparsifier is supervised, it may learn to preserve structures that reflect biases in the training labels or in the original hypergraph construction. In sensitive applications such as recommendation, social networks, hiring, healthcare, or scientific decision-making, removing hyperedges deemed “task-irrelevant” could also remove minority, rare, or underrepresented relational patterns. The authors should therefore avoid overstating interpretability of the learned masks and should discuss robustness to biased labels, label scarcity, and distribution shift. A short broader-impact discussion covering these points would improve the paper.

**Claims And Evidence:**

Yes

**Claims Explanation:**

The claims are partially supported. The empirical evidence is reasonably broad and supports the narrower claim that supervised sparsification can improve HGNN performance in heterophilic/noisy settings. However, the evidence is not fully convincing for broader claims of general superiority, scalability, and link-prediction effectiveness. Some baselines require clarification, the link-prediction gains are confounded by architectural changes, and the theoretical result only establishes mask stability rather than task-level guarantees. The presentation is mostly clear, but the empirical story would be stronger with cleaner ablations, low-label experiments, and more precise baseline definitions.

**Requested Changes:**

The authors should clarify and strengthen several parts of the paper. First, the baseline definitions need to be made more precise, especially the random sparsification baseline, where the stated probability appears inconsistent with the incidence-level budget. Second, the link-prediction experiments require stronger ablations, since the proposed adaptation adds contextual hypergraph aggregation in addition to sparsification; the paper should isolate the effect of learned masking from the effect of adding global message passing. Third, the authors should evaluate low-label regimes, since the method depends on supervised signals and may be sensitive to label scarcity. Fourth, the paper should explain how incidence-level masking handles empty hyperedges, singleton hyperedges, isolated nodes, and normalization issues. Fifth, the comparison with HSL should be better justified, since disabling its augmentation component may not reflect the original method fairly. Finally, the paper should provide a clearer recommendation on which EdgeMask-HGNN variant should be used by default, as the many variants make the empirical conclusions harder to interpret.

---

> ### Author Response · Authors · 2026-05-20
> **Response (part 1)**
>
> Thank you for your detailed feedback.
>
> > W1: Relationship to learned graph sparsification, hard attention, and related masking methods
>
> We agree that the elementary ingredients we use (sigmoid retention weights, budgeted top-k sampling, and a masked straight-through estimator) are related to prior graph sparsification and discrete-mask learning. We revised the manuscript to make this explicit, to avoid presenting the estimator itself as novel (Section 4.1).
>
> The revised paper now draws a sharper distinction between the optimization tool and the modeled object. EdgeMask-HGNN uses existing differentiable-mask estimators to solve a budget-constrained hypergraph substructure-selection problem (Section 3). The hard forward-pass object is a subhypergraph in the original incidence representation, not a weighted graph adjacency, an attention distribution, or a pruned neural parameter set. We outlined 4 concrete differences from graph sparsification: the maskable unit can be either a whole hyperedge or an individual incidence; these two units have different higher-order semantics; they change HGNN cardinalities, degrees, and two-stage normalization differently; and clique expansion loses the distinction because one hyperedge becomes many pairwise edges. We added this distinction explicitly in the related-work and method sections (Sections 2 and 4.1).
>
> We also added a formal observation for the clique-expansion mismatch (Section 2). For a hyperedge of size $r>2$, clique expansion creates $\binom{r}{2}$ pairwise edges. Hyperedge deletion corresponds to deleting all of them as a group; incidence deletion corresponds to deleting the $r-1$ pairwise edges incident to one node inside that hyperedge while preserving the rest of the relation; generic graph-edge pruning allows many pairwise subsets that correspond to neither operation. Thus graph-edge sparsification on the clique expansion optimizes over a different feasible set from both the incidence-level and hyperedge-level programs.
>
> > W2: Stability theorem scope
>
> We agree that the stability theorem should not be read as an optimization, recovery, or generalization guarantee. We revised Section 5 to state this explicitly and to frame the result as a sampler-stability guarantee: if the learned logits change slowly, the expected number of sampled mask flips is controlled.
>
> We also added the missing connection to training stability. If $\|\mathbf{s}^{t+1}-\mathbf{s}^{t}\|_1\to 0$, then the expected number of mask flips also vanishes. This is not a convergence proof for the joint nonconvex problem, but it is a necessary consistency condition for settling on a stable learned subhypergraph. It also explains why deterministic top-k/top-$\kappa$ inference removes the residual sampling noise once the scorer has stabilized (Section 5).
>
> To complement the theorem, we added three empirical mask-quality evaluations (Appendix J.1, Tables 21-23). In controlled Cora edge corruption, EHGNN-C removes corrupted hyperedges better than Random and Degree, with corruption-removal recall $56.59\pm1.52\%$ and clean-score AUPRC $62.99\pm1.87\%$. In frozen-mask retraining, the learned mask remains predictive after reinitializing the HGNN. In the one-swap local optimality check, only $3.03\pm0.67\%$ of sampled EHGNN-F swaps and $8.07\pm1.93\%$ of sampled EHGNN-C swaps improve validation loss by more than $10^{-3}$.
>
> These additions still do not claim global optimality, but they do support the claim that the learned masks can recover known corrupted structure in a controlled setting, and remain useful beyond the jointly trained HGNN weights.

---

> ### Author Response · Authors · 2026-05-20
> **Response (part 2)**
>
> > W3: HSL and spectral baseline fairness
>
> To the best of our knowledge, there is no prior supervised hypergraph sparsifier that only removes observed incidences under an explicit global sparsity budget. HSL is the closest task-aware baseline, but the original method both prunes observed incidences and adds new ones through augmentation/discovery. Since EdgeMask-HGNN, Random, EdgeDeg, and Spectral all select subhypergraphs without adding incidences, the main table compares against the pruning part of HSL under the same 50% retention budget; this is the row labeled `HSL` in Table 2 (Section 6, Table 2).
>
> As recommended, we also implemented the residual incidence augmentation step and added it as a separate `HSL w/ aug.` row in the accuracy and memory tables (Section 6, Table 2; Appendix C, Table 9). We report its hyperedge sparsification ratios so the comparison is not confounded by under-sparsification:
>
> | Dataset | Ratio | Dataset | Ratio | Dataset | Ratio |
> |---|---:|---|---:|---|---:|
> | 20news | 0.53 | ModelNet40 | 0.43 | Mushroom | 0.49 |
> | NTU2012 | 0.48 | Actor | 0.54 | Citeseer | 0.46 |
> | Cora-CA | 0.46 | DBLP-CA | 0.58 | Cora | 0.45 |
> | House | 0.36 | Pokec | 0.57 | PubMed | 0.58 |
> | Twitch | 0.52 | Walmart | 0.54 | Trivago | 0.50 |
>
> We did not include Yelp because `HSL w/ aug.` runs out of memory. For the completed datasets, the ratios stay close to the nominal 50% budget, so the improved `HSL w/ aug.` accuracy is not simply due to retaining a much denser hyperedge set. The result also reinforces the distinction we now make in the paper: full HSL has an augmented feasible set, while EdgeMask-HGNN is a budgeted subhypergraph selector.
>
> For spectral sparsification, the appendix contains approximate effective-resistance results using a Hutchinson-type estimator and conjugate-gradient solves; we now point to Table 8 and the accompanying discussion in Appendix C.3.
>
> > W4: Link-prediction gains confounded by contextual aggregation
>
> Thank you for raising this important issue. Our Hyper-SAGNN adaptation adds contextual hypergraph aggregation in addition to sparsification, whereas plain Hyper-SAGNN processes each candidate hyperedge independently. We therefore ran the requested WordNet ablation (Appendix E, Table 12):
>
> | Model | Accuracy | AUC | AUPR |
> |---|---:|---:|---:|
> | Hyper-SAGNN | 88.32 ± 0.09 | 87.90 ± 0.20 | 68.66 ± 0.30 |
> | FullContext | **99.35 ± 0.03** | **99.94 ± 0.01** | **99.75 ± 0.01** |
> | Random masked context | 93.28 ± 0.11 | 95.81 ± 0.27 | 87.36 ± 0.56 |
> | EHGNN-F(cond,LR) | 98.75 ± 0.06 | 99.67 ± 0.02 | 98.99 ± 0.07 |
>
> This confirms the reviewer’s point: much of the WordNet gain comes from adding contextual aggregation. FullContext is strongest on this clean-context ablation, while learned masking remains substantially better than random masked context and plain Hyper-SAGNN at the same 50% retention setting (Appendix E, Table 12).
>
> We also added a targeted GPS analysis using the GPS link-prediction dataset from the original Hyper-SAGNN paper [1]. GPS is a 3-uniform heterogeneous hypergraph of (user, location, activity) triples with 146 users, 70 locations, 5 activities, and 1,436 observed hyperedges. Because the activity type has only five nodes, many triples share activity nodes, making GPS a dense context-overlap setting. EHGNN-F(cond,LR) uses a 75% context budget, while FullContext uses all context (Appendix E, Table 13).
>
> | Context-noise ratio | FullContext AUC | FullContext AUPR | Random AUC | Random AUPR | EHGNN-F(cond,LR) AUC | EHGNN-F(cond,LR) AUPR |
> |---:|---:|---:|---:|---:|---:|---:|
> | 0.0 | 84.58 ± 5.03 | 54.12 ± 8.88 | 84.58 ± 4.11 | 54.97 ± 6.68 | **85.27 ± 3.73** | **55.81 ± 6.05** |
> | 0.2 | 84.80 ± 4.32 | 54.89 ± 8.15 | 84.02 ± 4.68 | 54.34 ± 9.31 | **86.21 ± 1.79** | **58.31 ± 4.77** |
> | 0.4 | 83.06 ± 3.88 | 50.78 ± 6.92 | 84.12 ± 2.49 | 54.02 ± 5.63 | **84.36 ± 2.41** | **54.61 ± 4.84** |
> | 0.8 | 81.89 ± 5.60 | 51.25 ± 10.97 | 81.67 ± 5.73 | 50.59 ± 10.83 | **84.73 ± 3.49** | **55.25 ± 6.54** |
> | 1.5 | 82.57 ± 4.89 | 50.94 ± 9.67 | 82.98 ± 5.24 | 52.20 ± 9.63 | **84.33 ± 2.98** | **55.46 ± 6.51** |
> | 3.0 | 81.79 ± 4.85 | 50.07 ± 8.56 | 82.19 ± 4.81 | 51.21 ± 8.36 | **83.33 ± 3.35** | **53.53 ± 5.51** |
>
> These GPS results show that FullContext does not always subsume learned sparsification: in this dense context-overlap setting, EHGNN-F(cond,LR) with a 75% context budget achieves higher mean AUC/AUPR than FullContext under both clean and noisy context (Appendix E, Table 13). At the same time, the WordNet ablation shows that FullContext can be the strongest clean-context model (Appendix E, Table 12). We therefore revised the claim to be conditional rather than absolute: FullContext is a strong no-sparsification control, while EHGNN is useful when the task benefits from selecting a subset of contextual hyperedges (Section 6, Table 5; Appendix E, Tables 12 and 13; Section 7).

---

> ### Author Response · Authors · 2026-05-20
> **Response (part 3)**
>
> > W5: Too many variants and model selection
>
> We added a default variant, budget-selection guideline, and practical variant-selection table (Section 6, Table 3). The 50% retention rate was chosen as a common controlled compression point so that learned, random, degree-based, spectral, and HSL can be compared under the same nominal budget. It was not tuned separately for each dataset. To address budget selection in practice, we added a validation-based procedure: choose the retention rate over a small grid such as $\\{0.3,0.5,0.7,0.9\\}$, using validation accuracy rather than test performance. In addition, we now explicitly note that a natural adaptive alternative would replace the hard budget with a Lagrangian $L\_0$-style penalty, e.g., $\mathcal{L}\_{task}+\lambda\|M\|\_0$ or $\mathcal{L}\_{task}+\lambda\|m\|\_0$, using differentiable hard-concrete style gates; we leave such joint budget learning to future work (Section 6 and Section 7).
>
> > W6: Limited-label sensitivity
>
> We added limited-label experiments at 1%, 5%, 10%, and 20% training labels (Appendix H.1, Table 18). The main question is whether the learned sparsifier collapses when supervision is scarce. In the evaluated settings, it does not: on two representative datasets, the learned mask outperforms both Full and Random at every low-label ratio, and on Cora, it remains consistently better than Random at the same budget.
>
> **Limited-label node classification results with 50% retention.**
>
> | Dataset | Train labels | Full | Random | EHGNN-F | EHGNN-C |
> |---|---:|---:|---:|---:|---:|
> | Actor | 1% | 58.80 ± 0.06 | 67.63 ± 0.50 | **69.05 ± 0.10** | 67.80 ± 0.30 |
> | Actor | 5% | 62.32 ± 0.03 | 74.53 ± 0.55 | **75.70 ± 0.04** | 74.45 ± 0.10 |
> | Actor | 10% | 63.36 ± 0.03 | 76.00 ± 0.37 | **76.45 ± 0.05** | 75.29 ± 0.12 |
> | Actor | 20% | 64.20 ± 0.02 | 77.06 ± 0.21 | **77.07 ± 0.07** | 76.19 ± 0.04 |
> | Cora | 1% | **47.61 ± 0.29** | 34.33 ± 5.34 | 45.70 ± 0.45 | 46.05 ± 0.21 |
> | Cora | 5% | **64.82 ± 0.61** | 58.47 ± 1.16 | 61.88 ± 0.20 | 63.81 ± 0.38 |
> | Cora | 10% | **71.77 ± 0.15** | 64.66 ± 0.68 | 67.93 ± 0.37 | 69.01 ± 0.25 |
> | Cora | 20% | **75.99 ± 0.27** | 70.19 ± 0.94 | 73.05 ± 0.18 | 72.75 ± 0.38 |
> | ModelNet40 | 1% | 85.92 ± 0.05 | 86.07 ± 0.95 | **86.57 ± 0.03** | 85.55 ± 0.08 |
> | ModelNet40 | 5% | 93.59 ± 0.09 | 94.39 ± 0.08 | **94.80 ± 0.04** | 93.58 ± 0.14 |
> | ModelNet40 | 10% | 94.15 ± 0.03 | 95.15 ± 0.16 | **95.40 ± 0.08** | 94.13 ± 0.17 |
> | ModelNet40 | 20% | 94.45 ± 0.05 | 95.56 ± 0.22 | **95.79 ± 0.03** | 95.17 ± 0.06 |
>
> These results suggest that supervised mask learning can remain useful even with very limited labels (Appendix H.1, Table 18; Section 7).
>
> > W7: Empty hyperedges, singleton hyperedges, isolated nodes, and normalization
>
> We added a paragraph "Degenerate hyperedges during sparsification" under Section 4.4 that discusses these cases. Singleton hyperedges remain well-defined in the bipartite incidence representation; thus, they are handled by the same message-passing rule as other retained hyperedges. Empty hyperedges simply have no retained incidences and therefore contribute no messages. In the implementation, message passing is carried out over the retained incidence list; if the retained list is empty, the layer returns a zero aggregate with the learned bias, and inverse degree normalizers that would otherwise divide by zero are set to zero. Thus isolated nodes receive no hypergraph message from the sparsified structure rather than producing invalid normalization values.

---

> ### Author Response · Authors · 2026-05-20
> **Response (part 4)**
>
> **On Recommended Changes**
>
> > 1. Baseline definitions
>
> We clarified Full, Random, EdgeDeg, Spectral, HSL, and `HSL w/ aug.`; Random is now defined at the incidence-pair level using the target retention ratio (Section 6, Table 2).
>
> > 2. Link-prediction ablations
>
> We added FullContext and random masked-context controls, plus a GPS context-budget/noise analysis (Appendix E, Tables 12 and 13).
>
> > 3. Low-label regimes
>
> We added 1%, 5%, 10%, and 20% label experiments on Actor, Cora, and ModelNet40 (Appendix H.1, Table 18).
>
> > 4. Degenerate cases
>
> We added how incidence masking handles singleton hyperedges, empty hyperedges, isolated nodes, and zero-degree normalization (Section 4.4).
>
> > 5. HSL comparison
>
> We clarified the pruning-oriented HSL row and added `HSL w/ aug.` with residual incidence augmentation and per-dataset sparsification ratios (Section 6, Table 2; Appendix C, Table 9).
>
> > 6. Recommended variant
>
> We added default-variant guidance, validation-based budget selection, and an incidence-scale variant table (Section 6, Table 3; Figure 5).
>
> ``Reference(s):``
>
> [1] Ruochi Zhang, Yuesong Zou, and Jian Ma. "Hyper-SAGNN: a self-attention based graph neural network for hypergraphs." International Conference on Learning Representations (ICLR), 2020.
>
> -----
> **Please let us know if you have any further questions.**

---

### Author Response · Authors · 2026-05-20
**Core Changes to the paper**

We thank the reviewers for taking the time to read our paper and provide valuable feedback. In the revision, we have narrowed several claims, added missing controlled experiments, and clarified what EdgeMask-HGNN does and does not establish. Revisions in the updated manuscript are colored in $\color{blue} \text{blue}$. The most important changes are:

- **[Reviewer EYAy]** We rewrote the problem statement as an explicit joint constrained optimization over HGNN parameters and a binary incidence or hyperedge mask (Section 3).
- **[Reviewer a2kT; Reviewer tSxH]** We clarified the relationship to learned graph sparsification, hard attention, discrete-mask estimators, and clique-expansion graph pruning (Sections 2 and 4.1).
- **[Reviewer a2kT]** We added reviewer-requested ablations for link prediction, including a full-context/no-sparsification control and a context-budget/noise analysis on a new dataset GPS  (Appendix E, Tables 12 and 13).
- **[Reviewer EYAy]** We added generalization analyses and controlled noisy-hyperedge experiments to sharpen the motivation (Appendix G, Tables 15-17).
- **[Reviewer a2kT; Reviewer tSxH]** We added limited-label node-classification experiments at 1%, 5%, 10%, and 20% labels (Appendix H.1, Table 18).
- **[Reviewer EYAy]** We re-inspected the House dataset and found label leakage through label-derived node features. We replaced the main House results with a corrected label-independent rerun (Section 6, Table 2; Appendix H.2, Table 19).
- **[Reviewer a2kT; Reviewer tSxH]** We clarified baseline definitions, including Random and HSL, and added validation-based budget and variant-selection guidance (Section 6, Tables 2 and 3; Figure 5).
- **[Reviewer a2kT; Reviewer tSxH]** We added `HSL w/ aug.` to the accuracy and memory tables to address the original HSL augmentation setting (Section 6, Table 2; Appendix C, Table 9).
- **[Reviewer a2kT; Reviewer EYAy; Reviewer tSxH]** We softened overbroad claims in the theory, motivation, and link-prediction discussions to match what the evidence supports (Sections 5-7 and Appendices E and J).